# Integrating antigen capturing nanoparticles and type 1 conventional dendritic cell therapy for in situ cancer immunization

Chih-Jia Chao[1], Endong Zhang[1], Duong N. Trinh[1], Edidiong Udofa [1], Hanchen Lin[2], Caylee Silvers[2], Jiawei Huo [2], Shan He[1], Jingtian Zheng[1], Xiaoying Cai[1], Qing Bao [1], Luyu Zhang[1], Philana Phan[1], Sara M. Elgendy [1], Xiangqian Shi[1], Joanna E. Burdette [1,3], Steve Seung-Young Lee [1,3], Yu Gao[1,3], Peng Zhang [2] & Zongmin Zhao [1,3] ✉

Eliciting a robust immune response against tumors is often hampered by the inadequate presence of effective antigen presenting cells and their suboptimal ability to present antigens within the immunosuppressive tumor microenvironment. Here, we report a cascade antigen relay strategy integrating antigen capturing nanoparticles (AC-NPs) and migratory type 1 conventional dendritic cells (cDC1s), named Antigen Capturing nanoparticle Transformed Dendritic Cell therapy (ACT-DC), to facilitate in situ immunization. AC-NPs are engineered to capture antigens directly from the tumor and facilitate their delivery to adoptively transferred migratory cDC1s, enhancing antigen presentation to the lymph nodes and reshaping the tumor microenvironment. Our findings suggest that ACT-DC improves in situ antigen collection, triggers a robust systemic immune response without the need for exogenous antigens, and transforms the tumor environment into a more "immune-hot" state. In multiple tumor models including colon cancer, melanoma, and glioma, ACT-DC in combination with immune checkpoint inhibitors eliminates primary tumors in 50-100% of treated mice and effectively rejects two separate tumor rechallenges. Collectively, ACT-DC could provide a broadly effective approach for in situ cancer immunization and tumor microenvironment modulation.

Cancer immunotherapy marks a transformative development in cancer treatment. By activating the body's immune system to fight malignant cells, cancer immunotherapy has the potential to induce lasting immunological memories, thereby preventing tumor recurrence[1–6]. Various types of immunotherapies, including immune checkpoint inhibitors and adoptive cell therapies, have shown promising efficacy in the clinic in extending overall and progression-free survival, offering new hope to patients who did not benefit from conventional interventions like chemotherapy[7–21].

Inducing a strong immune response against tumors hinges on effectively activating a complex series of cellular and molecular interactions in the cancer-immunity cycle (CIC)[22,23]. Dendritic cells (DCs), a key type of antigen-presenting cells (APCs), play an essential role in the CIC. DCs capture tumor-associated antigens and present them to T cells, inducing antigen-specific T cells and augmenting T cell-mediated tumor killing[24–26]. Due to DC's crucial role in the CIC, DC-based cell therapies, especially DC vaccines that involve the adoptive transfer of antigen-pulsed DCs, have been explored as a promising

[1]Department of Pharmaceutical Sciences, University of Illinois Chicago, Chicago, IL, USA. [2]Department of Neurological Surgery, Lou and Jean Malnati Brain Tumor Institute, Northwestern University Feinberg School of Medicine, Chicago, IL, USA. [3]University of Illinois Cancer Center, Chicago, IL, USA. ✉e-mail: zhaozm@uic.edu

immunotherapeutic approach[27–31]. In contrast to genetically engineered cell therapies such as chimeric antigen receptor (CAR) T cells and CAR natural killer (NK) cells, which are tailored to target a single tumor antigen, DC-based cell therapies can be used to tackle different tumor antigens, which can more effectively counteract the tumor's antigen heterogeneity and may be personalized with different patients[24,28]. However, despite extensive studies, ex vivo engineered DC therapies have only shown modest clinical efficacy, largely due to their suboptimal activity induced by the ex vivo preparation process and their inability to effectively counteract the tumor's immunosuppressive microenvironment[32,33]. A distinguishing challenge is the poor match between the antigens displayed by the DC therapies and the heterogenous antigens within the patient's tumor, due to the high inter- and intra-patient tumor antigenic heterogeneity as well as antigen loss during the ex vivo process, which collectively compromises the effectiveness of DC-based cell therapies in generating a broad T cell response[28,34,35]. Additionally, the effectiveness of ex vivo DC-based cell therapies is further hampered by limited intratumoral infiltration of T cells because of their inability to alter the tumor's immunosuppressive microenvironment. In situ immunization, which leverages native tumor-derived antigens directly within the tumor, holds potential to enhance antigen coverage and stimulate systemic antitumor immunity against heterogenous tumor antigens[36–41]. However, inducing a robust immune response via in situ immunization is hampered by the lack of effective DC subtypes in the tumor and their suboptimal capability for antigen presentation within the immunosuppressive tumor microenvironment.

Here, we develop a cascade antigen relay strategy based on the integration of antigen-capturing nanoparticles (AC-NPs) and migratory CD103+ type 1 conventional dendritic cells (cDC1s), which we refer to as Antigen-Capturing nanoparticle Transformed Dendritic Cell therapy (ACT-DC), for effective in situ immunization and remodeling of the tumor microenvironment. Emerging evidence suggests that migratory CD103+ cDC1s are a crucial DC subtype that most effectively present tumor antigens to CD8 T cells, with other critical functions in regulating cell-cell interactions within the tumor microenvironment[42–47]. However, these cells are poorly present in the tumor, and their antigen-capturing ability is significantly restricted by the immunosuppressive tumor microenvironment[46]. ACT-DC employs the adoptive transfer of migratory CD103+ cDC1s to increase their overall frequency and alter their spatial distribution within the tumor. This is coupled with the use of an AC-NP that directly collects antigens from the tumor and in situ delivers them to these cDC1s, facilitating antigen presentation to activate CD8 T cells in the lymph nodes. Moreover, activation of cDC1s by AC-NPs also transforms the local tumor microenvironment into an "immune-hot" state, facilitating immune cell infiltration and tumor eradication. Our results show that ACT-DC, especially when combined with immune checkpoint inhibitors, leads to the effective eradication of primary tumors across multiple tumor models, including colon cancer, melanoma, and glioma. Moreover, this strategy results in a robust immune memory, leading to the rejection of distant tumors after two separate tumor rechallenges. These findings highlight ACT-DC's capability to harness local tumor antigens for systemic antitumor immune response induction, offering a broadly effective strategy for in situ cancer immunization and tumor microenvironment modulation.

## Results

### Engineering AC-NPs for efficient capture of tumor antigens

A crucial element of the ACT-DC approach is the utilization of AC-NPs, designed to capture tumor antigens in situ, transport them to adoptively transferred cDC1s, and activate these DCs. Given that many tumor antigens possess negative charges and/or hydrophobic sequences[48–50], we postulated that a hydrophobic nanoparticle (NP) with a positively charged surface would efficiently capture such antigens through both electrostatic and hydrophobic interactions. To implement this concept, we synthesized a polymer-based composite AC-NP using acid-ended poly(lactic-co-glycolic) acid (PLGA) and polyethylenimine (PEI). In this design, PLGA imparts hydrophobicity to the NP, while PEI contributes a positive surface charge. As controls, we created two additional NPs, NP$^{Neg}$ and NP$^{PEG}$. NP$^{Neg}$, derived from acid-ended PLGA, features a negatively charged hydrophobic surface, while NP$^{PEG}$, constructed from PEGylated PLGA, has a hydrophilic surface. To effectively activate cDC1s, we encapsulated polyinosinic:polycytidylic acid (PIC), a toll-like receptor 3 (TLR3) agonist[51,52], in all three types of NPs.

AC-NP exhibits a uniform spherical shape with an average diameter of 160 nm (Fig. 1a–c) and a zeta potential of +41.2 mV (Fig. 1d). The energy-dispersive X-ray spectroscopy (EDS) analysis revealed that blank AC-NP shows a characteristic nitrogen peak derived from PEI in its elemental spectrum (Supplementary Fig. 1), further indicating the successful incorporation of PEI into AC-NP. NP$^{Neg}$ and NP$^{PEG}$ share a similar size (~153 nm) with AC-NP but carry a negative charge (Fig. 1b–d). PIC were encapsulated into AC-NPs with a loading capacity of 128 μg/mg (Supplementary Table 1). The release of PIC from AC-NPs followed a sustained pattern, with a cumulative 45.5% of drug released over 72 h (Supplementary Fig. 2). Subsequently, we assessed AC-NP's ability to capture tumor proteins from MC38 tumor lysates. Under all tested conditions, AC-NP indeed captured a significantly higher amount of tumor proteins compared to NP$^{Neg}$ and NP$^{PEG}$ (Fig. 1e). This is further evidenced by AC-NP's substantial increase in NP size and shift of surface charge from positive to neutral upon incubation with increasing amounts of tumor lysate (Supplementary Fig. 3). Similar results were observed using ovalbumin (OVA) as a model tumor antigen (Supplementary Fig. 4), reaffirming AC-NP's robust protein-capturing capability. To further determine the composition of the tumor proteins captured by different NPs, we conducted a proteomic analysis. While NP$^{Neg}$ and NP$^{PEG}$ only captured around 640 and 500 proteins, respectively, AC-NP captured a broader range (~800) of proteins (Fig. 1f and Supplementary Fig. 5). Notably, the proteins captured by AC-NP included several frequently mutated proteins in MC38 tumor cells, such as Hnrnpf, Aatf, Copb2, and Kpna6 (Fig. 1g)[53,54]. These proteins have the potential to generate neoantigens in vivo, contributing to the development of a tumor-specific immune response. Additionally, AC-NP also captured a higher number and/or quantity of damage-associated patterns (DAMPs) than NP$^{Neg}$ and NP$^{PEG}$ (Fig. 1h). As endogenous danger signals, the captured DAMPs have the potential to bind to pattern recognition receptors[55,56] and activate cDC1s. Overall, these data demonstrate that the engineered AC-NP has a robust capability to concentrate and capture tumor-derived proteins. It is important to note that AC-NPs do not selectively capture only frequently mutated proteins or DAMPs; instead, they capture a broad spectrum of tumor proteins, which include some frequently mutated proteins and DAMPs.

### AC-NPs boost the delivery of antigen to and activation of CD103+ cDC1s

CD103+ cDC1s, a distinctive subset of DCs, are notable for their robust migratory capacity to lymph nodes, facilitating effective antigen presentation[44,45]. Previous research has established the pivotal role of cDC1 infiltration in fostering a potent antitumor immune response[44,45]. However, their presence within tumors is typically limited, and their activity is hindered by the immunosuppressive tumor microenvironment[46]. The ACT-DC approach combines immunoactive AC-NPs with the adoptive transfer of CD103+ cDC1s. This cascade antigen relay strategy modulates the tumor microenvironment, initiating a sequential process that amplifies antigen presentation and enhances cDC1 activity, ultimately enabling potent in situ immunization.

We first adapted a robust method for generating CD103+ cDC1s from bone marrow, which involves the use of granulocyte macrophage

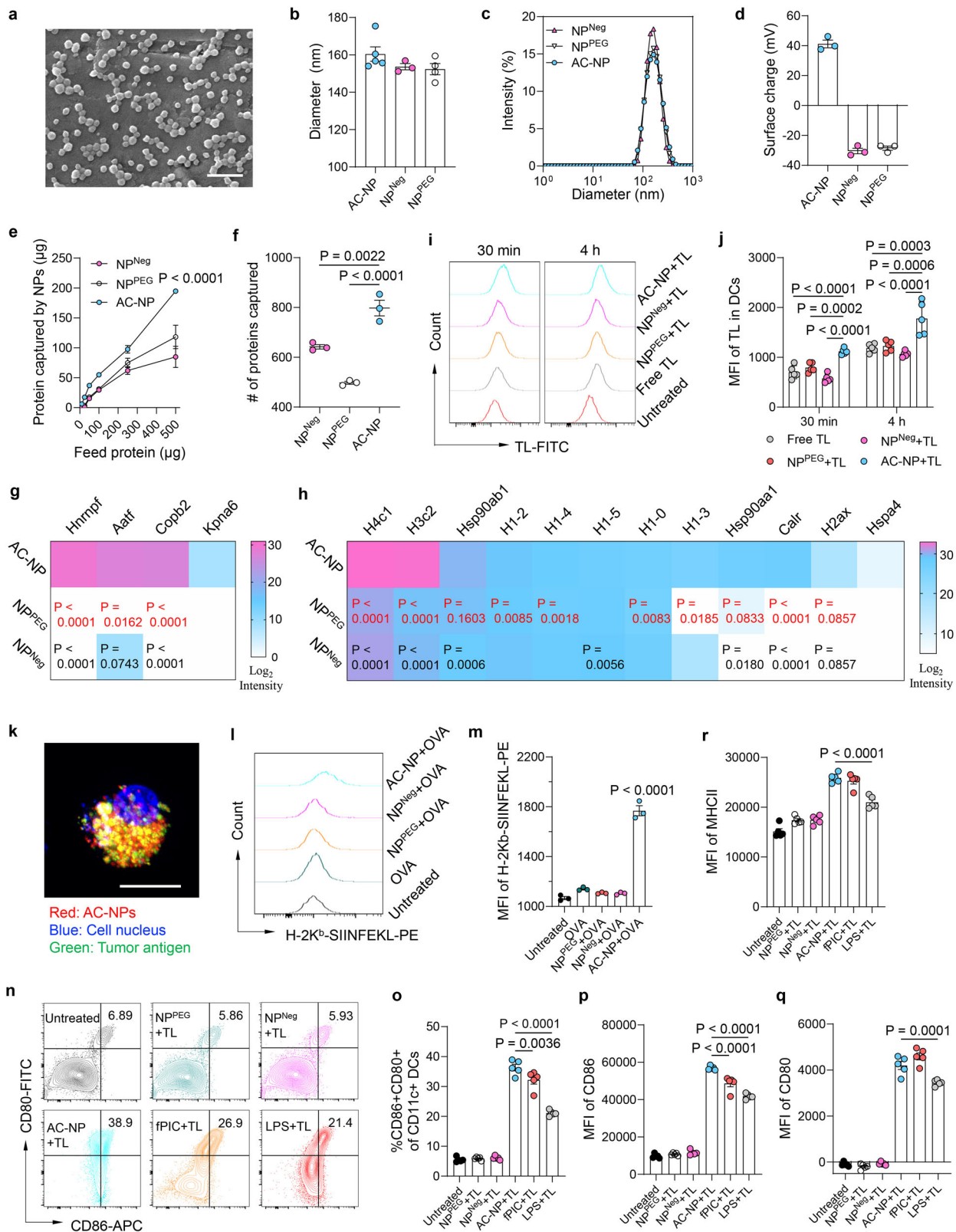

colony-stimulating factor (GM-CSF) and FMS-like tyrosine kinase 3 ligand (FLT3L)[57]. cDC1s obtained using this method exhibited the characteristic dendritic morphology upon activation by PIC (Supplementary Fig. 6a). They also displayed higher expression levels of CD103 and Clec9A, distinctive surface markers of cDC1s[57], than conventional bone marrow-derived DCs (BMDCs) (Supplementary Fig. 6b, c). Specifically, 92.6% of the obtained cells are CD103-positive,

indicating that CD103+ cDC1s constitute the majority, while other DCs, such as cDC2s and monocyte-derived DCs, represent only a small fraction. AC-NP at a concentration up to 1 mg/mL, did not show obvious toxicity to the obtained CD103+ cDC1s (Supplementary Fig. 7). We then assessed the efficacy of AC-NP in enhancing the delivery of tumor antigens to CD103+ cDC1s. Our flow cytometry results revealed that AC-NP significantly increased the uptake of tumor proteins by

**Fig. 1 | AC-NPs efficiently capture tumor antigens and enhance antigen delivery to CD103+ cDC1s. a** Scanning electron microscopic (SEM) images of AC-NPs. Scale bar: 500 nm. Representative image from two independent experiments. **b**–**d** Physicochemical properties of AC-NPs, including particle diameter ($n = 5$ for AC-NP, $n = 3$ for NP$^{Neg}$, $n = 4$ for NP$^{PEG}$, independent samples) (**b**), size distribution (**c**), and surface charge ($n = 3$, independent samples) (**d**). **e** Tumor protein binding capability of AC-NPs ($n = 3$, independent samples). **f**–**h** Analysis of the proteins captured by AC-NPs from MC38 tumor lysates. **f** Number of captured unique proteins ($n = 3$, independent samples). **g** Relative quantity of captured frequently mutated proteins ($n = 3$, independent samples). **h** Relative quantity of captured DAMPs ($n = 3$, independent samples). **i**–**k** AC-NPs enhanced antigen delivery to CD103+ cDC1s. **i** Representative flow cytometry plots showing the uptake of FITC-labeled tumor lysate (TL-FITC) into cDC1s assisted by AC-NPs after 30 min or 4 h incubation. **j** Mean fluorescence intensity (MFI) of FITC-tumor lysate in cDC1s ($n = 5$, biologically independent samples). **k** CLSM image showing AC-NP assisted tumor antigen delivery into cDC1s after 4 h incubation. Scale bar: 10 µm.

Representative image from three independent experiments. **l**, **m** AC-NPs enhanced antigen presentation on cDC1s. **l** Representative flow cytometry plot showing the expression of H-2Kb-SIINFEKL on cDC1s using OVA as a model tumor antigen. **m** Relative quantity of H-2Kb-SIINFEKL expressed on cDC1s ($n = 3$, biologically independent samples). **n**–**r** AC-NPs efficiently activated cDC1s ($n = 5$, biologically independent samples). **n** Representative flow cytometry plots showing the expression of activation markers (CD80 and CD86) on cDC1s treated with tumor lysate and NPs for 24 hrs. **o** Percentage of CD80 + CD86+ double-positive cDC1s. **p**–**r** Relative expression of DC activation markers including CD86 (**p**), CD80 (**q**), and MHCII (**r**). For **b**, **d**–**f**, **j**, **m**, **o**–**r**, data were presented as mean values ± SEM. Statistical analysis for (**e**): two-way ANOVA followed by Dunnett test. $P < 0.0001$ as compared to NP$^{PEG}$ and NP$^{Neg}$. Statistical analysis for (**f**–**h**, **j**, **m**, **o**–**r**): one-way ANOVA followed by Dunnett test. For **g**, **h**, black colored $p$ values compare AC-NPs to NP$^{Neg}$, and red colored $p$ values compare AC-NPs to NP$^{PEG}$. Source data are provided as a Source Data file.

cDC1s at both 30 minutes and 4 h post-incubation, while the control NPs (NP$^{Neg}$ and NP$^{PEG}$) showed limited efficacy (Fig. 1i, j and Supplementary Fig. 8). Further confocal fluorescence (CLSM) imaging revealed substantial co-localization of tumor lysate proteins with AC-NPs inside cDC1s after a 4-hour AC-NP treatment (Fig. 1k and Supplementary Fig. 9), suggesting that the enhanced protein uptake is likely attributed to AC-NP's ability to capture tumor proteins. Increased accumulation of tumor antigens within cDC1s is crucial for subsequent antigenic peptide presentation on DC surfaces, a key step for DC-T cell crosstalk and the activation of antigen-specific T cells. Indeed, using ovalbumin (OVA) as a model tumor antigen, we observed that AC-NP resulted in a significantly higher level of H-2Kb-SIINFEKL (OVA peptide) presented on cDC1 surface compared to free OVA (with or without free PIC) or free OVA plus control NPs (Fig. 1l, m and Supplementary Fig. 10). Additionally, AC-NP caused increased cell death in doxorubicin-treated MC38 tumor cells (Supplementary Fig. 11b), likely due to their vulnerability to NP-binding, which could induce cell membrane destabilization or rupture. While healthy MC38 cells appear resistant to these effects (Supplementary Fig. 11a), doxorubicin-treated cells are more sensitive. Notably, NP$^{PEG}$ caused less cell death than AC-NP or NP$^{Neg}$, likely because PEGylation reduces NP interaction and binding with cells, thereby minimizing binding-induced membrane destabilization. The enhanced cell death induced by AC-NP may increase antigen release and facilitated the internalization of MC38 cells to co-cultured cDC1s (Supplementary Fig. 12). Moreover, when co-incubated with a mixture of cells which were dissociated from an MC38 tumor, AC-NPs were more efficiently taken up by cDC1s than by cDC2s (Supplementary Fig. 13). Next, we evaluated the ability of AC-NP to activate cDC1s. AC-NP induced approximately a sixfold increase in the percentage of CD80+ CD86+ double-positive activated cDC1s, along with significantly higher expression of individual activation markers, compared to NP$^{Neg}$ and NP$^{PEG}$ (Fig. 1n–r). The robust efficiency of AC-NP in activating DCs may be partly attributed to its stronger ability to capture DAMPs. Overall, our data demonstrated that, owing to its efficient capture of tumor proteins, AC-NP enhances tumor antigen delivery to and presentation on cDC1s while promoting efficient cDC1 activation.

## ACT-DC migrates to tumor-draining lymph nodes (tDLNs) and activates cDC1s

While CD103+ cDC1s play a crucial role in initiating and enhancing antitumor immune response, their presence within tumors has been shown to be notably limited, partly due to the immunosuppressive characteristics of the tumor microenvironment[46]. We first measured the abundance of endogenous CD103+ cDC1s within the tumor microenvironment in the MC38 tumor model. As shown in Fig. 2a, while CD11c+ DCs are distributed both in the marginal and central regions of the tumor, CD103+ cDC1s are primarily localized at the

tumor periphery. Quantitative analysis revealed that endogenous CD103+ cDC1s only constitute <1.5% of total CD45+ cells within the MC38 tumor (Fig. 2b, c). These findings provide a rationale for incorporating CD103+ cDC1s into the ACT-DC approach. Indeed, ACT-DC resulted in a 5.8-fold increase in the total number of CD103+ cDC1s within the tumor, 6 hours after intratumoral administration (Fig. 2b, c). Additionally, ACT-DC also induced a shift in the spatial distribution of cDC1s within the tumor. Unlike in an untreated tumor where cDC1s predominantly accumulate at the tumor margin, in ACT-DC treated tumors, cDC1s show a more broad distribution throughout both the marginal and central areas (Fig. 2d). This altered spatial distribution, coupled with the increased abundance of cDC1s facilitated by ACT-DC, may reshape the tumor microenvironment into a "hot" state, leveraging native tumor antigens with the assistance of AC-NPs. Ultimately, this approach could enable in situ immunization, triggering a robust antitumor immune response. Notably, in all studies involving the injection of ACT-DC, AC-NPs were administered first, followed by cDC1s, with a 15-minute interval between the two injections. The ACT-DC approach hinges on the incorporation of AC-NPs, facilitating their hitchhiking onto adoptively transferred CD103+ cDC1s. This process triggers the subsequent transport and presentation of in situ captured tumor antigens to the tDLNs. To prove this mechanism, we determined the cellular-level distribution of AC-NPs upon intratumoral administration, either in their free form or as part of the ACT-DC formulation (Fig. 2e and Supplementary Fig. 14). As shown in Fig. 2f, g, 6 h after intratumoral injection, AC-NPs in the free form were mainly distributed in tumor cells (non-CD45+ cells) and tumor-resident myeloid cells (CD11b + CD11c- cells). In contrast, when integrated into the ACT-DC approach, most of the AC-NPs were associated with the adoptively transferred cDC1s, though they were also present in tumor cells and tumor-resident myeloid cells at a lower frequency. Moreover, in a separate study, where a model tumor antigen (AF488-OVA), AC-NPs, and cDC1s were sequentially administered intratumorally (Supplementary Fig. 15a), we observed a markedly higher uptake of AF488-OVA by the adoptively injected cDC1s compared to other resident cells in the tumor (Supplementary Fig. 15b). These findings indicate that AC-NPs efficiently target the adoptively injected cDC1s in the ACT-DC approach, crucial for the cascade antigen relay mechanism facilitating in situ immunization enabled by ACT-DC.

Next, we evaluated the efficiency of ACT-DC in trafficking and delivering tumor antigens to the tDLNs. The comparison between 20 and 6 h after intratumoral administration of ACT-DC revealed an increase in the quantity of injected cDC1s in the tDLNs and a decrease in the tumor (Fig. 2h, i and Supplementary Fig. 16). This result indicates that ACT-DC efficiently traffics from the tumor to the tDLNs. Notably, a substantial number of injected cDC1s remained in the tumor 20 h post-administration. In addition, ACT-DC demonstrated a trend toward higher accumulation in tDLNs compared to BMDCs plus AC-NPs,

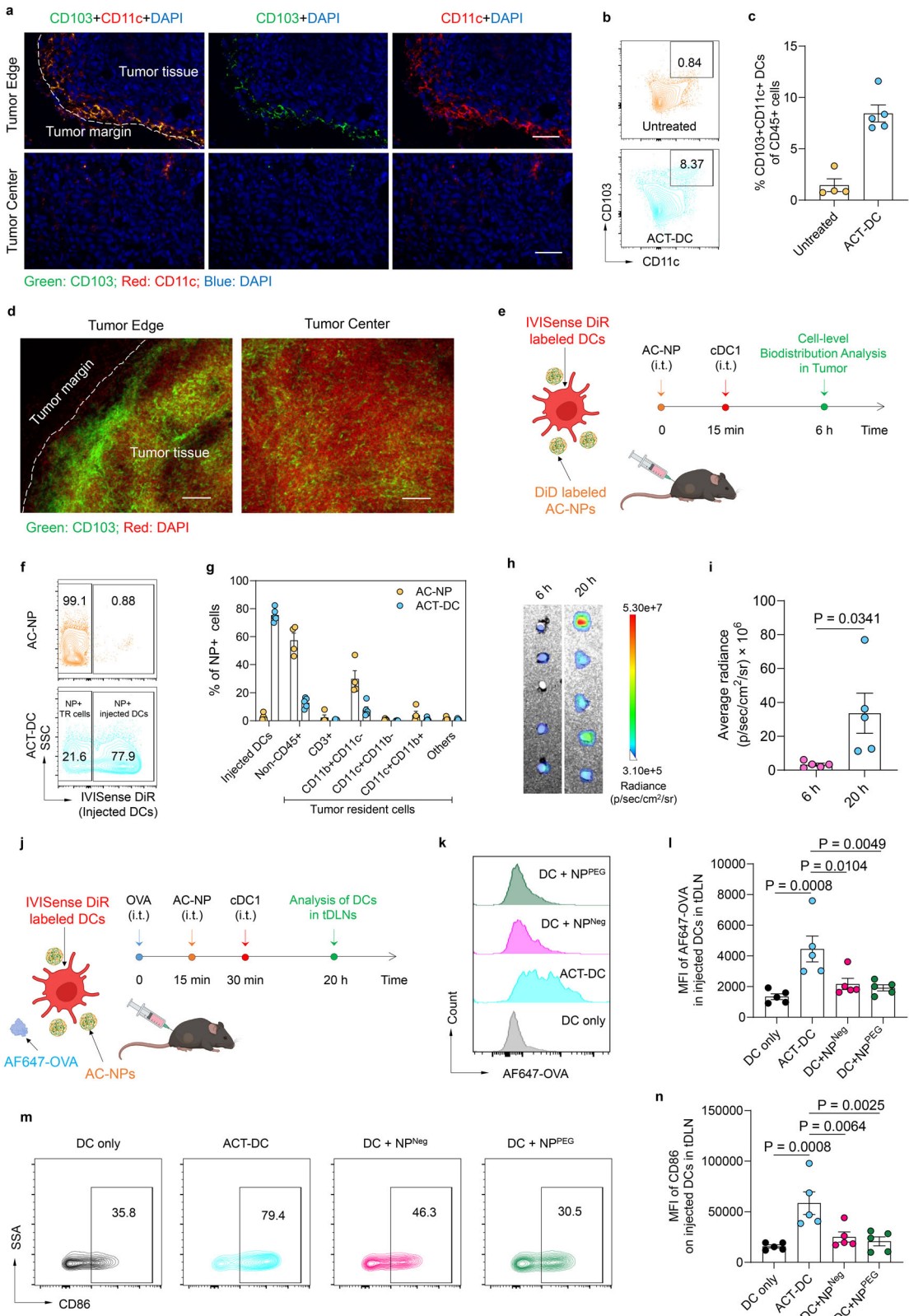

indicating the stronger migratory capability of CD103+ cDC1s (Supplementary Fig. 17).

We subsequently assessed the intratumoral retention of doxorubicin (a chemotherapeutic drug to enhance antigen release), AC-NP, and cDC1 over time following serial intratumor administration (Supplementary Fig. 18a). Upon injection, doxorubicin rapidly diffused from the injection site, and by 24 h post-injection (prior to ACT-DC administration), its signal was almost undetectable (Supplementary Fig. 18b, c). In contrast, cDC1 and AC-NP exhibited prolonged retention at the injection site (Supplementary Fig. 18d–f), with their signals gradually declining over >7 days and becoming undetectable by day 9. Additionally, further imaging of tDLNs revealed that the intratumorally injected cDC1s remained detectable in tDLNs even 9 days after ACT-DC administration (Supplementary Fig. 18g).

**Fig. 2 | ACT-DC captures AC-NPs in situ and migrates to the tumor-draining lymph nodes. a** CLSM images of MC38 tumors at the margin and center regions. Scale bars: 50 μm. Green: anti-CD103 (Alexa Fluor 647); Red: anti-CD11c (FITC); Blue: nucleus (DAPI). Representative of two independent experiments. **b** Flow plots showing CD103+ cDC1s in untreated or ACT-DC-treated MC38 tumors. **c** Percentage of CD103+ cDC1s in untreated ($n = 4$, biologically independent animals) or ACT-DC-treated ($n = 5$, biologically independent animals) MC38 tumors. **d** CLSM images of MC38 tumors 6 h after ACT-DC intratumoral administration. Scale bars: 100 μm. Green: anti-CD103 (Alexa Fluor 647); Red: nucleus (DAPI). Representative of two independent experiments. **e–g** Cell-level distribution of AC-NPs. **e** Schematic of experimental design. **f** Flow plots showing AC-NP distribution in tumor-resident (TR) cells and injected CD103+ cDC1s. **g** Distribution of AC-NPs in different cells in the tumor, 6 h after AC-NPs or ACT-DC injection ($n = 4$ for AC-NP, $n = 5$ for ACT-DC, biologically independent animals). AC-NPs and CD103+ cDC1s were labeled with DiD and IVISense DiR, respectively. **h, i** ACT-DC trafficking to tDLNs. **h** LagoX images of tDLNs, 6- or 20-h post ACT-DC injection. **i** Quantification of injected CD103+ cDC1s migrated to tDLNs ($n = 5$, biologically independent mice). CD103+ cDC1s were labeled with IVISense DiR. **j–n** AC-NPs enhanced antigen uptake and activation of transferred CD103+ cDC1s. **j** Schematic of experimental design. **k** Flow plot showing AF647-OVA in injected cDC1s migrated to tDLNs, 20 h after different treatments. **l** MFI of AF647-OVA in transferred CD103+ cDC1s in tDLNs ($n = 5$, biologically independent animals). **m** Flow plots showing CD86 expression on injected cDC1s in tDLN. **n** CD86 expression level on injected cDC1s in tDLNs ($n = 5$, biologically independent animals). In **j–n**, AF647-OVA was intratumorally administered 15 min before the injection of ACT-DC or control formulations. For **c, g, i, l, n**, data were presented as mean values ± SEM. Statistical analysis for (**i**): two-tailed unpaired Student's $t$-test. Statistical analysis for (**l, n**): one-way ANOVA followed by Dunnett test. For **e, j**, created in BioRender. Zhao, Z. (2025) https://BioRender.com/3ow34vf. Source data are provided as a Source Data file.

We further assessed the ability of AC-NPs to enhance antigen delivery to the adoptively transferred cDC1s and their activation status in the tDLNs (Fig. 2j–n). In this study, we injected AF647-labeled OVA as a model tumor antigen to fluorescently track its delivery to cDC1s and lymph nodes (Fig. 2j). Notably, we acknowledge that AF647-OVA is not an endogenous tumor antigen, although its use as a model antigen could provide a feasible method for monitoring antigen delivery. ACT-DC led to a 2.1–3.3-fold higher uptake of OVA by the adoptively transferred cDC1s compared to cDC1 alone or cDC1s combined with control NPs (Fig. 2k, l). Moreover, ACT-DC resulted in a 2.3- to 4.7-fold increase in the expression level of CD86 on the injected cDC1s, compared to cDC1s alone or with control NPs (Fig. 2m, n). Furthermore, data from a separate study showed that ACT-DC led to a 2.7- to 3.8-fold higher number of OVA and NP double-positive injected cDC1s in the tDLNs, compared to cDC1s with control NPs (Supplementary Fig. 19). Additionally, CLSM imaging confirmed the co-localization of OVA and AC-NPs within the injected cDC1s in the tDLNs (Supplementary Fig. 20). Collectively, our data demonstrate that ACT-DC captures AC-NPs within the tumor, efficiently traffics to tDLNs, and results in the presence of activated, antigen-carrying cDC1s in the tDLNs.

## ACT-DC eradicates small tumors and inhibits established tumors in the MC38 model

We first assessed the therapeutic efficacy of ACT-DC in an early-stage subcutaneous MC38 tumor model (Fig. 3a). Two doses of ACT-DC resulted in complete regression of primary tumors (Fig. 3b and Supplementary Fig. 21), leading to survival to day 80 without any detectable tumor recurrence (Fig. 3c). Treatment with CD103+ cDC1 alone did not induce tumor size reduction or prolonged survival. Administration of AC-NP alone led to partial tumor regression in only 50% of the mice.

Next, we evaluated the therapeutic efficacy of ACT-DC in controlling larger tumors in the subcutaneous MC38 tumor model, with an average tumor volume of ~100 mm³ at the initiation of treatment (Fig. 3d). Compared to cDC1+NP^Neg and cDC1+NP^PEG, ACT-DC showed significantly better efficacy in inhibiting tumor growth and extending animal survival (Fig. 3e, f and Supplementary Fig. 22). On day 25 post-primary tumor inoculation, ACT-DC exhibited a 6.9-fold and 9.1-fold greater efficacy in reducing tumor size compared to cDC1+NP^Neg and cDC1+NP^PEG, respectively. We also compared the therapeutic efficacy of ACT-DC to two clinically approved or investigated immunotherapies—immune checkpoint blockade (anti-PD1 antibody, aPD1) and ex vivo pulsed/activated DC vaccine. ACT-DC displayed better efficacy in inhibiting tumor growth and extending survival than the conventional DC vaccine pulsed with tumor lysate and activated with PIC ex vivo (Supplementary Fig. 23). ACT-DC also demonstrated better therapeutic efficacy than aPD1 (Fig. 3e, f). Moreover, the combination of ACT-DC and aPD1 further enhanced ACT-DC's efficacy (Fig. 3e, f). Notably, 75% of mice treated with

ACT-DC plus aPD1 achieved complete tumor eradication and survived without detectable tumor recurrence on day 80 post-primary tumor inoculation, while the complete remission rate in the ACT-DC group is 33%. This synergy between ACT-DC and aPD1 likely stems from their complementary mechanisms of action—ACT-DC enhances the induction and infiltration of cytotoxic CD8 T cells, while the aPD1 improves the activity of infiltrated CD8 T cells through immune checkpoint blockade. Notably, the similar average tumor volume curves observed between the ACT-DC and ACT-DC+aPD1 groups during the first 32 days (Fig. 3e) were attributed to two non-responder mice in the ACT-DC+aPD1 group that developed large tumors (Supplementary Fig. 22). Additionally, we used more mice (15) in the ACT-DC group than the other treatment groups (7–8) in this study to ensure an adequate number of surviving mice for subsequent rechallenge studies.

Next, we rechallenged the surviving tumor-free mice subcutaneously on day 86 to evaluate the immune memory generated by ACT-DC (Fig. 3d). ACT-DC, either alone or in combination with aPD1, demonstrated significantly superior efficacy in inhibiting the growth of the rechallenged tumors compared to age-matched naïve mice (Fig. 3g, h). Notably, 100% of the mice treated with ACT-DC plus aPD1, which had survived the primary tumors, completely rejected the rechallenged tumors, and remained tumor-free for 165 days. We next conducted a second rechallenge subcutaneously in mice that had survived the first rechallenge to further evaluate the long-term immune memory induced by ACT-DC. All mice treated with ACT-DC or ACT-DC plus aPD1, which had previously thwarted the first rechallenge, exhibited complete rejection of the second tumor rechallenge (Fig. 3i, j). We also measured the immune cell profiles in the blood 15 days after the second rechallenge. Notably, we didn't include the ACT-DC group in this study because only two mice remained survival which is not sufficient for statistical analysis. In comparison to age-matched naïve mice, mice treated with ACT-DC plus aPD1 showed elevated levels of antigen-specific CD8 T cells (Fig. 3k, l), memory CD8 T cells (Fig. 3m, n), antigen-specific memory CD8 T cells (Supplementary Fig. 24a, b), memory CD4 T cells (Supplementary Fig. 24c, d), along with reduced number of immunosuppressive cells, including regulatory T cells (Tregs) and myeloid-derived suppressor cells (MDSCs) (Supplementary Fig. 24g–j). ACT-DC plus aPD1 also increased the CD8/CD4 T cell ratio in the blood (Supplementary Fig. 24e, f). These data provide additional mechanistic support for the enduring immune memory effect induced by ACT-DC. In a separate study, we compared the therapeutic efficacy of ACT-DC to BMDC combined with AC-NPs. ACT-DC, which includes CD103+ cDC1s and AC-NPs, demonstrated significantly greater efficacy in eradicating MC38 tumors (Supplementary Fig. 25a–c). This enhanced performance is likely due to the superior migratory capacity of CD103+ cDC1s toward tDLNs and their stronger antigen-presenting capabilities.

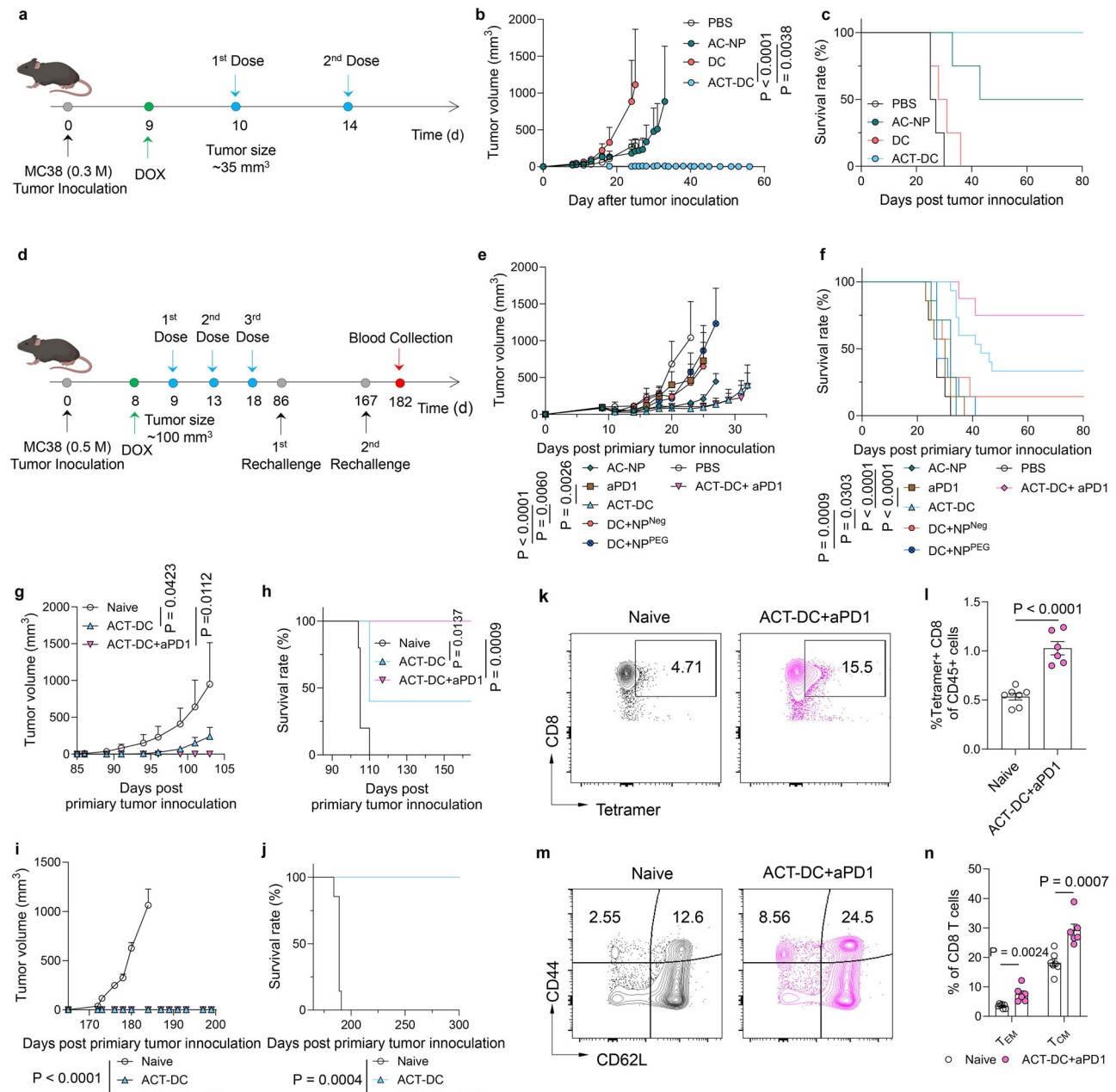

**Fig. 3 | ACT-DC eradicates small tumors and suppresses larger tumors in combination with immune checkpoint blockade in an MC38 model.**
**a**–**c** Therapeutic efficacy of ACT-DC in early-stage MC38 tumors ($n = 4$, biologically independent animals). **a** Treatment schedule. **b** Tumor growth curve. **c** Survival curve of mice treated with ACT-DC or control formulations. **d**–**n** Therapeutic efficacy of ACT-DC in an established MC38 tumor model. **d** Treatment schedule. **e** Growth curve of primary tumors ($n = 15$ for ACT-DC, $n = 8$ for ACT-DC+aPD1, $n = 7$ for the other groups, biologically independent animals). **f** Survival curve. **g**, **h** Efficacy of ACT-DC in controlling the first subcutaneous (s.c.) tumor rechallenge ($n = 5$ for ACT-DC and Naïve groups, $n = 6$ for ACT-DC+aPD1, biologically independent animals). Rechallenge was conducted on day 86 post-primary tumor inoculation. **g** Tumor growth curve. **h** Survival curve post the first rechallenge. **i**, **j** Efficacy of ACT-DC in controlling the second s.c. tumor rechallenge ($n = 7$ for Naïve, $n = 6$ for ACT-DC+aPD1, $n = 2$ for ACT-DC, biologically independent animals).

The second rechallenge was conducted on day 167 post-primary tumor inoculation. **i** Growth curve of the second rechallenged tumors. **j** Survival curve of mice after the second rechallenge. **k**, **l** Adpgk-specific CD8 T cells in the blood 15 days after the second rechallenge ($n = 7$ for Naïve, $n = 6$ for ACT-DC+aPD1, biologically independent animals). **k** Representative flow cytometry plot. **l** Percentage of Adpgk tetramer-positive cells in the blood. **m**, **n** Memory CD8 T cells in the blood 15 days after the second rechallenge ($n = 7$ for Naïve, $n = 6$ for ACT-DC+aPD1, biologically independent animals). **m** Representative flow cytometry plot. **n** Percentage of central memory (CD44 + CD62L+) and effector memory (CD44 + CD62L−) CD8 T cells in the blood. For **b**, **e**, **g**, **i**, **l**, **n**, data were presented as mean values ± SEM. Statistical analysis for (**b**, **e**, **g**, **i**): two-way ANOVA followed by Dunnett test. Statistical analysis for (**f**, **h**, **j**): two-sided Mantel–Cox tests. Statistical analysis for (**l**, **n**): two-tailed unpaired Student's $t$-test. For **a**, **d**, created in BioRender. Zhao, Z. (2025) https://BioRender.com/3ow34vf. Source data are provided as a Source Data file.

## ACT-DC induces a systemic immune response and reshapes the tumor microenvironment

ACT-DC is designed to transport captured tumor antigens in situ to tDLNs and, in turn, initiate a cascading systemic antitumor immune response. To evaluate this mechanism, we measured the immune cell profiles in the tDLNs of mice treated with ACT-DC (Fig. 4a). As shown in Fig. 4b–d, compared to PBS, ACT-DC resulted in a significant increase in the number of innate immune cells in tDLNs, including

macrophages, cDC1s, and cDC2s. A 5.7-fold higher number of cDC1s was detected in the tDLNs of mice treated with ACT-DC than in those treated with PBS. In addition, the combination of ACT-DC and aPD1 further elevated the number of all these tested innate immune cells in the tDLNs. Additionally, compared to PBS treatment, ACT-DC, either alone or in combination with aPD1, increased the expression of DC activation marker CD86 on cDC1s in the tDLN by 1.8- to 1.9-fold (Supplementary Fig. 26a, b). ACT-DC also led to an increase in CD86 expression on cDC2s compared to PBS, although this difference is not statistically significant (Supplementary Fig. 26c). Furthermore, ACT-DC also resulted in an enhanced adaptive immune response, as evidenced by the increased number of T cells (Fig. 4e), antigen-specific CD8 T cells (Fig. 4f), IFN-γ + CD8 T cells and Th1 cells (Fig. 4g), Granzyme B+ and perforin+ CD8 T cells (Supplementary Fig. 27a, b), TCF-1+ CD8 T cells (Supplementary Fig. 27c), and proliferating Ki67+ CD8 T cells (Supplementary Fig. 27d). A 3.4-4.1-fold higher number of antigen-specific CD8 T cells against two different epitopes (Adpgk and Rpl18) were observed in the ACT-DC group compared to the PBS group (Fig. 4f and Supplementary Fig. 27e). Notably, AC-NP alone, cDC1 alone, and cDC1 plus free PIC also led to increased numbers of innate and adaptive immune cells in the tDLNs, however, their efficacy was not as potent as ACT-DC (Fig. 4b–g and Supplementary Fig. 28), again highlighting the importance of both the AC-NP and cDC1 components in the ACT-DC approach. Moreover, we also evaluated the capability of ACT-DC to induce immune memory, which is crucial for preventing tumor relapse. Evidently, ACT-DC, either alone or in combination with aPD1, significantly elevated the number of both central memory and effector memory CD8 T cells in the tDLNs compared to PBS or other control formulations (Fig. 4h, i and Supplementary Fig. 29b). ACT-DC also resulted in more central memory CD8 T cells in the spleen, while its combination with aPD1 further increased the frequency of both central memory and effector memory CD8 T cells in the spleen (Fig. 4j and Supplementary Fig. 29a). Overall, these tDLN and spleen immune cell profiling data indicate that ACT-DC induced a potent systemic anti-tumor immune response with robust memory.

Next, we analyzed the immune cell profiles within tumors to evaluate ACT-DC's capability to modulate the tumor immune microenvironment (Fig. 5a). ACT-DC significantly increased the infiltration of CD4 and CD8 T cells, including Th1 cells, effector CD8 T cells (IFN-γ + CD8 T cells, Granzyme B+ CD8 T cells, and perforin+ CD8 T cells), TCF-1+ CD8 T cells, proliferating Ki67+ CD8 T cells, and antigen-specific CD8 T cells against two different epitopes (Adpgk and Rpl18), compared to the PBS treatment (Fig. 5b–e and Supplementary Fig. 30). Specifically, in comparison to PBS and other control formulations, ACT-DC led to a 1.8-4.2-fold increase in effector CD8 T cells and a 5.3-12.4-fold increase in Adpgk tetramer-positive CD8 T cells (Fig. 5c, e and Supplementary Fig. 31a–c). ACT-DC also changed the profiles of DCs within the tumor microenvironment. ACT-DC significantly increased the number of cDC1s while decreasing the number of cDC2s in the tumor compared to PBS (Fig. 5f–i and Supplementary Fig. 31d–f). cDC2s were the dominant DC subtype within the tumors of PBS-treated mice, however, ACT-DC treatment shifted this predominance towards cDC1s. ACT-DC's effect on the cDC1/cDC2 ratio may not be merely due to injecting cDC1s directly into the tumor, as the treatments with cDC1s alone or with cDC1s plus free PIC did not significantly alter this ratio. Moreover, ACT-DC also significantly changed the frequency of immunosuppressive cells, including Tregs and macrophages, within the tumor. ACT-DC led to a 36.5-72.7% reduction in Tregs and a 58.5-63.1% reduction in macrophages compared to PBS and other control formulations (Fig. 5j–l and Supplementary Fig. 31g, h). Furthermore, data from a separate study indicates that ACT did not significantly increase the expression of T cell exhaustion markers (LAG-3, TIM-3, and PD1) on intratumoral CD4 T cells (Supplementary Fig. 32a–c). However, ACT-DC led to a significant increase in TIM-3 expression on intratumoral CD8 T cells,

without affecting LAG-3 or PD1 levels (Supplementary Fig. 32d–f). Combining ACT-DC with TIM-3 blockade could be a promising strategy to further enhance the therapeutic efficacy of ACT-DC. Overall, our data suggest that ACT-DC reshapes the innate and adaptive immune cell profiles within the tumor microenvironment, transforming it into a more "immune-hot" state and facilitating tumor elimination.

## ACT-DC eliminates primary tumors and rejects tumor rechallenge in other tumor models

To evaluate the broad applicability of ACT-DC, we examined its therapeutic efficacy in two additional tumor models with lower immunogenicity, specifically the B16F10 melanoma model and the CT-2A glioma model. In the B16F10 model, mice were initially inoculated with a primary tumor, and surviving tumor-free mice underwent two subsequent challenges to assess the long-term immune memory effect (Fig. 6a). ACT-DC demonstrated significantly better efficacy in tumor eradication compared to AC-NPs, cDC1, and the combination of cDC1 with free PIC (Fig. 6b, c and Supplementary Figs. 33, 34). Notably, 80% of mice treated with ACT-DC survived without detectable tumors on day 78 post-primary tumor inoculation. The combination of ACT-DC and aPD1 further enhanced ACT-DC's efficacy, resulting in 100% tumor-free survival on day 78 post-primary tumor inoculation. Next, we subcutaneously rechallenged the surviving tumor-free mice on day 79 (Fig. 6a). ACT-DC and ACT-DC plus aPD1 led to tumor rejection in 75% and 80% of the mice, respectively (Fig. 6d, e and Supplementary Fig. 35). The blood immune cell profiles 14 days after the first rechallenge revealed that ACT-DC, either alone or in combination with aPD1, significantly increased effector memory CD4 T cells (2.5–3.4-fold enhancement) and effector memory CD8 T cells (1.6–2.1-fold enhancement) compared to naïve mice (Fig. 6f). Moreover, ACT-DC elevated the CD8/CD4 T cell ratio and CD8/Treg ratio while reducing the number of MDSCs in the blood (Supplementary Fig. 36). Additionally, we intravenously rechallenged the mice that survived the first rechallenge to assess ACT-DC's ability to induce long-term immune memory. Despite all age-matched naïve mice developing lung metastasis, mice treated with ACT-DC or ACT-DC plus aPD1 continued to survive without detectable tumors for over 260 days (Fig. 6g–i). Blood immune cell profiling further revealed an increased presence of effector and central memory CD4 and CD8 T cells (Fig. 6j, k) and increased ratios of CD8/CD4 T cells and CD8/Tregs (Supplementary Fig. 37) in ACT-DC and ACT-DC plus aPD1-treated mice, supporting ACT-DC's efficacy in rejecting rechallenged tumors. In a separate study, we directly compared ACT-DC with a conventional ex vivo DC vaccine loaded with two defined antigens (gp100$_{25-33}$ and TRP-2$_{180-188}$). ACT-DC demonstrated significantly better therapeutic efficacy (Supplementary Fig. 38), further indicating the advantages of the in situ immunization approach enabled by ACT-DC.

To further assess the systemic and multivalent immune response triggered by ACT-DC, we utilized a bilateral tumor model. In this model, B16F10-OVA cells and B16F10 cells were inoculated into the right and left flanks, respectively, with only the B16F10-OVA tumor receiving ACT-DC treatment (Fig. 6l). Remarkably, ACT-DC, both alone and in combination with aPD1, significantly inhibited not only the treated primary B16F10-OVA tumors but also the untreated distant B16F10 tumors, compared to PBS or aPD1 treatments (Fig. 6m–o). On day 21, the distant tumors became undetectable in 42.8% of mice in the ACT-DC group and in 71.4% of mice treated with ACT-DC plus aPD1. This inhibition of distant B16F10 tumors suggests that ACT-DC induced a robust systemic immune response targeting B16F10 tumor antigens. Additionally, we observed significantly higher numbers of both OVA-specific and TRP-2-specific CD8 + T cells (TRP-2 being a B16F10 neoantigen) in the draining lymph nodes of the primary B16F10-OVA tumor (Fig. 6p, q), supporting the induction of a multivalent antigen-specific immune response.

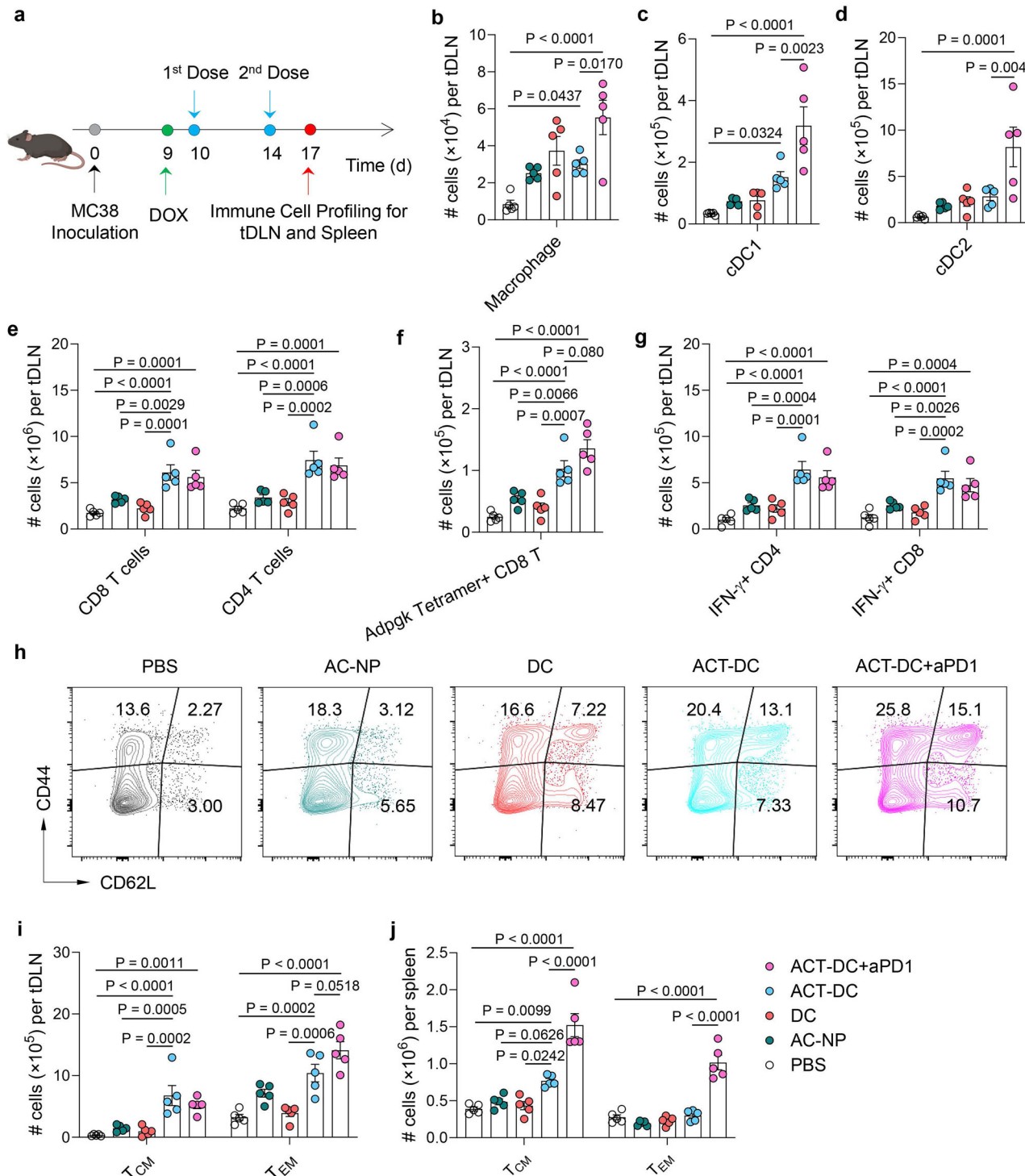

**Fig. 4 | ACT-DC induces a systemic immune response with memory. a** Schedule of the study to profile immune cells in tDLNs and spleen. Created in BioRender. Zhao, Z. (2025) https://BioRender.com/3ow34vf. **b–g** Number of different immune cells in the tDLN of mice treated with ACT-DC or control therapies ($n = 5$ biologically independent animals per group) including macrophages (**b**), cDC1 (**c**), cDC2 (**d**), CD8 and CD4 T cells (**e**), Adpgk tetramer-positive CD8 T cells (**f**), and IFN-γ expressing CD4 and CD8 T cells (**g**). **h–j** Number of memory CD8 T cells in the tDLN

and spleen of mice receiving the ACT-DC or control therapies ($n = 5$ biologically independent animals per group). **h** Representative flow cytometry plots showing the memory CD8 T cells in the tDLN. **i** Total number of effector memory (CD44 + CD62L−) and central memory (CD44 + CD62L+) CD8 T cells in the tDLN. **j** Total number of memory CD8 T cells in the spleen. For **b–g**, **i**, **j**, data were presented as mean values ± SEM. Statistical analysis for (**b–g**, **i**, **j**): one-way ANOVA followed by Dunnett test. Source data are provided as a Source Data file.

To investigate the roles of CD4 and CD8 T cells, as well as their egress from lymph nodes, on ACT-DC's therapeutic efficacy, we treated mice with ACT-DC followed by the administration of anti-CD4 antibody, anti-CD8 antibody, or the T cell egress inhibitor FTY720 (an

S1PR inhibitor) (Fig. 6r). Depletion of either CD4 or CD8 T cells significantly reduced the efficacy of ACT-DC (Fig. 6s), indicating the essential role of both T cell subsets in the ACT-DC approach. Notably, CD8 T cell depletion resulted in a more dramatic reduction in

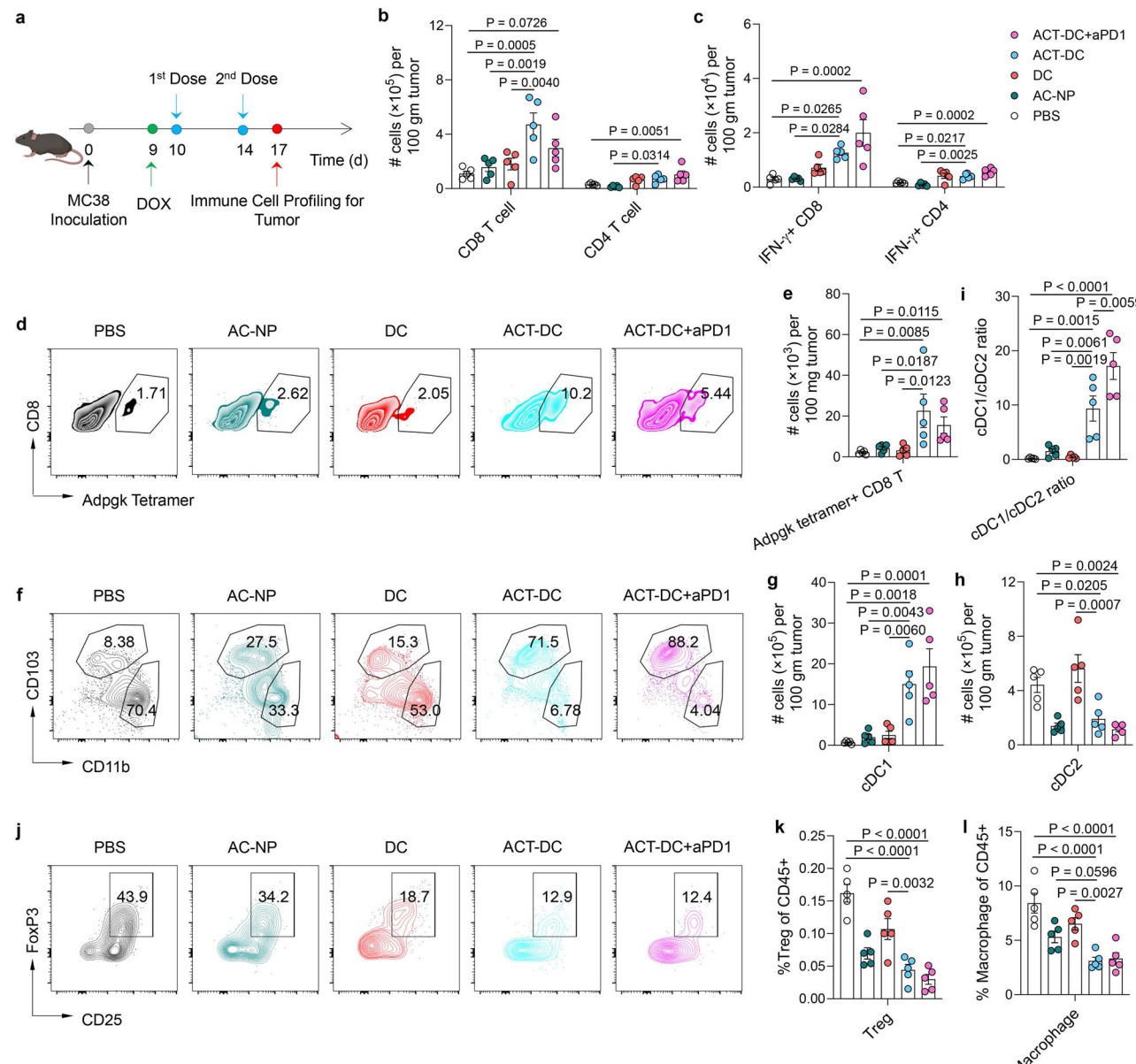

**Fig. 5 | ACT-DC transforms the tumor microenvironment into a more "immune-hot" state. a** Schedule of the study to profile immune cells in the tumor. Created in BioRender. Zhao, Z. (2025) https://BioRender.com/3ow34vf. **b, c** Number of CD8/CD4 T cells (**b**) and IFN-γ expressing CD8/CD4 T cells (**c**) in the tumor. **d, e** Representative flow cytometry plots (**d**) and the number of Adpgk tetramer-positive CD8 T cells (**e**) in the tumor. **f–i** Profiles of cDC1s and cDC2s in the tumor. **f** Representative flow cytometry plots showing the presence of cDC1s and cDC2s in the tumor. **g, h** The number of cDC1s (**g**) and cDC2s (**h**) in the tumor. **i** cDC1/cDC2

ratio in the tumor. **j–l** Profiles of Tregs and macrophages in the tumor. **j** Representative flow cytometry plots showing the presence of Tregs in the tumor. **k, l** Percentage of Tregs (**k**) and macrophages (**l**) in the tumor following different treatments. For **b, c, e, g–i, k, l**, data were presented as mean values ± SEM. For **a–l**, *n* = 5 biologically independent mice per group. Statistical analysis for (**b, c, e, g–i, k, l**): one-way ANOVA followed by Dunnett test. Source data are provided as a Source Data file.

therapeutic efficacy compared to CD4 T cell depletion, suggesting a potentially more critical function of CD8 T cells in ACT-DC's effectiveness. FTY720 treatment also significantly impaired ACT-DC's efficacy, indicating that T cell trafficking and egress from lymph nodes are essential for ACT-DC's therapeutic effectiveness. To evaluate the contribution of endogenous cDC1s to ACT-DC's efficacy, we conducted therapeutic studies in *Batf3*^−/−^ mice, which lack endogenous cDC1s (Supplementary Fig. 39a and Fig. 6t). While ACT-DC treatment in *Batf3*^−/−^ mice significantly delayed tumor growth (Supplementary Fig. 39b) and improved survival rates (Fig. 6t), its therapeutic efficacy was markedly reduced compared to that in the wild-type mice. These data indicate that in addition to the adoptively transferred cDC1s,

endogenous cDC1s are also critical to the success of the ACT-DC approach.

Notably, in both the B16F10 and MC38 models, tumors were pretreated with intratumoral doxorubicin before ACT-DC therapy to promote tumor antigen release. We conducted a comparative study to assess the impact of doxorubicin pretreatment and its administration route on ACT-DC's therapeutic efficacy (Supplementary Fig. 40a–d). Even without doxorubicin pretreatment, ACT-DC significantly delayed tumor growth and achieved tumor-free survival in 42.9% of treated mice, although its efficacy was less potent than in doxorubicin-pretreated mice. Moreover, the route of doxorubicin administration did not significantly influence ACT-DC's efficacy. Pretreatment with

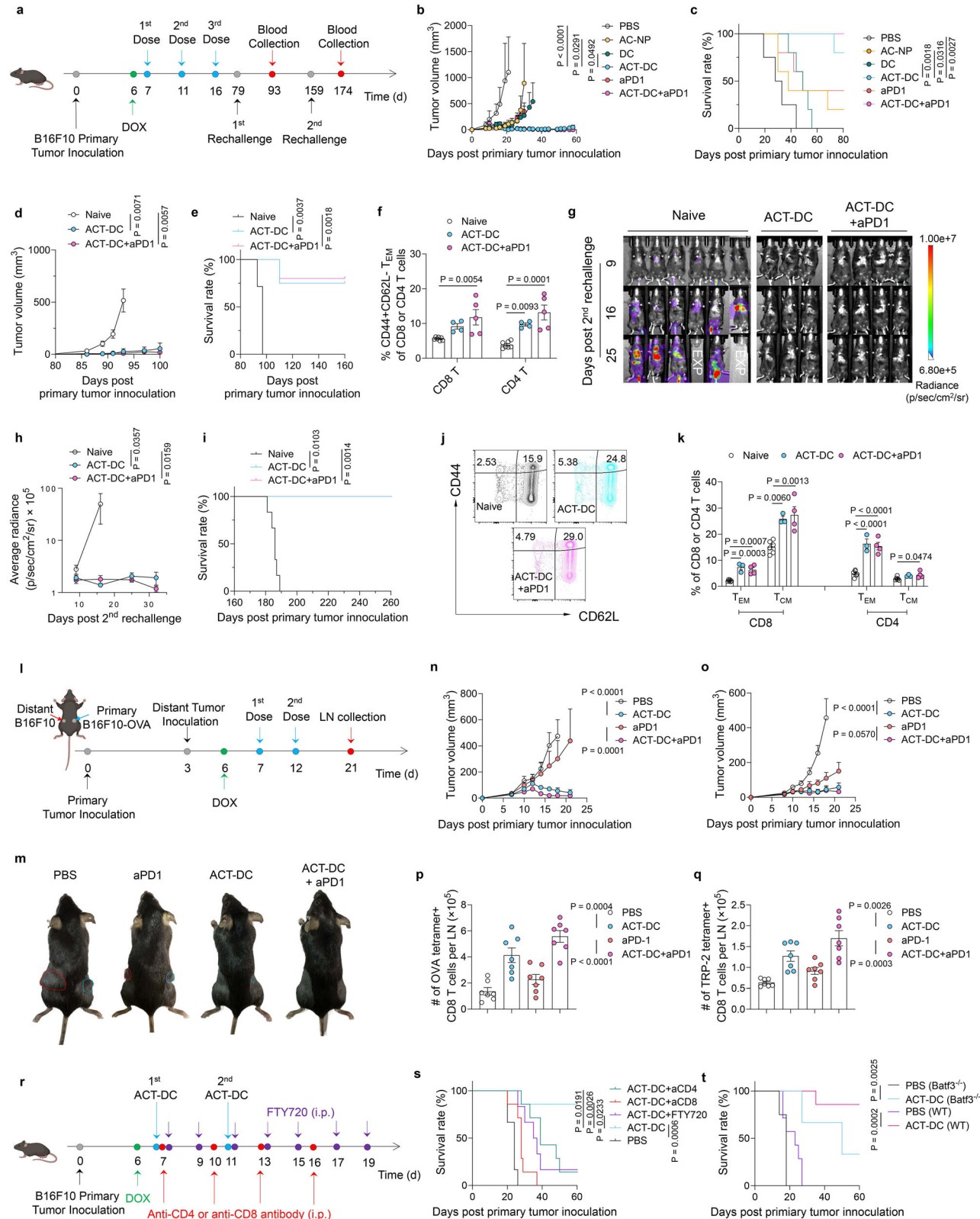

either intravenous or intratumoral doxorubicin before ACT-DC therapy resulted in comparable tumor growth inhibition and an 85.7% tumor-free survival rate.

We then evaluated the therapeutic efficacy of ACT-DC in an orthotopic CT-2A glioma model (Fig. 7a). Glioma is one of the deadliest tumors, known for its immunologically cold microenvironment, characterized by limited lymphocyte infiltration, resistance to both conventional and immune-based therapies, and high recurrence rate[58,59]. Radiation therapy (RT) is a standard-of-care treatment for glioma and is known to trigger the release of tumor antigens[60]. Our data indicates that RT alone or its combination with aPD1 did not improve animal survival (Fig. 7b), confirming the therapy resistance feature of this model. Encouragingly, although ACT-DC did not greatly improve the RT effects as compared to RT alone (Supplementary

**Fig. 6 | ACT-DC eradicates primary tumors, rejects rechallenged tumors, and induces a potent abscopal effect in a B16F10 model. a** Treatment schedule. **b** Growth curve of primary tumors. **c** Survival curve before the first rechallenge. **d, e** Efficacy of ACT-DC in controlling the first s.c. tumor rechallenge. Tumor growth curve (**d**) and survival curve (**e**) after the first rechallenge were shown. **f** Effector memory CD4 and CD8 T cells in blood 14 days post the first rechallenge. **g–i**, Efficacy of ACT-DC in controlling the second intravenous (i.v.) rechallenge. **g** LagoX images showing the progression of i.v. rechallenged tumors. "EXP" indicates mice reached humane endpoints. **h** Tumor burden in the lungs. **i** Survival curve after the second rechallenge. **j, k** Memory T cells in blood 15 days after the second rechallenge. **j** Representative flow plots of memory CD8 T cells. **k** Percentage of memory T cells in blood. **l–q** Systemic abscopal effect induced by ACT-DC. **l** Schematic of the study design. **m** Representative images of mice on day 21 after primary tumor inoculation. Blue circles: primary tumors; red circles: distant tumors. **n** Growth curve of primary tumors. **o** Growth curve of distant tumors. **p–q** OVA-specific (**p**) and TRP-2-specific (**q**) CD8 T cells in the tDLN of the primary tumor on day 21. **r, s** Impact of T cell depletion and egress inhibition on the therapeutic efficacy of ACT-DC. **r** Schematic of the study

design, **s** Mouse survival curve. **t** Impact of endogenous cDC1s on the therapeutic efficacy of ACT-DC. Survival curves of wild-type or Batf3$^{-/-}$ mice after ACT-DC treatment are shown. For **b, d, f, h, k, n–q, s, t**, data were presented as mean ± SEM. For **b, c**, $n = 4$ for PBS, $n = 5$ for the other groups, biologically independent mice. For **d–f**, $n = 7$ for Naïve, $n = 4$ for ACT-DC, $n = 5$ for ACT-DC+aPD1, biologically independent mice. For **h–k**, $n = 6$ for Naïve, $n = 3$ for ACT-DC, $n = 4$ for ACT-DC+aPD1, biologically independent mice. For **n, o**, $n = 9$ for PBS, $n = 7$ for the other groups, biologically independent mice. For **p, q**, $n = 7$ biologically independent mice per group. For **r, s** $n = 6$ for PBS and ACT-DC + FTY720, $n = 7$ for the other groups, biologically independent mice. For **t**, $n = 7$ biologically independent mice per group for wild-type mice; for Batf3$^{-/-}$ mice, $n = 4$ for PBS, $n = 6$ for ACT-DC, biologically independent mice. Statistical analysis for (**b, d, n, o**): two-way ANOVA with Dunnett test. Statistical analysis for (**h**) was performed on day 16 using a two-sided Mann–Whitney test. Statistical analysis for (**c, e, i, s, t**): two-sided Mantel–Cox tests. Statistical analysis for (**f, k, p, q**): one-way ANOVA with Dunnett test. For **a, l, r**, Created in BioRender. Zhao, Z. (2025) https://BioRender.com/3ow34vf. Source data are provided as a Source Data file.

Fig. 41), a combination of RT, ACT-DC, and aPD1 led to tumor regression in 50% of the treated mice (Fig. 7b). It is worth noting that ACT-DC significantly outperformed the conventional ex vivo tumor lysate-pulsed DC vaccine in improving glioma response to RT and aPD1 therapy. We further analyzed the immune composition in the brains of long-term survivor (LTS) mice, defined as mice surviving 100 days post-tumor inoculation. Compared to mice with existing primary CT-2A tumors (the "Tumor" group), LTS mice exhibited a significantly higher ratio of tumor-infiltrating lymphocytes (TILs) to tumor-associated myeloid cells (TAMCs) in the brain (Fig. 7c–e). Additionally, LTS mice had 1.7-fold and 4.4-fold higher abundance of CD8 and CD4 T cells in their brain, respectively, compared to the "Tumor" mice (Fig. 7f–h). Further analysis of CD4 T cell populations indicates that over 95% of CD4 T cells in LTS brains were CD4 T helper cells, and there was an 86.2% reduction in Tregs as compared to "Tumor" mice (Fig. 7i–k). It is also worth noting that, compared to healthy control mice without tumor implantation (the "non-tumor" group), LTS mice, which were also tumor-free, demonstrated a significantly higher abundance of lymphocytes, both CD8 and CD4 T cells, in the brain, indicting an immune surveillance or memory induced by ACT-DC therapy. These immune cell profiling results support the robust therapeutic efficacy of ACT-DC in the CT-2A model. Notably, the activation status and phenotype of TAMCs following ACT-DC treatment need to be further investigated in future studies. Our data collectively demonstrated that ACT-DC is effective in treating solid tumors, inducing immune memory, and displaying applicability across multiple tumor models.

## Discussion

In this work, we have demonstrated an effective approach, ACT-DC, to enable in situ immunization for systemic tumor eradication. In situ cancer immunization, leveraging the native antigens present within a tumor to stimulate an immune response, is a promising approach to induce a broad T cell response against the heterogeneous tumor antigens specific to a patient[36–41]. This approach depends on activating APCs such as DCs within the tumor to capture and present antigens to T cells, thereby initiating a systemic immune response. Unlike ex vivo tumor vaccines, which target a limited number of tumor antigens, in situ immunization has the potential to target a wider array of antigens in a manner tailored to each patient. This could more effectively address the variability in tumor antigens between and within patients and reduce the possibility of antigen escape and immune evasion[36–41]. However, current approaches to improve in situ cancer immunization (e.g., intratumoral or systemic administration of immunogenic cell death inducers and adjuvants[61–63]) have only achieved modest clinical efficacy. A significant barrier is the lack of effective DC subtypes that

most effectively present antigens and their impaired antigen-presenting functions due to the immunosuppressive nature of the tumor microenvironment. ACT-DC provides an effective modular approach to counteract this barrier.

ACT-DC is an effective NP-boosted cDC1 therapy for enhancing antigen presentation and potentiating in situ immunization. ACT-DC hinges on two key integrative components, including the migratory CD103+ cDC1s and AC-NPs. Migratory CD103+ cDC1s, a unique subset of DCs, are known for their strong ability to migrate and present antigens[44,45]. However, this DC subset is present within tumors at a very low frequency (<2% of total CD45+ cells) even in some "immune-hot" tumors and is primarily distributed in the tumor peripheral[48,64], restricting their access to tumor antigens throughout the whole tumor. In the ACT-DC approach, the intratumoral dosing of migratory CD103+ cDC1s not only increased their total number within a tumor but altered their spatial distribution to cover a wider area for better access to diverse tumor antigens. Notably, ACT-DC employs a cascade antigen relay mechanism to achieve in situ immunization (Fig. 8). Specifically, upon intratumoral administration, AC-NPs capture native tumor antigens and in situ deliver them to as well as activate migratory CD103+ cDC1s. This process triggers the active trafficking of these cDC1s to tDLNs and thus relays antigen transport to tDLNs for efficient T cell priming. In tDLNs, cDC1s efficiently present captured tumor antigens to naïve CD8 T cells, inducing a polyclonal, antigen-specific T cell response that overcomes antigen heterogeneity. Additionally, a portion of intratumorally retained cDC1s, activated by AC-NPs, could reshape the local tumor microenvironment, enhancing immune cell infiltration after induction of an antitumor immune response in the tDLNs. The integration of the cDC1s and AC-NPs, as evidenced by our in vivo data, is crucial for ACT-DC's immunological efficacy, as either cDC1s or AC-NPs alone led to limited effectiveness in tumor control.

Our cellular-level biodistribution data suggested that both AC-NPs and the model tumor antigen were predominantly taken up by the adoptively injected cDC1s compared to other immune cells within the tumor. Although AC-NPs don't have an active targeting ligand specific to cDC1s, two key factors likely contributed to their dominant delivery to cDC1s. First, in the ACT-DC approach, AC-NPs and cDC1s were sequentially (15 min apart) injected intratumorally into the same or nearby location. This spatial proximity and co-localization at the injection site and surrounding diffusion areas increased the likelihood of AC-NPs being internalized by cDC1s. Second, as professional antigen-presenting cells with strong antigen-processing capabilities, cDC1s are known for their superior ability to recognize and take up particulate matter, such as damaged tumor cells and large tumor cell debris[44,65,66]. Our data indicates that upon capturing tumor proteins, AC-NPs undergo size increase. The particulate nature of AC-NPs

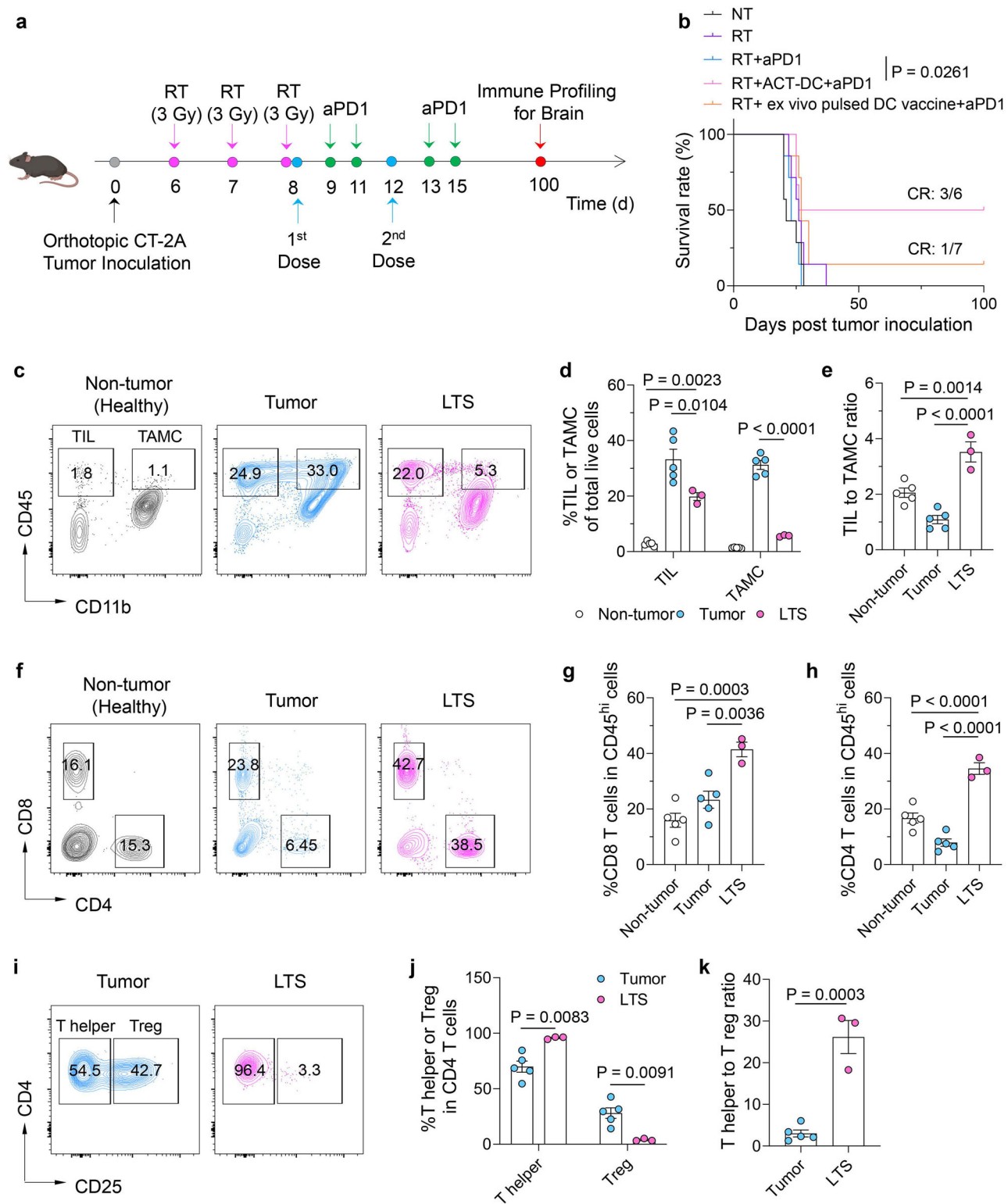

combined with the display of captured tumor antigens and DAMPs could trigger and enhance their recognition and uptake by cDC1s. These factors collectively lead to the dominant delivery of AC-NPs to adoptively injected cDC1s despite AC-NPs not having an active targeting ligand for cDC1s.

We have demonstrated the effectiveness and broad applicability of ACT-DC in eliminating primary tumors and inducing immune memory in different tumor models. The broad effectiveness of ACT-DC can be in part attributed to the robust efficiency of AC-NPs to

capture tumor proteins via universal interactions, including hydrophobic and electrostatic interactions, irrespective of tumor type. ACT-DC demonstrated better therapeutic efficacy than two types of conventional DC vaccines that were ex vivo activated and loaded with tumor lysate-derived antigens or defined antigens. Unlike ex vivo DC vaccines, ACT-DC uses the native antigens directly from a tumor and may provide a platform approach to produce DC therapies in vivo, facilitating a robust antitumor immune response in a patient-specific manner. Importantly, the ACT-DC approach does not involve any

**Fig. 7 | ACT-DC in combination with anti-PD1 antibody induces anti-tumor immune response in an orthotopic CT-2A glioma model. a** Schedule of the therapeutic study in the CT-2A model. Created in BioRender. Zhao, Z. (2025) https://BioRender.com/3ow34vf. **b** Survival curves of mice treated with different formulations. **c–e** Tumor-infiltrating lymphocyte (TIL) and tumor-associated myeloid cell (TAMC) in the brains of mice 100 days after tumor inoculation. **c** Representative flow cytometry plots of TIL and TAMC. **d** Quantification of the abundance of TIL and TAMC. **e** Relative ratio of TIL to TAMC in the brain. **f–h** CD4 and CD8 T cells in the brain of mice 100 days after tumor inoculation. **f** Representative flow cytometry plots of CD4 and CD8 T cells. **g, h** Percentage of CD8 T cells (**g**) and CD4 T cells (**h**) in CD45+ cells in the brain. **i–k** T helper CD4 T cells and Tregs in the brain of mice 100 days after tumor inoculation.

**i** Representative flow cytometry plots. **j** Percentage of T helper CD4 T cells and Tregs. **k** Relative ratio of T helper cells to Tregs. For **c–k**, "Non-tumor" refers to healthy mice without tumor implantation. "Tumor" refers to mice inoculated with CT-2A tumors for 21 days. "LTS (long-term survivor)" refers to mice that received RT + ACT-DC+aPD1 treatments and survived on day 100 after tumor inoculation. For **d**, **e**, **g,h**, **j**, **k**, data were presented as mean values ± SEM. For **b**, n = 6 for the RT + ACT-DC+aPD1 group, n = 7 for the other groups, biologically independent mice. For **c–k**, n = 3 for the LTS group, n = 5 for the "Tumor" and "Non-tumor" groups, biologically independent mice. For **b**, statistical analysis was performed using two-sided Mantel–Cox tests. Statistical analysis for (**d**, **e**, **g**, **h**): one-way ANOVA followed by Dunnett test. Statistical analysis for (**j**, **k**): two-tailed unpaired Student's t-test. Source data are provided as a Source Data file.

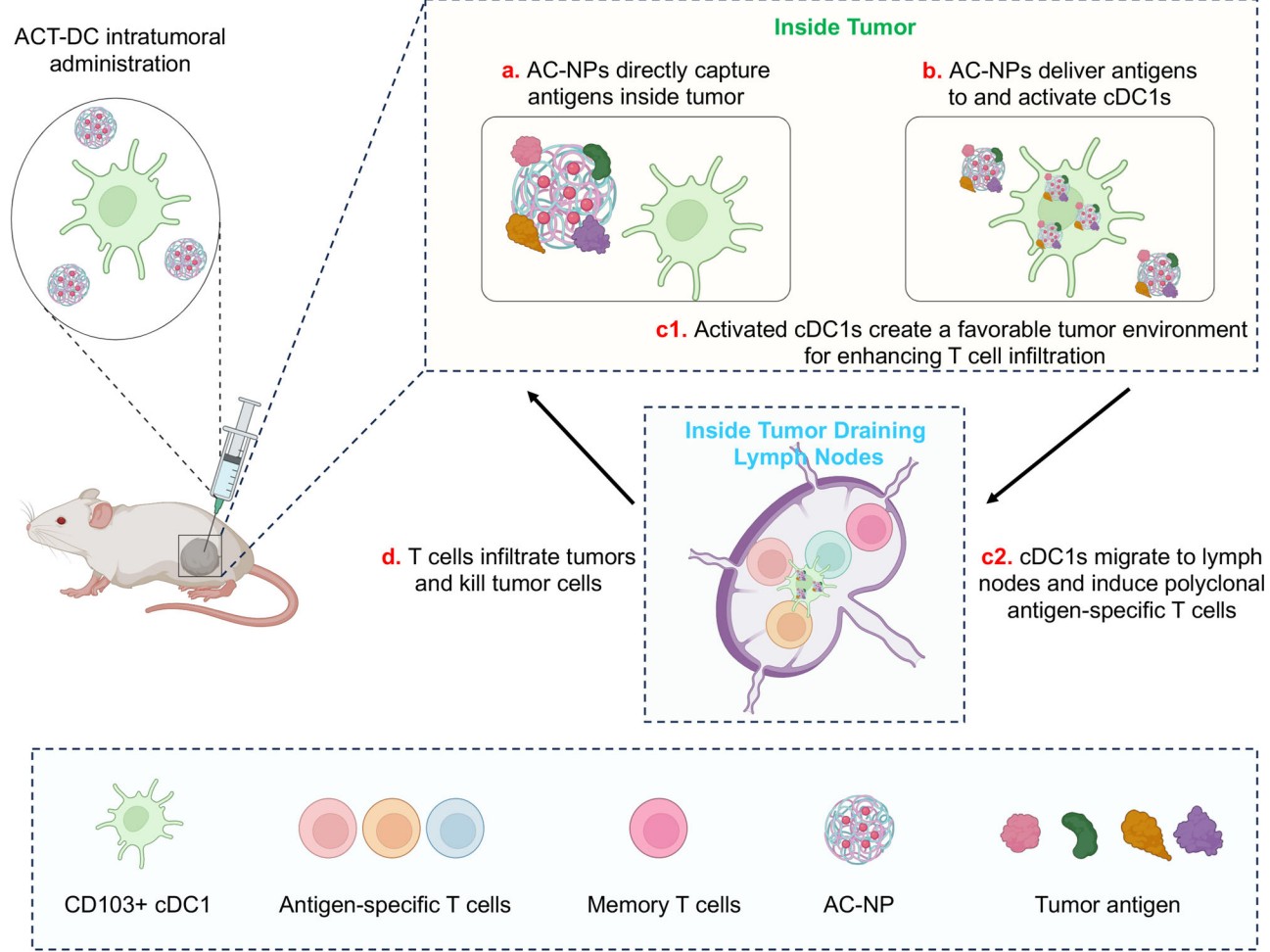

**Fig. 8 | Schematic illustration of the mechanism of ACT-DC for in situ immunization against solid tumors.** By integrating AC-NPs with the adoptive transfer of CD103+ cDC1s, ACT-DC harnesses native tumor antigens to induce potent and long-lasting systemic immune responses against solid tumors. Upon intratumoral administration, AC-NPs capture native tumor antigens in situ (**a**) and enhance their delivery to migratory CD103+ cDC1s while simultaneously activating them (**b**). The activated cDC1s then migrate to tDLNs, where they enhance antigen presentation, leading to the induction of potent polyclonal antigen-specific T cells and memory T cells (**c2**). Meanwhile, activated cDC1s retained within the tumor modulate the tumor microenvironment to reduce immunosuppression and enhance T cell infiltration (**c1**). The T cells generated in tDLNs effectively infiltrate tumors, leading to the eradication of solid tumors (**d**). Created in BioRender. Zhao, Z. (2025) https://BioRender.com/qgi6dnp.

ex vivo biochemical stimulation or genetic engineering of the cultured cDC1s, blunting the manufacturing challenges associated with conventional cell therapies. ACT-DC also outperformed the immune checkpoint inhibitor anti-PD1 antibody and resulted in a synergistic effect in their combination, leading to a 50-100% complete response rate in the MC38, B16F10, and CT-2A models. Through further optimization, ACT-DC can be a new cancer immunotherapy with a high response rate or an adjuvant approach to improving current cancer

immunotherapies, for which a low patient response rate is one major challenge.

We demonstrated the feasibility of ACT-DC to function post chemotherapy and radiotherapy, which facilitates tumor antigen release. ACT-DC's applicability in other contexts, such as post-surgery, needs to be further investigated in future studies. The modularity and complexity of the ACT-DC approach also needs to be further considered for clinical translation. With its modular design, the AC-NP

component of ACT-DC can be further modified to incorporate other rationally selected immunomodulatory agents, allowing adaptation to tumors with varying immune microenvironments. As AC-NPs are designed to capture native tumor antigens in situ, their intratumoral administration can maximize direct access to tumor antigens within the tumor microenvironment. While many clinically approved NPs (e.g., Doxil and Abrexane) are often administered intravenously, some NPs (e.g., NBTXR3, a hafnium oxide nanoparticle) have also been approved for intratumoral injection in cancer therapy[67]. Notably, autologous cDC1-based ex vivo antigen-pulsed dendritic cell vaccines are currently under evaluation in clinical trials (e.g., NCT05773859), highlighting the clinical translatability of using cDC1s in human studies. Since cDC1s account for only 0.2–0.3% of peripheral blood mononuclear cells, isolating sufficient quantities of native autologous cDC1s from a patient's blood may present challenges. However, a more feasible approach could involve generating cDC1s using human induced pluripotent stem cell (iPSC)-based methods[68,69]. Given that iPSCs can theoretically produce an unlimited number of autologous cells, this method could generate sufficient cDC1s for repeated dosing in human treatments.

ACT-DC is designed for intratumoral injection, a drug administration method known to enhance local bioavailability and therapeutic efficacy[70,71]. Intratumoral injection is clinically applicable for both surface-accessible tumors (e.g., melanoma) and internal tumors (e.g., breast, colon, and brain tumors) with the aid of imaging techniques[72]. Intratumoral administration has been investigated in over 200 clinical trials for the delivery of adjuvants, viral therapies, chemotherapies, cytokines, immune checkpoint inhibitors, nanoparticle therapeutics, and cell therapies, including more than ten clinical trials focused on DC therapies[70]. While intratumoral injection is a clinically translatable approach, the dosing regimen of ACT-DC needs to be further optimized to reduce the dosing frequency, enhance efficacy, and minimize potential adverse effects. Overall, the ACT-DC strategy presents a promising, broadly effective approach for in situ cancer immunization and tumor microenvironment modulation.

## Methods

### Ethical statement

All the animal experiments were performed in compliance with National Institutes of Health and institutional guidelines. All animal procedures were conducted according to approved protocols by the Institutional Animal Care and Use Committee (IACUC) at the University of Illinois, Chicago (21-098, 24-085) and Northwestern University (IS00029388). In murine subcutaneous tumor models, the maximum tumor size permitted by our institutional IACUC is 2 cm in the largest diameter. Additional key humane endpoints include body weight loss exceeding 20%, body condition score below 2, and significant tumor ulceration (>3 mm in diameter). We confirm that all our animal studies complied with these guidelines, and the tumor size did not exceed the 2 cm limit in any dimension.

### Materials

Poly(lactic-co-glycolic acid) (PLGA, 50:50, Resomer RG503H), linear polyethylenimine (PEI, MW 2500), polyinosinic:polycytidylic acid (PIC), non-essential amino acid solution, and polyvinyl alcohol (PVA) were purchased from Sigma-Aldrich (St. Louis, MO). mPEG-PLGA (MW: 5k/20k) was purchased from Nanosoft Polymers (Winston-Salem, NC). RPMI-1640 medium, high glucose-Dulbecco's modified Eagle's medium (DMEM), and penicillin/streptomycin were obtained from Cytiva (Marlborough, MA). Heat-inactivated fetal bovine serum (FBS) was purchased from Corning (Corning, NY). Recombinant murine FTL3L, GoInVivo™ anti-PD1 antibody, and recombinant murine GM-CSF were obtained from BioLegend (San Diego, CA). 2-mercaptoethanol, HEPES buffer, ACK buffer, sodium pyruvate, 1,1′-dioctadecyl-3,3,3′,3′-tetramethylindodicarbocyanine (DiD), 3,3′-

dioctadecyloxacarbocyanine perchlorate (DiO), 1,1′-dioctadecyl-3,3,3′,3′-tetramethylindocarbocyanine perchlorate (DiI), Hoechst 33342, and blasticidin were purchased from Thermo Fisher Scientific (Waltham, MA).

### Cell lines and animals

The MC38 cell line (Catalog # ENH204-FP) was purchased from Kerafast (Boston, MA). MC38 cells were cultured in DMEM medium supplemented with 10% fetal bovine serum (FBS), 1% penicillin−streptomycin, 1% non-essential amino acids, 0.01 M HEPEs, and 50 µg/mL Gentamicin. The CT-2A cells were obtained as a gift from Dr. Tom Seyfried at Boston College. CT-2A cells were maintained in DMEM with 10% fetal bovine serum and 1% penicillin/streptomycin. The B16F10 cell line expressing luciferase (B16F10-Luc) (Catalog # CRL-6475-LUC2™) and the B16F10 cell line (Catalog # CRL-6475) were obtained from ATCC (VA, USA). The B16F10-OVA cells were obtained as a gift from Dr. Darrel Irvine at the Massachusetts Institute of Technology. B16F10-Luc cells were cultured in DMEM medium supplemented with 10% FBS, 1% penicillin−streptomycin, and 10 µg/mL blasticidin. B16F10 and B16F10-OVA cells were cultured using a similar method as B16F10-Luc, without adding blasticidin to the culture medium. Male/female C57BL/6 mice (6–8 weeks of age) and female B6.129S(C)-Batf3tm1Kmm/J mice (6–8 weeks of age) were purchased from Jackson Laboratory (ME, USA). Mice were housed in a facility with controlled conditions, including a 14:10-h light:dark cycle, an ambient temperature maintained at $22 \pm 2$ °C, and a relative humidity of 30–70%.

### Preparation and characterization of AC-NPs

AC-NPs were prepared using a double emulsion method[73,74]. Briefly, 20 mg of PLGA (Resomer RG503H) and 10 mg of PEI (linear, MW 2500) were dissolved in 1 mL of chloroform as the organic phase with the assistance of a water-bath sonicator. For the preparation of fluorescently labeled NPs, 10 µL of 5 mg/mL DiD or DiO was dissolved in the organic phase. Subsequently, 150 µL of water containing 3 mg PIC was added to the organic phase, followed by a water-bath sonication for 1 min. The emulsion was then dropwise added to 11 mL of 0.5% polyvinyl alcohol solution. Particle formation was achieved through probe-sonication for 20 s twice with a 20-s break in between, followed by stirring for over 12 h in a fume hood to completely evaporate the organic solvent. Negatively charged (NP$^{Neg}$) and PEGylated (NP$^{PEG}$) NPs were prepared using a similar method without the addition of PEI. NPs were washed three times with deionized water for characterization of their physicochemical properties. The size and surface charge of NPs were measured using dynamic light scattering (DLS) (Malvern Nano-ZS Zetasizer). Additionally, the morphology of NPs was characterized by scanning electron microscopy (SEM) (JEOL JSM-IT500HR). For quantifying the loading of PIC into NPs, NPs were lysed in a buffer containing 100 mM sodium hydroxide and 0.05% sodium dodecyl sulfate with gentle shaking overnight at 37 °C. The encapsulated PIC was quantified using Nanodrop 2000 (Thermo Fisher).

### CD103+ cDC1 culture

CD103+ cDC1s were cultured using a previously reported method with modifications[57]. Bone marrow was obtained from the femurs of freshly euthanized C57BL/6 mice. The harvested bone marrow was seeded to a 150 mm × 20 mm petri dish at a concentration of $1.5 \times 10^{6}$ cells/mL and cultured in RPMI-1640 media supplemented with 10% FBS, 1% Penicillin−Streptomycin, 50 µM 2-mercaptoethanol, 200 ng/mL FTL3L, and 2 ng/mL GM-CSF. The media was refreshed on days 3 and 6. Non-adherent cells were collected on day 9, counted, and replated. Non-adherent cells were harvested on days 12–15. Prior to use, cells were washed two to three times with PBS. The obtained CD103+ cDC1s were characterized by surface marker expression, including CD11c, B220,

MHCII, CD103, and Clec9A via antibody staining followed by flow cytometry (CytoFLEX, Beckman).

## Tumor lysate preparation

MC38, B16F10, and CT-2A cells were initially seeded and cultured until reaching 80–90% confluency. Subsequently, the cells were collected and lysed through five freeze-thaw cycles involving rapid freezing at −80 °C and thawing at 37 °C for 5 min each cycle. Following this, the tumor lysates underwent centrifugation to eliminate cell debris, and the resulting supernatant was collected. Protein concentrations in the tumor lysates were quantified using a bicinchoninic acid (BCA) assay. For FITC conjugation to tumor lysates, 1 mg/mL of tumor lysates were combined with 5 mg of fluorescein isothiocyanate (FITC) in a carbonate buffer (0.1 M $Na_2CO_3$, 0.1 M $NaHCO_3$, pH 9.5) under stirring for 24 h. Unconjugated FITC was removed through continuous dialysis for 3 days using a dialysis membrane (12,000–14,000 MW) (Spectrum Labs). The FITC-labeled tumor lysates were stored at −80 °C until use.

## Evaluation of the antigen-capturing efficiency of NPs

The antigen-capturing ability of different NPs was studied using the model antigen ovalbumin (AF647-OVA) and MC38 tumor lysate. The protein content in the tumor lysate was pre-quantified using a BCA assay. In brief, 500 µg of NPs were added to a solution containing different concentrations of AF647-OVA or tumor lysate and incubated at 37 °C for 30 min. Following incubation, the mixture was centrifuged at 12,000 × $g$, and the pellet containing NPs bound with proteins was collected. The unbound protein in the supernatant was quantified using a BCA assay (for tumor lysates) or a fluorescence-based method using a plate reader (for AF647-OVA). The amount of protein bound to the NPs was determined by subtracting the unbound protein from the total added protein. The charge and size of NPs after protein binding were measured using DLS.

The composition of proteins bound to NPs was evaluated through proteomics analysis. Briefly, 2 mg of NPs were washed three times with ultrapure water before use and then incubated with 100 µg of tumor lysates at 37 °C for 30 min. The NPs were then washed with 1 mL phosphate-buffered saline (PBS) once and resuspended in 300 µL PBS. The NPs with bound proteins (1–10 µg) were resuspended in 45 µL of 8 M urea/100 mM Tris buffer (pH 8.5). Next, 5 µL of 50 mM tris(2-carboxyethyl)phosphine (TCEP) (Sigma-Aldrich) was added and incubated at 56 °C for 1 h. Then, 5.5 µL of 500 mM 2-chloroacetamide (Alfa Aesar) was added and incubated for 30 min in the dark at room temperature. The reaction was diluted with 3 volumes of 100 mM Tris buffer (pH 8.5) to reduce the urea concentration to 2 M before trypsin (Sigma-Aldrich) was added to a final protease/protein ratio of 1:50 (w/w) and incubated overnight at 37 °C. After that, the sample solution was heated up to 56 °C for 5 min before being spun down at 18,000 × $g$ for 15 min to collect the supernatant with peptides. Formic acid was added to a final concentration of 1%, and samples were analyzed by liquid chromatography-mass spectrometry (LC/MS). A total of nine samples (three types of NPs and three biological replicates for each NP type) were prepared and analyzed. For the LC/MS run, up to 1 µg of peptides was loaded onto Evotips and cleaned up following the manufacturer's instructions (Evosep Biosystems, Denmark). Samples were injected into a Thermo Q Exactive HF quadrupole-Orbitrap MS equipped with a nanospray ESI source (Thermo Fisher Scientific) using an Evosep One instrument (Evosep Biosystems). The standard preset method for 60 samples per day (60 SPD) was used for the LC component of the run. The spray voltage was set at 1.9 kV. The mass spectrometer was operated in positive ionization and data-dependent acquisition mode, automatically switching between MS1 and MS2 spectra. The MS1 scan range was set to 200–2000 m/z with a resolution of 60,000 and an automatic gain control (AGC) target value of 3 × $10^6$. Up to 15 peptide precursors were selected for MS2 analysis with an isolation width of 2 m/z at the resolution of 30,000 and AGC

target value of 1 × $10^5$. The maximum injection time for both MS1 and MS2 was 100 ms. The normalized collision energy (NCE) was set at 30% for ion fragmentation by higher-energy collisional dissociation (HCD).

For data analysis, the RAW files were searched with MSfragger in FragPipe v20.0 against UniProt *Mus musculus* (mouse) reference database (accessed May 2023) with the addition of more entries of frequently mutated proteins reported before[53,54]. Minimum peptide length was set to 7. The precursor and fragment mass tolerances were set at 20 ppm. The maximum missed cleavage was set to 2. Carbamidomethylation of cysteine was set as a fixed modification and N-terminal acetylation and methionine oxidation were set as variable modifications. The false discovery rate was set at 1% at the peptide and protein levels. For label-free quantification, maxFLQ and match between runs (MBR) options were enabled. Other parameters were used as default.

## Cell viability induced by NP treatment

The viability of CD103+ cDC1s, MC38 cells, and doxorubicin (DOX)-pretreated MC38 cells after co-incubation with NPs with different concentrations was measured using a cell counting kit-8 (CCK-8, Boster Bio). CD103+ cDC1s were seeded at a density of 1.5 × $10^4$ cells/well, while MC38 cells were seeded at 5 × $10^3$ cells/well in clear 96-well tissue plates. For DOX-treated MC38 cells, DOX was added to the MC38 cell culture with a final concentration of 1 µg/mL. After 24 h, AC-NPs, NP$^{Neg}$, and NP$^{PEG}$ (0–1000 µg/mL) were added to each well. The cells were incubated for another 24 h followed by the CCK-8 assay according to the manufacture's protocol. The absorbance was determined at 450 nm using a plate reader. For each scenario of CD103+ cDC1s, MC38, and DOX-pretreated MC38 cells, the control group consisted of cells without NP treatment, with their values normalized to 100%.

## In vitro evaluation of tumor protein uptake by cDC1s and their activation

To assess AC-NPs' efficiency in enhancing tumor protein uptake into cDC1s, DiD- or DiO-labeled NPs were prepared for fluorescent tracking. CD103+ cDC1s were seeded in a 12-well plate (0.5 × $10^6$ per well) overnight and stained with Hoechst 33342. DiD-labeled NPs were then mixed with FITC-labeled tumor lysates, and the mixture was incubated for 30 min. Next, CD103+ DCs were exposed to the mixture at a concentration of 20 µg/mL NPs and 2 µg/mL FITC-labeled tumor lysate. The cells were further incubated with the mixture for 10 min or 4 h. Following the incubation, CD103+ DCs were harvested, washed, and analyzed using flow cytometry (CytoFLEX, Beckman). For imaging purposes, samples were placed in glass-bottom dishes (Thermo Fisher) and imaged using a confocal microscope (ZEISS LSM710). To assess the activation of CD103+ cDC1s, the expression of activation markers on CD103+ DCs was evaluated. CD103+ DCs were initially seeded in a 12-well plate (0.5 × $10^6$ per well) and cultured overnight. Subsequently, 200 µg/mL NPs or 1 µg/mL lipopolysaccharide were added to the cells, along with 20 µg/mL tumor lysates, followed by further incubation for 24 h. The cells were then harvested, stained with antibodies against MHCII, CD80, CD86, CD11c, CD103, and Zombie NIR (BioLegend), and analyzed by flow cytometry (CytoFLEX, Beckman). To evaluate the level of antigen presented on cDC1, a solution containing 100 µg of AC-NPs and 20 µg of ovalbumin was incubated for 30 min. The mixture was then added to the cDC1 cell culture (1 × $10^6$ per well). After a 4-h incubation, free OVA and AC-NPs were washed out. The surface expression of SIINFEKL was subsequently detected by staining with anti-mouse H-2Kb/SIINFEKL (PE-labeled) coupled with flow cytometry analysis after an additional 24-h culture period.

## Whole cell uptake

To evaluate the internalization of tumor cells by CD103+ cDC1s, MC38 cells underwent incubation in culture medium with or without 1 µg/mL of DOX overnight. Afterwards, they were stained with 10 µg/mL DiO for

20 min. Following three washes, 300 μg/mL AC-NP, NP[Neg], and NP[PEG] were added to both DOX-treated and non-treated MC38 cells. The mixture was incubated for 30 min. CD103+ cDC1 were stained with Hoechst 33342 (1:5000 dilution) for 20 min and washed three times. Then, the mixture of NPs and MC38 cells was added to CD103+ cDC1s and incubated for 10 or 30 h. The samples were washed three times with PBS and analyzed by Flow cytometry. The percentage of DiO-labeled MC38 cells captured/internalized by Hoechst 33342-labeled CD103+ cDC1s was analyzed.

### In vivo trafficking of ACT-DC and cell-level distribution of AC-NPs

We evaluated the trafficking ability of ACT-DC to tDLNs. MC38 cells ($0.5 \times 10^6$) were subcutaneously injected into the right flank of C57BL/6 mice. On day 10, a 30 μL intratumoral injection of a model tumor antigen AF647-OVA (50 μg) was administered. Fifteen minutes later, tumors were injected with 30 μL NPs or PBS, followed by an intratumoral injection of $3 \times 10^6$ IVISense DiR-labeled CD103+ DCs another 15 min later. After 6 or 20 h, tumors and tDLNs were collected. LagoX images were taken to assess the amount of OVA and injected CD103+ cDC1s in the organs. The tumors and tDLNs were then processed into a single cell solution. The cells were stained for CD45, CD11c, CD103, CD86, MHCII, CD11b, F4/80, and Zombie NIR, and analyzed by flow cytometry (Aurora).

The cell-level distribution of AC-NPs in the free or ACT-DC form was also assessed. Briefly, DiD-labeled AC-NPs were prepared as previously described. MC38 cells ($0.5 \times 10^6$) were subcutaneously injected into the right flank of C57BL/6 mice. On day 10, an intratumoral injection of ACT-DC (containing $3 \times 10^6$ IVISense DiR-labeled CD103+ cDC1s and 333 μg DiD-labeled AC-NPs) or free DiD-labeled AC-NPs were administered. For ACT-DC formulation injection, AC-NPs were first injected, followed by administration of CD103+ cDC1s 15 min later. Tumors and tDLNs were dissociated 6 h after cDC1 injection and imaged using LagoX to track the distribution of AC-NPs in the organs. The tumors were then processed into a single cell solution, stained for CD45, CD11c, CD103, CD11b, F4/80, CD49b, CD3, CD4, CD8, and Zombie UV, and analyzed using flow cytometry to determine AC-NPs' distribution in different cells.

### Immunostaining and imaging for tumor tissues

To image the resident DCs and ACT-DCs in tumor tissues, MC38 cells ($5 \times 10^5$) were implanted subcutaneously into the right flank of C57BL/6 mice. ACT-DC in 60 μL PBS was intratumorally injected on day 10 after tumor inoculation. Six hours after injection, tumors were collected, embedded in 2% agarose gel (GeneMate, E3119500), and sectioned to 400-μm-thick slices using a vibratome (VT1200S, Leica). Slices were fixed with 2% paraformaldehyde solution for 15 min at room temperature and washed three times in PBS. For immunofluorescence staining, the tumor slices were incubated in a staining buffer (0.5 mL RPMI-1640 cell culture media, 1% IgG-free bovine serum albumin) containing 0.7 μL of DAPI (5 μg/mL), 5 μL of AF647-labeled anti-mouse CD103 antibody (0.5 mg/mL), and 5 μL of FITC-labeled anti-mouse CD11c antibody (0.5 mg/mL) for 18 h at 4 °C under gentle shaking. Stained tumor slices were washed in PBS three times and then cleared by immersion in 100% D-fructose solution (F0127, Sigma-Aldrich) for 30 min under gentle shaking. Non-treated control tumors were processed by the same method. The stained tumor slices were imaged by a confocal fluorescence microscope (Caliber ID, RS-G4) using 20× air and 40× oil objectives.

### Preparation of ex vivo pulsed DC vaccines

To prepare DC vaccines ex vivo loaded with tumor lysate, CD103+ cDC1s were seeded at a concentration of $30 \times 10^6$ cells in 10 mL RPMI-1640 media supplemented with 10% FBS, 1% Penicillin–Streptomycin, 50 μM 2-mercaptoethanol, 200 ng/mL FTL3L, and 2 ng/mL GM-CSF.

The cells were pulsed with 20 μg/mL PIC and tumor lysates that were collected from $60 \times 10^6$ MC38 or CT-2A cells (tumor cells to DCs ratio of 2) for 4 h. Another DC vaccine ex vivo loaded with defined antigens were prepared using the same method except that cDC1s were pulsed with gp100$_{25-33}$ (50 μg/mL) and TRP-2$_{180-188}$ (50 μg/mL) instead of tumor lysate. Cells were then washed three times and resuspended in PBS for subcutaneous injection. The viability of cells was >95%.

### Therapeutic efficacy studies in the MC38 and B16F10 tumor models

ACT-DC's therapeutic efficacy was evaluated in the MC38 and B16F10 tumor models. For the MC38 models, MC38 cells ($3 \times 10^5$ for the small tumor model and $5 \times 10^5$ for the large-established model) were subcutaneously injected into the right flank of female C57BL/6 mice (6–8 weeks of age). Tumor volume was calculated using the formula: (length × width$^2$)/2, with the longest diameter considered as the length and the shortest as the width. When the average tumor volume reached ~35 mm$^3$ (for the small tumor model) or ~100 mm$^3$ (for the large-established model), an intratumoral injection of 0.1 mg/kg doxorubicin was administered to induce antigen release. The next day, tumors were treated intratumorally with different therapies for the first dose, followed by one or two more doses 4–5 days apart. For the ex vivo MC38 tumor lysate-pulsed DC vaccine group, DC vaccines were prepared according to the method described above and subcutaneously administered. For formulations containing anti-PD1 antibody, two doses of 100 μg antibody per dose were intraperitoneally injected on days 1 and 3 following each dose of ACT-DC or other control formulations. For formulations containing NPs plus DCs, 30 μL of NPs were first intratumorally injected, followed by intratumoral administration of DCs after 15 min. The respective formulations contained $3 \times 10^6$ CD103+ DCs and/or NPs containing 40 μg PIC. Mice were euthanized when tumor size reached 20 mm in any dimension or presented significant ulceration (>3 mm in diameter). For the first rechallenge study, $5 \times 10^5$ MC38 cells were subcutaneously injected into the left flank of tumor-free mice on day 86. For the second rechallenge study, $5 \times 10^5$ MC38 cells were injected into the right flank of mice that survived from the first rechallenge on day 167. To analyze the immune cells in the blood, blood was collected from mice 15 days after the second rechallenge and lysed using ACK buffer to obtain peripheral blood mononuclear cells (PBMCs). The cells were stained with antibodies (CD45, CD11c, CD103, CD86, MHCII, CD11b, Gr1, CD49b, CD3, CD4, CD8, IFN-γ, CD62L, CD44, Adpgk tetramer, CD25, FoxP3, and Zombie NIR) and analyzed by flow cytometry (Aurora).

For the B16F10 model, B16F10-Luc cells ($5 \times 10^5$) were injected into the right flank of female C57BL/6 mice (6–8 weeks of age). The mice were treated following a similar schedule and dosing regimen as in the MC38 model. For the ex vivo DC vaccine loaded with gp100$_{25-33}$ and TRP-2$_{180-188}$, the vaccine was prepared as described above and subcutaneously administered. In the B16F10 model, for the first rechallenge, $5 \times 10^5$ B16F10-Luc cells were subcutaneously injected into the left flank of tumor-free mice on day 79. For the second rechallenge, tumor-free mice survived from the first rechallenge received an intravenous injection of $1 \times 10^5$ B16F10-Luc cells on day 159. Blood collection and immunostaining analysis were performed 14–15 days after the first and second rechallenges to assess the immune cells in the blood. For the bilateral B16F10 tumor model, $1 \times 10^6$ B16F10-OVA cells were inoculated into the right flank of female C57BL/6 mice (6–8 weeks of age) on day 0, and $5 \times 10^5$ B16F10 cells were inoculated into the left flank of the same mice on day 3. An intratumoral injection of 0.1 mg/kg doxorubicin was administered to the B16F10-OVA tumors on day 6 to induce antigen release. ACT-DC was intratumorally administered to the B16F10-OVA tumors on days 7 and 12. For treatments involving anti-PD1 antibody, anti-PD1 antibody (100 μg per dose) was intraperitoneally injected on days 8, 11, 13, and 15. Mice were euthanized on day 21 to collect lymph nodes for immune cell analysis.

To investigate the impact of doxorubicin pretreatment and administration route on the therapeutic efficacy of ACT-DC, B16F10 cells ($5 \times 10^5$) were inoculated into the right flank of female C57BL/6 mice (6–8 weeks of age) on day 0. On day 6, mice received one of three treatments: an intratumoral doxorubicin injection (0.1 mg/kg, 50 μL), an intravenous doxorubicin injection (3 mg/kg, 100 μL), or no doxorubicin treatment. ACT-DC was administered intratumorally on days 7 and 12. To evaluate the effects of T cell depletion and egress inhibition on ACT-DC efficacy in the B16F10 tumor model, mice were injected intratumorally with doxorubicin (0.1 mg/kg) on day 6, followed by two doses of ACT-DC on days 7 and 12. For CD4 or CD8 T cell depletion, anti-CD4 or anti-CD8 antibodies (200 μg per dose) were administered intraperitoneally on days 7, 10, 13, and 16. For T cell egress inhibition, FTY720 (3 mg/kg) was administered intraperitoneally on days 7, 9, 11, 13, 15, 17, and 19.

### Orthotopic mouse glioma model, cannula implantation, and treatments
Male/female C57BL/6 mice (6–8 weeks of age) were intracranially implanted with cannulas (Plastics One), and CT-2A glioma cells at $5 \times 10^4$ cells per mouse were implanted following the procedures as described before[59,75]. All the mice were randomly assigned to different treatment groups. Brain-focused radiotherapy was given using a Gammacell 40 Exactor (Best Theratronics) at a 3 Gy daily dose for three consecutive days starting on day 6 after tumor implantation. Some groups of mice received anti-PD1 antibody (100 μg per dose) treatment intraperitoneally on days 9, 11, 13, 15 post-tumor implantation. In some groups, mice also received conventional ex vivo tumor lysate-pulsed DC vaccine (3 million cells per injection) through subcutaneous injection or ACT-DC (3 million cells per injection) intracranially through the implanted cannulas on days 8 and 12 post-tumor implantation. Supportive care of mice post-tumor implantation and treatments was provided in full compliance with the approved animal protocols. Long-term survivor (LTS) mice were euthanized 100 days after initial tumor implantation, and brains were collected for immunophenotypic analysis by flow cytometry. Non-tumor-bearing mice or CT-2A-bearing mice (21 days post-tumor inoculation) were used as controls.

### Immune cell profiling in the tumor and peripheral immune organs
To assess the capability of ACT-DC to generate a systemic immune response and modulate tumor microenvironment, the large-established MC38 tumors were established and treated according to a similar schedule and dosing regimen from the efficacy studies. On day 17, the tumor, spleen, and tDLNs of mice were collected, weighed, and processed into single cells. The total number of cells obtained from each whole tumor/spleen/tDLN were counted and recorded. The cells were washed with PBS, and 1.5 million cells were stained with antibodies with titrated concentrations. Antibodies were obtained from BioLegend including CD45 (catalog # 157616), CD11c (catalog # 117339), CD103 (catalog # 121435), CD86 (catalog # 105043), MHCII (catalog # 107639), CD11b (catalog # 101205), F4/80 (catalog # 123135), CD206 (catalog # 141716), Gr1 (catalog # 108430), CD49b (catalog # 108924), CD3 (catalog # 100272), CD4 (catalog # 100474), CD8 (catalog # 100798), IFN-γ (catalog # 505860), CD62L (catalog # 104410), CD44 (catalog # 103037), CD25 (catalog # 102016), Foxp3 (catalog # 126408), Zombie NIR (catalog # 423106), Zombie UV (catalog # 423107), and aPD1 (catalog # 135221). PE-conjugated Adpgk tetramer and BV421-conjugated Rpl18 tetramer were obtained from the Tetramer Core of the National Institutes of Health (NIH). Stained samples were subsequently measured by flow cytometry (Aurora). The total number of a specific cell population (e.g., effector memory CD8 T cell) in the whole spleen or tDLN was calculated using the following

method. First, the percentage of the specific cell population within live single cells (denoted as A) was determined by analyzing flow cytometry data using FlowJo 10. The total number of the specific cell population per whole organ was then calculated as the product of A and the total number of live cells in that organ. The total number of a specific cell population in 100 mg of tumor was calculated using a similar method.

### Statistical analysis
All experiments were repeated at least two times. All statistical analyses were carried out using GraphPad Prism (version 10). All flow cytometry data were analyzed using FlowJo (version 10). Confocal imaging data were analyzed using Zen Software (Version 3.9) and ImageJ. All data were presented as mean ± s.e.m. Two-sided unpaired Student's $t$-test, one-way ANOVA with Dunnett analysis, two-way ANOVA with Dunnett analysis, or Mann–Whitney test were used to determine significance. For the analysis of Kaplan–Meier survival curves, the two-sided Mantel–Cox test was used. All statistical analyses were performed on GraphPad Prism (version 10).

### Reporting summary
Further information on research design is available in the Nature Portfolio Reporting Summary linked to this article.

### Data availability
The mass spectrometry proteomics data have been deposited to the ProteomeXchange Consortium via the PRIDE partner repository with the dataset identifier PXD062399 at https://www.ebi.ac.uk/pride/archive/projects/PXD062399. The remaining data supporting the results in this study are available within the paper, Supplementary Information, and Source Data file. The raw numbers for charts and graphs are available in the Source Data file whenever possible. Any additional requests for information can be directed to and will be fulfilled by the corresponding authors. Source data are provided with this paper.

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

## Acknowledgements

Z.Z. acknowledges funding from the National Institutes of Health (NIH) (R35GM150507 and R21CA291723), the Medical Research Program of the TEAL foundation, and the Vahlteich Award from the College of Pharmacy at the University of Illinois, Chicago. P.Z. acknowledges funding from NIH (R37CA266487) and SPORE for Translational Approaches to Brain Cancer Developmental Research Program (P50CA221747). We thank the NIH Tetramer Core Facility for providing the tetramers used in this study.

## Author contributions

C.-J.C. and Z.Z. conceived and designed the experiments. C.-J.C., E.Z., D.N.T., E.U., H.L., C.S., J.H., S.H., J.Z., X.C., Q.B., L.Z., P.P., S.M.E., and X.S. performed the experiments. C.-J.C., Z.Z., P.Z., D.N.T., and Y.G. analyzed the data. C.-J.C., Z.Z., P.Z., D.N.T., Y.G., S.S.L., and J.E.B. discussed results. Z.Z. conceived and supervised the project. C.-J.C. and Z.Z. wrote the initial draft of the manuscript. All authors read, revised, and approved the manuscript.

## Competing interests

C.-J.C. and Z.Z. are inventors on a patent application with aspects related to this work (titled "Antigen capturing nanoparticle engineered dendritic cells for in situ cancer immunization", US Provisional Application no. 63/682,533) filed by the University of Illinois Chicago. The remaining authors declare no competing interests.
