## [Transparent Peer Review file · Nature Communications]

Integrating antigen capturing nanoparticles and type 1 conventional dendritic cell therapy for in situ cancer immunization

Corresponding Author: Professor Zongmin Zhao

Version 0:

Reviewer comments:

Reviewer #1

(Remarks to the Author)

The authors present a compelling approach named Antigen Capturing Nanoparticle Transformed Dendritic Cell Therapy (ACT-DC) for in situ cancer immunization. This study integrates antigen-capturing nanoparticles (AC-NPs) with migratory type 1 conventional dendritic cells (cDC1s) to enhance antigen presentation and modulate the tumor microenvironment. The findings demonstrate that ACT-DC, particularly in combination with immune checkpoint inhibitors, significantly improves tumor eradication and induces long-term immune memory across multiple tumor models, including colon cancer, melanoma, and glioma. Overall, this is an interesting study with extensive data, but several key issues need to be addressed. I believe that the manuscript can be considered for publication in Nature Communications following a major revision. Below are the essential points for improvement:

Major Issues:

1. Clarity on the "Trojan Horse" Mechanism: The term "Trojan horse" is mentioned seven times throughout the manuscript, including in the abstract, indicating the authors consider this concept central to the work. However, the relationship between this "Trojan horse" strategy and the core mechanisms of the study is not clearly explained. The manuscript lacks a detailed description of how the "Trojan horse mechanism" manifests in the proposed therapy. I suggest expanding on this point to clarify the underlying processes through which the Trojan horse strategy is realized in ACT-DC therapy.
2. Specificity of Antigen Capture by AC-NPs: According to the manuscript, the AC-NPs capture antigens via electrostatic interactions (contributed by PEI) and hydrophobic forces (from PLGA) to deliver them to cDCs. However, since the AC-NPs do not have a specific binding mechanism for cDCs, the antigen delivery process seems non-specific. How can the authors ensure that the AC-NPs efficiently deliver antigens to cDCs rather than other cell types in the tumor microenvironment? This aspect requires more thorough discussion and perhaps experimental validation to confirm that the antigen capture and delivery are optimally targeted to cDC1s.
3. Data on PIC Loading and Release Kinetics: The manuscript lacks critical data on the drug loading efficiency and encapsulation efficiency of polyinosinic:polycytidylic acid (PIC) within the nanoparticles. Moreover, the release kinetics of PIC should be included, as these are standard parameters that will enhance the reproducibility of the study. Adding these data will also strengthen the technical robustness of the approach.
4. Impact on T Cell Exhaustion: Given the potential for long-term T cell activation in cancer immunotherapy, it is important to investigate whether ACT-DC induces T cell exhaustion. I recommend that the authors assess the expression of T cell exhaustion markers, such as PD1, TIM-3, and LAG-3, to evaluate whether ACT-DC therapy negatively impacts the functionality of tumor-infiltrating T cells.
5. Comparison with Ex Vivo-Loaded DC Therapies: The antigen-capturing ability of AC-NPs is a critical aspect of this therapy, but the manuscript provides insufficient comparisons with traditional ex vivo-loaded DC therapies. To better illustrate the advantage of in vivo antigen capture, the authors should include additional controls or comparisons, such as using ex vivo-loaded DCs with defined antigens. This would provide a direct assessment of the benefits of capturing antigens directly within the tumor microenvironment, as opposed to pre-loading antigens in vitro.
6. Schematic Diagram: To improve the clarity of the manuscript and enhance the reader's understanding of the experimental design and key concepts, I strongly suggest including a schematic that summarizes the overall principles, approach, and experimental design. This would make the manuscript more accessible and reinforce its conceptual framework.

Minor Issues:

1. Reference Formatting: There are some inconsistencies in the reference formatting, particularly with journal abbreviations.

Please ensure that all references follow the correct journal abbreviation format, and that page numbers are complete.

Reviewer #2

(Remarks to the Author)

The Chao et al. reported an Antigen Capturing nanoparticle Transformed Dendritic Cell therapy (ACT-DC) immunotherapy strategy, composed of antigen-capturing nanoparticles (AC-NPs) and migratory type 1 conventional dendritic cells (cDC1s). This approach aims to enable AC-NPs to capture in situ tumor antigens and deliver them to cDC1s, thereby enhancing antigen presentation efficiency and activating systemic immune responses. The authors demonstrated through extensive experiments that AC-NPs could effectively capture tumor-associated antigens and present to cDC1 cells, resulted in the ACT-DC strategy exhibited significant antitumor efficacy across various tumor models. However, the article still presents several unresolved issues:

1. According to the article, AC-NPs are composed of PLGA encapsulating polycyidylic acid (PIC) and is surface-modified with PEI. However, the article lacks structural characterization of AC-NPs. The assembly process of PLGA drug-loaded nanoparticles with hydrophilic PEI needs further clarification, and evidence of successful PEI modification on the PLGA particle surface should be provided. Additionally, the molecular weight of the PEI used should be clearly specified.
2. Figure 1k only presents confocal images showing the uptake of tumor antigens by AC-NPs within cDC1 cell. CLSM images from the other three groups in Figures 1i and 1j should also be displayed. Moreover, while Figure 1k is described as 3D CLSM, but it shown 2D CLSM images.
3. In the investigation of AC-NPs' in vivo distribution (Figure 2e), following the simultaneous injection of AC-NPs and cDC1s, a majority of AC-NPs (77.9%) were found to be taken up by DCs, with only a small portion (21.6%) taken up by tumor cells. Whether the AC-NPs will be highly uptake by cDC1s during their mixing? If is, how can AC-NPs capture tumor antigens after being internalized by cDC1s? The authors should provide data on the uptake of AC-NPs by cDC1s during the construct process of the ACT-DC formulation.
4. In Figure 2j, the authors only detect the fluorescence intensity of AF647-OVA protein antigens in lymph nodes to demonstrate the transfer of captured antigens from AC-NPs to CD103+ cDC1s. A more comprehensive analysis of the number of double-positive cDC1 cells for AC-NPs and AF647-OVA in the lymph nodes would better substantiate this claim.
5. Can the authors provide direct in vivo data showing the transfer of captured antigens from AC-NPs to cDC1 cells in lymph nodes? For instance, immunofluorescence images of lymph nodes would significantly enhance the understand of this process.
6. In multiple tumor model experiments, the authors administered a certain dose of DOX intratumorally prior to the ACT-DC injection to enhance tumor immunogenicity. Could the residual DOX in the tumor tissue potentially kill the injected cDC1 cells? Further evaluation of the retention or metabolism of different formulations (DOX, cDC1 cells, AC-NPs) at the tumor site following sequential injections is warranted.
7. In the therapeutic experiment depicted in Figure 3d, the tumor suppression effects of ACT-DC and ACT-DC+aPD1 show minimal difference (Figure 3e), yet the final survival rate of mice treated with ACT-DC+aPD1 is significantly higher than that of the ACT-DC group (Figure 3f). Further investigation is needed to elucidate the immunological mechanisms underlying the enhanced survival duration of mice following the combination treatment of ACT-DC and aPD1.
8. Figures 3e-h and 3i-j refer to the same cohort of mice, with the entire treatment period lasting 182 days. The survival curves shown in Figures 3f and 3h indicated a higher mortality rate in the ACT-DC treatment group (comprising a total of 15 mice). It is necessary to confirm whether the number of mice in each group is sufficient to support the statistical analysis of the final survival curve experiment presented in Figure 3j. Additionally, the caption should specify the exact number of surviving mice in each group.
9. The TEM statistical data in Figures 4h and 4i are inconsistent. The statistical methods used should be detailed in the methods section to prevent misunderstanding. Additionally, it should be clearly indicated whether Figures 4i and 4j report the total number of TCM and TEM in lymph nodes and spleens, or the cell count per unit weight of tissue.
10. In Figure S8b, the AC-NP and NPNeg exhibited stronger cytotoxicity towards DOX-pretreated MC38 tumor cells compared to the NPPEG group. More discussion should be added to explain this phenomenon.
11. There are several errors in the manuscript, such as the y-axis label "100 gm tumor" in Figure 5e, the incorrect labeling of primary and distant tumors in Figures 6i and 6m, and the absence of a fluorescence intensity scale bar in Figure S11a, and etc..

Reviewer #3

(Remarks to the Author)

In this manuscript, Chao et al. engineered and characterized novel antigen capturing nanoparticles (AC-NPs) designed to capture tumor antigens in situ to efficiently transfer them to the antigen-presenting conventional dendritic cells type 1 (cDC1s), hence inducing T-cell activation and anti-tumor immunity. Given the rarity of cDC1s within the tumor microenvironment (TME), the authors have combined AC-NPs with the adoptive transfer of bone marrow-derived cDC1s and named the approach Antigen Capturing nanoparticles Transformed Dendritic Cell therapy (ACT-DC). ACT-DC synergy with the immune check point blocker (ICB) anti-programmed cell death (PD) 1 induced systemic anti-tumor immunity across multiple tumor models, including the immunogenic MC38 model and the less immunogenic B16 and CT-2A models. ACT-DC induced cDC1 activation and migration toward the lymph node (LN), which reshaped the TME and the LN, leading to an increase in the numbers of IFN- γ + CD8+ and IFN- γ + CD4+ T cells. In general, the therapeutic efficacy of ACT-DC is good, the figures are informative and the experiments are well-designed. Although this study is relevant, there are some concerns that need to be clarified or experimentally addressed.

Major concerns:

- The title of Fig. 3 is misleading. The authors claim that that ACT-DC eradicated small tumors as a standalone therapy, however, the tumors were pre-treated with doxorubicin. Also, in the text, the mentioning “standalone treatment” is misleading since mice are treated with dox. What would the ACT-DC therapeutic efficacy give without dox?
- AC-NP was shown to result in increased cell death in doxorubicin-treated MC38 cancer cells. Why is this the case? An explanation or speculation would be helpful.
- The impact of ACT-DC on specific CD8+ T-cell populations remains unclear. It is recommended to integrate CD8+ T-cell differentiation, exhaustion and cytotoxic markers, including TCF1, PD1, TIM3, Lag3, perforins and granzymes into the flow cytometry analysis.
- Given that ACT-DC increased the numbers of CD8+ T cells, CD4+ T cells, IFN- γ + CD8+ T cells and IFN- γ + CD4+ T cells within the TME, evaluating the impact of CD8+ and/or CD4+ T cell-depletion on the therapeutic efficacy of the ACT-DC would be necessary to assess their role in the therapeutic efficacy.
- Given that ACT-DC increased the numbers of tetramer+ CD8+ T cells, CD8+ T cells, CD4+ T cells, IFN- γ + CD8+ T cells and IFN- γ + CD4+ T cells within the tdLN, it is intriguing to assess if the anti-tumor efficacy is dependent on T-cell trafficking and T-cell egress from the LN. To this the end, the therapeutic efficacy of ACT-DC could be evaluated in the presence or absence of the S1PR inhibitor FTY720.
- It would also be interesting to assess the contribution of the tumor-residing cDC1s compared to the transferred cDC1s, for example by doing the therapy experiment in Batf3KO or XCR1-DTR mice.

Minor concerns:

- For full transparency, please also show the pre-gating strategy in fig S4. Since the expression of Clec9a is relatively low, to which extent does the gated population also contains cDC2s or monocyte-derived cells?
- In the legend of Fig. 2b-c, it says that the mice were either untreated or injected with ACT-DC, but in the bar graph it says otherwise. Were they untreated or treated with AC-NP?
- Line 167: replace “macrophage” by “myeloid cell” in the text, as these could also be neutrophils or other CD11b+CD11c cells.
- Add statistics in Fig S8.
- In Fig. S14 and S15, the ACT-DC-treated group had a total of 15 mice while all the other treatment groups had 7-8 mice each. This can be misleading and the reasons behind this difference should be pointed out.
- The arrangement of the panels in Fig. 4 is slightly confusing and in 4h it should be “ACT-DC” instead of “ACD-DC”.

Reviewer #4

(Remarks to the Author)

Chao et al have demonstrated a novel therapy called ACT-DC which consist of antigen capturing nanoparticles that enhance migratory cDC1 activation and T cell immune responses. They demonstrated this with various tumor models (MC38, B16F10, B16F10-OVA and CT-2A) which reinforced the strength of such therapy. However, some experiments were not done under physiological conditions hence limiting the strength of some conclusion. Moreover, some more experiments should be done to fully demonstrate the efficacy of the ACT-DC.

Major comments :

- 1) In figure 1E, could the authors please specify how were they able to quantify the specific number of proteins retained? Especially for tumor lysate.
- 2) In figure 1, the authors examined the uptake of the ACT-DC with cDC1. However, this model was very subjective since the BMDCs were forced to generate cDC1. It would be interesting if the authors could test this in a more physiological representation. In general, cDC2 exceed the amount of cDC1 in both humans and mice. Hence, could the authors re-test the uptake of the ACT-DC in mixture of cDC1 and cDC2. This would provide an actual advantage of this system.
- 3) With regard to in vivo trafficking of ACT-DC and cell-level distribution of AC-NPs, why was AF647-OVA (50 μ g) administered? B16OVA by itself should be expressing OVA. This system is again forcing certain results and not demonstrating physiological/clinical conditions.
- 4) Why is doxorubicin injected intratumorally (IT)? The normal dose of such therapy in the clinic is IV. Moreover, nanoparticles are given IV also in the clinic. These conditions are not realistic. Could the authors please repeat therapeutic experiments where therapies are administered according to FDA/EMA accepted routes?
- 5) In the therapeutic models, could the authors shed some light on the activation status of the DC cells? It was shown in previous experiments and even hypothesized that ACT-DC could increase activation markers for DCs.
- 6) In figure 3, the authors have nice data concerning the memory phenotyping of CD8+ T cells and tetramer staining. Could

the authors correlate the enhanced TEM population with the tetramer staining. This would really provide solid information that the ACT-DC therapy is inducing a tumor specific TEM population.

7) In figure 4 and 5, the authors are representing some graphs with cell number. However, in the methods it is not described how this is done. Was this done with counting beads or what other methods were used to quantify the number of cells?

8) The authors show a nice increase of TEM CD4+ population with B16F10. However, they do not demonstrate this with MC38. Could this be done?

9) In figure 6i, how sure are the authors that the ACT-DC therapy induced a strong and effective T cell populations against both OVA or TRP2? Could it not be that the TRP2 specific T cells induced tumor killing in the B16F10-OVA since they also express TRP2? The authors should try to isolate the TRP2 and OVA specific T cells to demonstrate that the therapy did indeed induce two powerful T cell populations against two different epitopes. This would show a clear mitigation of immunodominance and advantage to combat tumor heterogeneity.

10) Could the authors clarify how were TAMCs phenotyped?

Minor comments:

1) Line 46-49, DCs used to target multiple antigens is not the most effective strategy. Immunodominance can exist and limit multiple T cell responses against different epitopes. Even in the reference used they state this. Hence, I would not word it in such a form but rather saying that DCs can be used to tackle different antigens which may be personalized with different patients

2) In figure 2m, why is there count on the y-axis? The count should then give you a histogram plot and not what is represented.

3) In lines 174-175 "These retained cDC1s, activated by AC-NPs, could reshape the local tumor microenvironment, enhancing immune cell infiltration after induction of an antitumor immune response in the tDLNs." This is discussion and not results. Please remove this and add it to the discussion section.

4) The authors would further boost the quality of this paper if they could further look at the activated and exhausted profile of the T cells. By simply using PD-1 and TIGIT/LAG3/TIM3 this could be achieved.

Version 1:

Reviewer comments:

Reviewer #1

(Remarks to the Author)

The authors have added substantial amount of data and my pervious concerns have been addressed. This paper is a significant contribution to the field, I have no further questions.

Reviewer #2

(Remarks to the Author)

The authors have addressed all my questions. I recommend it to be accepted for publication without any further changes.

Reviewer #3

(Remarks to the Author)

The authors addressed all my comments through new experiments or discussion of results, significantly improving the manuscript, which I now consider suitable for publication.

Reviewer #4

(Remarks to the Author)

the authors have replied to all our comments sufficiently. I believe the manuscript is ready for acceptance.

Response to reviewer comments

We thank the reviewers for their constructive comments and feedback. We have thoroughly addressed all comments by conducting additional experiments and further clarifications. Point-by-point response to reviewer comments are shown below. All responses to reviewer comments are shown as *blue* text. All related changes in the manuscript and supplementary information have been highlighted in *yellow* in this response letter and in the revised manuscript.

Reviewer #1 (Remarks to the Author): with expertise in nanomedicine, cancer therapy

The authors present a compelling approach named Antigen Capturing Nanoparticle Transformed Dendritic Cell Therapy (ACT-DC) for *in situ* cancer immunization. This study integrates antigen-capturing nanoparticles (AC-NPs) with migratory type 1 conventional dendritic cells (cDC1s) to enhance antigen presentation and modulate the tumor microenvironment. The findings demonstrate that ACT-DC, particularly in combination with immune checkpoint inhibitors, significantly improves tumor eradication and induces long-term immune memory across multiple tumor models, including colon cancer, melanoma, and glioma. Overall, this is an interesting study with extensive data, but several key issues need to be addressed. I believe that the manuscript can be considered for publication in Nature Communications following a major revision.

Response: We appreciate the reviewer's constructive suggestions to further improve the manuscript. All the reviewer's comments have been carefully considered, addressed, and incorporated in the revised manuscript.

Major Issues:

1. Clarity on the "Trojan Horse" Mechanism: The term "Trojan horse" is mentioned seven times throughout the manuscript, including in the abstract, indicating the authors consider this concept central to the work. However, the relationship between this "Trojan horse" strategy and the core mechanisms of the study is not clearly explained. The manuscript lacks a detailed description of how the "Trojan horse mechanism" manifests in the proposed therapy. I suggest expanding on this point to clarify the underlying processes through which the Trojan horse strategy is realized in ACT-DC therapy.

Response: We thank the reviewer for raising this important point. Upon more thorough rethinking, we concluded that "Trojan horse" is not the most appropriate word to describe the mechanism of ACT-DC, as in ACT-DC, the tumor antigens are not hidden or smuggled into the immune system, but rather actively captured and transported by cDC1s to lymph nodes for immune activation. Toward this end, we think "cascade antigen relay" is a more appropriate word to precisely describe the mechanism of ACT-DC, as it effectively conveys the stepwise, sequential transfer of tumor antigens from the tumor site to cDC1s and then to T cells in the lymph nodes.

We have replaced "Trojan horse" with "cascade antigen relay" in the revised manuscript. Moreover, per the reviewer's suggestion, we have added more discussions to further clarify the cascade antigen relay processes triggered by the ACT-DC therapy in the Discussion section.

*"Notably, ACT-DC employs a cascade antigen relay mechanism to achieve *in situ* immunization (Fig. 8). Specifically, upon intratumoral administration, AC-NPs capture native tumor antigens and *in situ* deliver them to as well as activate migratory CD103+ cDC1s. This process triggers the active trafficking of these cDC1s to tDLNs and thus relays antigen transport to tDLNs for efficient T cell priming. In tDLNs, cDC1s efficiently present captured tumor antigens to naïve CD8 T cells, inducing a polyclonal, antigen-specific T cell response that overcomes antigen heterogeneity."*

2. Specificity of Antigen Capture by AC-NPs: According to the manuscript, the AC-NPs capture antigens via electrostatic interactions (contributed by PEI) and hydrophobic forces (from PLGA) to deliver them to cDCs. However, since the AC-NPs do not have a specific binding mechanism for cDC1s, the antigen delivery process seems non-specific. How can the authors ensure that the AC-NPs efficiently deliver antigens to cDC1s rather than

other cell types in the tumor microenvironment? This aspect requires more thorough discussion and perhaps experimental validation to confirm that the antigen capture and delivery are optimally targeted to cDC1s.

Response: We thank the reviewer for this question. Our data from the cellular-level distribution of AC-NPs upon intratumoral administration of ACT-DC in the original manuscript (**Fig. 2e-g**) suggests that when delivered as an ACT-DC formulation, AC-NPs were mainly delivered to the adoptively injected cDC1s compared to other tumor-resident cells. To further validate the antigen delivery, we conducted a new experiment in which a model tumor antigen (Alexa Fluor 488-OVA), AC-NPs, and cDC1s were sequentially injected intratumorally and the cellular-level distribution of AF488-OVA within tumor was analyzed (**Fig. S15**). Our new data indicates that the adoptively transferred cDC1s contained a markedly larger amount of AF488-OVA than other tumor-resident cells, indicating the dominant targeting of captured antigens to adoptively injected cDC1s.

The data above collectively suggest that both AC-NPs and the model tumor antigen were predominantly taken up by the adoptively injected cDC1s compared to other immune cells within the tumor. Although AC-NPs don't have an active targeting ligand specific for cDC1s, two key factors likely contributed to their dominant delivery to cDC1s. First, in the ACT-DC approach, AC-NPs and cDC1s were sequentially (15 minutes apart) injected intratumorally to the same or nearby location. This spatial proximity and co-localization at the inject site and surrounding diffusion areas increased the likelihood of AC-NPs being internalized by cDC1s. Second, as professional antigen presenting cells with strong antigen-processing capabilities, cDC1s are known for their superior ability to recognize and take up particulate matter, such as damaged tumor cells and large tumor cell debris. Our data indicates that upon capturing tumor proteins, AC-NPs undergo size increase (Fig. S3). The particulate nature of AC-NPs combined with the display of captured tumor antigens and DAMPs could trigger and enhance their recognition and uptake by cDC1s. These factors collectively lead to the dominant delivery of AC-NPs to adoptively injected cDC1s despite AC-NPs not having an active targeting ligand for cDC1s.

We have added the new data (**Fig. S15**) and more thorough discussions to the revised manuscript.

“Moreover, in a separate study, where a model tumor antigen (AF488-OVA), AC-NPs, and cDC1s were sequentially administered intratumorally (**Fig. S15a**), we observed a markedly higher uptake of AF488-OVA by the adoptively injected cDC1s compared to other resident cells in the tumor (**Fig. S15b**).”

“Our cellular-level biodistribution data suggested that both AC-NPs and the model tumor antigen were predominantly taken up by the adoptively injected cDC1s compared to other immune cells within the tumor. Although AC-NPs don't have an active targeting ligand specific for cDC1s, two key factors likely contributed to their dominant delivery to cDC1s. First, in the ACT-DC approach, AC-NPs and cDC1s were sequentially (15 minutes apart) injected intratumorally to the same or nearby location. This spatial proximity and co-localization at the inject site and surrounding diffusion areas increased the likelihood of AC-NPs being internalized by cDC1s. Second, as professional antigen presenting cells with strong antigen-processing capabilities, cDC1s are known for their superior ability to recognize and take up particulate matter, such as damaged tumor cells and large tumor cell debris. Our data indicates that upon capturing tumor proteins, AC-NPs undergo size increase. The particulate nature of AC-NPs combined with the display of captured tumor antigens and DAMPs could trigger and enhance their recognition and uptake by cDC1s. These factors collectively lead to the dominant delivery of AC-NPs to adoptively injected cDC1s despite AC-NPs not having an active targeting ligand for cDC1s.”

Fig. S15. Relative amount of model antigen OVA in adoptively injected cDC1s and other tumor-resident cells following intratumoral administration of ACT-DC in the MC38 tumor model. a, Schematic illustrating the experimental design. A model tumor antigen (AF488-OVA) was first intratumorally injected. After 15 minutes, AC-NPs were intratumorally injected, followed by intratumoral injection of IVISense DiR labeled cDC1s another 15 minutes later. Tumors were collected 20 hours post cDC1 injection and processed to quantify the relative amount of AF488-OVA in different cell populations using flow cytometry. **b,** MFI of AF488-OVA in various cells in tumors.

Editorial note: Panel a created in BioRender. Zhao, Z. (2025) <https://BioRender.com/3ow34vf>

3.Data on PIC Loading and Release Kinetics: The manuscript lacks critical data on the drug loading efficiency and encapsulation efficiency of polyinosinic:polycytidylic acid (PIC) within the nanoparticles. Moreover, the release kinetics of PIC should be included, as these are standard parameters that will enhance the reproducibility of the study. Adding these data will also strengthen the technical robustness of the approach.

Response: We thank the reviewer for this suggestion. In response to the reviewer’s suggestion, we have added the following data and related discussions into the revised manuscript: PIC loading efficiency, PIC encapsulation efficiency, and PIC release kinetics (new Fig. S2 and Supplementary Table 1).

“PIC were loaded into AC-NPs with a loading capacity of 128 $\mu\text{g}/\text{mg}$ (Supplementary Table 1). The release of PIC from AC-NPs followed a sustained pattern, with a cumulative 45.5% of drug released over 72 hours (Fig. S2).”

Fig. S2. Profiles of PIC release from nanoparticles. Release study was conducted in PBS (10 mM, pH 7.4) at 37 °C. n=3 independent samples for each nanoparticle type.

4.Impact on T Cell Exhaustion: Given the potential for long-term T cell activation in cancer immunotherapy, it is important to investigate whether ACT-DC induces T cell exhaustion. I recommend that the authors assess the expression of T cell exhaustion markers, such as PD1, TIM-3, and LAG-3, to evaluate whether ACT-DC therapy negatively impacts the functionality of tumor-infiltrating T cells.

Response: We thank the reviewer for this suggestion. We have conducted a new experiment to assess the expression of exhaustion markers on tumor-infiltrating T cells. Our data (Fig. S32) suggests that overall, ACT-DC, either alone or in combination with anti-PD1 antibody did not markedly increase T cell exhaustion. Specifically, ACT-DC (with or without anti-PD1 antibody) did not significantly increase the expression of PD-1, TIM-3, or LAG-3 on intratumoral CD4 T cells. As for intratumoral CD8 T cells, ACT-DC (with or without anti-PD1 antibody) treatment significantly increased the expression of TIM-3 but not LAG-3 or PD-1. As TIM-3 expression is elevated on CD8 T cells after ACT-DC treatment, combination of ACT-DC with TIM-3 blockade may provide a new strategy to further enhance the therapeutic efficacy.

We have added the new data (Fig. S32) and corresponding discussions in the revised manuscript.

“Furthermore, data from a separate study indicates that ACT did not significantly increase the expression of T cell exhaustion markers (LAG-3, TIM-3, and PD-1) on intratumoral CD4 T cells (Fig. S32a-c). However, ACT-DC led to a significant increase in TIM-3 expression on intratumoral CD8 T cells, without affecting LAG-3 or PD-1 levels (Fig. S32d-e). These findings suggest that combining ACT-DC with TIM-3 blockade could be a promising strategy to further enhance the therapeutic efficacy of ACT-DC.”

Fig. S32. Expression of exhaustion markers on intratumoral CD4 and CD8 T cells after two doses of ACT-DC therapy in the MC38 tumor model. a-c, Relative expression levels of LAG-3 (a), TIM-3 (b), and PD-1 (c) on CD4 T cells in tumors. d-f, Relative expression levels of LAG-3 (d), TIM-3 (e), and PD-1 (f) on CD8 T cells in tumors. Statistical analysis was performed using one-way ANOVA followed by Dunnett test: * P < 0.05, ** P < 0.01, n.s. indicates not significantly different. These data were collected from a separate study according to the same experimental schedule shown in Fig. 5a. Fig. S27, Fig. S30, and Fig. 32 were from the same study.

5. Comparison with Ex Vivo-Loaded DC Therapies: The antigen-capturing ability of AC-NPs is a critical aspect of this therapy, but the manuscript provides insufficient comparisons with traditional ex vivo-loaded DC therapies. To better illustrate the advantage of in vivo antigen capture, the authors should include additional controls or comparisons, such as using ex vivo-loaded DCs with defined antigens. This would provide a direct assessment of the benefits of capturing antigens directly within the tumor microenvironment, as opposed to pre-loading antigens in vitro.

Response: We thank the reviewer for this suggestion. In our original manuscript, we have compared ACT-DC to ex vivo tumor lysate-loaded DC therapy, which is a conventional ex vivo DC therapy tested in clinical trials, in both the MC38 and CT-2A models (Fig. S23 and Fig. 7b). Our data show that ACT-DC outperforms this type of conventional ex vivo DC therapy in both models, suggesting the benefits of in situ immunization enabled by ACT-DC over pre-loading antigens in vitro.

In response to the reviewer's comment, we have conducted a new experiment to compare the therapeutic efficacy of ACT-DC to that of another type of ex vivo DC therapy loaded with two neoantigens (gp100₂₅₋₃₃ and TRP-2₁₈₀₋₁₈₈) in the B16F10 model. Our data indicate that ACT-DC showed significantly better efficacy than the ex vivo DC therapy loaded with two defined antigens (Fig. S38), indicating again the benefits of ACT-DC-enabled in situ vaccination over conventional ex vivo antigen pre-loaded DC therapies.

We have added the new data (Fig. 38) and more discussions in the revised manuscript.

“In a separate study, we directly compared ACT-DC with a conventional ex vivo DC vaccine loaded with two defined antigens (gp100₂₅₋₃₃ and TRP-2₁₈₀₋₁₈₈). ACT-DC demonstrated significantly better therapeutic efficacy (Fig. S38), further indicating the advantages of the in situ immunization approach enabled by ACT-DC.”

“ACT-DC demonstrated better therapeutic efficacy than two types of conventional DC vaccines that were ex vivo activated and loaded with tumor lysate derived antigens or defined antigens. Unlike ex vivo DC vaccines, ACT-DC uses the native antigens directly from a tumor and may provide a platform approach to produce DC therapies in vivo, facilitating a robust antitumor immune response in a patient-specific manner.”

Fig. S38. Comparison of the therapeutic efficacy of ACT-DC to ex vivo DC vaccine loaded with gp100₂₅₋₃₃ and TRP-2₁₈₀₋₁₈₈ in the B16F10 tumor model. Tumor growth curve is shown (n=7 independent animals per group). Data are expressed as mean ± SEM. Statistical analysis was performed using two-way ANOVA: **** P < 0.0001.

6. Schematic Diagram: To improve the clarity of the manuscript and enhance the reader's understanding of the experimental design and key concepts, I strongly suggest including a schematic that summarizes the overall

principles, approach, and experimental design. This would make the manuscript more accessible and reinforce its conceptual framework.

Response: We appreciate the reviewer for this suggestion. Per the reviewer's suggestion, we have added a schematic diagram (Fig. 8) to illustrate the conceptual design of ACT-DC.

Fig. 8. Schematic illustration of the mechanism of ACT-DC for *in situ* immunization against solid tumors. By integrating AC-NPs with adoptive transfer of cDC1s, ACT-DC harnesses native tumor antigens to induce potent and long-lasting systemic immune responses against solid tumors. Upon intratumoral administration, AC-NPs capture native tumor antigens *in situ* (a) and enhance their delivery to migratory CD103+ cDC1s while simultaneously activating them (b). The activated cDC1s then migrate to tDLNs, where they enhance antigen presentation, leading to the induction of potent polyclonal antigen-specific T cells and memory T cells (c2). Meanwhile, activated cDC1s retained within the tumor modulate the tumor microenvironment to reduce immunosuppression and enhance T cell infiltration (c1). The T cells generated in tDLNs effectively infiltrate tumors, leading to the eradication of solid tumors. Created with Biorender.

Editorial note: Created in BioRender. Zhao, Z. (2025) <https://BioRender.com/qgi6dnp>

Minor Issues:

1. Reference Formatting: There are some inconsistencies in reference formatting, particularly with journal abbreviations. Please ensure that all references follow the correct journal abbreviation format, and that page numbers are complete.

Response: Thank you for bringing this to our attention. We have carefully checked all references to ensure the consistency of the formatting.

Reviewer #2 (Remarks to the Author): with expertise in nanomedicine, cancer therapy

The Chao et al. reported an Antigen Capturing nanoparticle Transformed Dendritic Cell therapy (ACT-DC) immunotherapy strategy, composed of antigen-capturing nanoparticles (AC-NPs) and migratory type 1 conventional dendritic cells (cDC1s). This approach aims to enable AC-NPs to capture in situ tumor antigens and deliver them to cDC1s, thereby enhancing antigen presentation efficiency and activating systemic immune responses. The authors demonstrated through extensive experiments that AC-NPs could effectively capture tumor-associated antigens and present to cDC1 cells, resulting in the ACT-DC strategy exhibited significant antitumor efficacy across various tumor models. However, the article still presents several unresolved issues:

Response: We appreciate the reviewer's constructive suggestions to further improve the manuscript. All the reviewer's comments have been considered, addressed accordingly, and incorporated in the revised manuscript.

1. According to the article, AC-NPs are composed of PLGA encapsulating polycytidylic acid (PIC) and is surface-modified with PEI. However, the article lacks structural characterization of AC-NPs. The assembly process of PLGA drug-loaded nanoparticles with hydrophilic PEI needs further clarification, and evidence of successful PEI modification on the PLGA particle surface should be provided. Additionally, the molecular weight of the PEI used should be clearly specified.

Response: We thank the reviewer for this comment. In response to the reviewer's comment, we have further clarified the preparation process of the AC-NPs and specified the molecular weight of PEI in the Method section. We have also conducted a new SEM EDS experiment to provide additional evidence to show the successful incorporation of PEI to AC-NPs.

Our zeta-potential data (**Fig. 1d**) show that upon incorporation of PEI, the nanoparticle surface charge shifted from -20mV (NP^{Neg}) to +40mV (AC-NP). The only difference between NP^{Neg} and AC-NP is that AC-NP has PEI incorporated. The positive surface charge of AC-NP is evidence indicating the successful incorporation of PEI into the nanoparticle. Additionally, we have conducted a new scanning electron microscopy (SEM)- energy-dispersive X-ray spectroscopy (EDS) study to further characterize the presence of PEI in AC-NP. As shown in **Fig. S1**, AC-NPs (composed of PLGA and PEI) show a characteristic N peak while NP^{Neg} (composed of only PLGA) did not show. This SEM-EDS data provided additional evidence for the successful incorporation of PEI in AC-NPs.

We have added the new data (**Fig. S1**) and more discussions in the revised manuscript.

“The energy-dispersive X-ray spectroscopy (EDS) analysis revealed that blank AC-NP shows a characteristic nitrogen peak derived from PEI in its elemental spectrum (**Fig. S1**), further indicating the successful incorporation of PEI to AC-NP.”

Fig. S1. EDS profiles of NPs. SEM images and corresponding element spectrum profiles of (a) NP^{Neg} (composed of only PLGA) and (b) AC-NP (composed of PLGA and PEI) were shown.

2. Figure 1k only presents confocal images showing the uptake of tumor antigens by AC-NPs within cDC1 cell. CLSM images from the other three groups in Figures 1i and 1j should also be displayed. Moreover, while Figure 1k is described as 3D CLSM, but it shown 2D CLSM images.

Response: We thank the reviewer for this suggestion. Per the suggestion, we have added additional CLSM data to show the uptake of tumor antigens by other control nanoparticles (Fig. S9). We have also corrected the word “3D” to avoid confusion.

Fig. S9. Representative confocal microscopic images of cDC1 incubated with MC38 tumor lysates and different NPs for 4 hours. Tumor lysate was labeled by FITC. NPs were labeled by DiD. Scale bar: 10 μ m.

3. In the investigation of AC-NPs' in vivo distribution (Figure 2e), following the simultaneous injection of AC-NPs and cDC1s, a majority of AC-NPs (77.9%) were found to be taken up by DCs, with only a small portion (21.6%) taken up by tumor cells. Whether the AC-NPs will be highly uptake by cDC1s during their mixing? If is,

how can AC-NPs capture tumor antigens after being internalized by cDC1s? The authors should provide data on the uptake of AC-NPs by cDC1s during the construct process of the ACT-DC formulation.

Response: We thank the reviewer for this question. For ACT-DC administration throughout the entire manuscript, we did not pre-mix cDC1s with AC-NPs. Instead, AC-NPs were first injected and cDC1s were then injected 15 mins later into the same/nearby spot. In response to the reviewer's comment, we have modified the schematic (Fig. 2e) to avoid confusion and added further clarifications.

“Notably, in all studies involving the injection of ACT-DC, AC-NPs were administered first, followed by cDC1s, with a 15-minute interval between the two injections.”

“On day 10, an intratumoral injection of ACT-DC (containing 3×10^6 IVISense DiR labeled CD103+ cDC1s and 333 μ g DiD-labeled AC-NPs) or free DiD-labeled AC-NPs were administered. For ACT-DC formulation injection, AC-NPs were first injected, followed by administration of CD103+ cDC1s 15 minutes later. Tumors and tDLNs were dissociated 6 hours after cDC1 injection and imaged using LagoX to track the distribution of AC-NPs in the organs. The tumors were then processed into a single cell solution, stained for CD45, CD11c, CD103, CD11b, F4/80, CD49b, CD3, CD4, CD8, and Zombie UV, and analyzed using flow cytometry to determine AC-NPs' distribution in different cells.”

4. In Figure 2j, the authors only detect the fluorescence intensity of AF647-OVA protein antigens in lymph nodes to demonstrate the transfer of captured antigens from AC-NPs to CD103+ cDC1s. A more comprehensive analysis of the number of double-positive cDC1 cells for AC-NPs and AF647-OVA in the lymph nodes would better substantiate this claim.

Response: We thank the reviewer for this suggestion. In response to this suggestion, we have performed an additional experiment to quantify the number of double positive cDC1s for AC-NPs and AF488-OVA. Our data suggests that ACT-DC led to a 2.7- to 3.8-fold higher number of double positive injected cDC1s compared to the combination of cDC1 with control nanoparticles. This data (Fig. S19) and relevant discussions have been added to the revised manuscript.

“Furthermore, data from a separate study showed that ACT-DC led to a 2.7- to 3.8-fold higher number of OVA and NP double-positive injected cDC1s in the tDLNs, compared to cDC1s with control NPs (Fig. S19).”

Fig. S19. ACT-DC enhanced the accumulation of model tumor antigen and NP double positive cDC1s in tDLNs in the MC38 tumor model. a, Schematic depicting the experimental design to measure the number of injected cDC1s that are double positive for model tumor antigen and NPs. **b**, Relative number of double positive injected cDC1s in tDLNs. For **b**, statistical analysis was performed using one-way ANOVA followed by Dunnett test: ** P < 0.01, *** P < 0.001. n=4 biologically independent animals per group.

5. Can the authors provide direct *in vivo* data showing the transfer of captured antigens from AC-NPs to cDC1 cells in lymph nodes? For instance, immunofluorescence images of lymph nodes would significantly enhance the understanding of this process.

Response: We thank the reviewer for this suggestion. In response to this suggestion, we have performed an additional experiment to image the co-localization of injected cDC1s (IVISense 680 labeled) with fluorescently labeled model antigen (AF488-OVA) and AC-NP (DiI labeled). Our data show the co-localization of OVA, AC-NP, and injected cDC1s in the tumor draining lymph node, providing direct *in vivo* evidence to show the transfer of captured antigens from AC-NPs to cDC1 cells in lymph nodes. This data (**Fig. S20**) and corresponding discussions have been added to the revised manuscript.

“Additionally, CLSM imaging confirmed the co-localization of OVA and AC-NPs within the injected cDC1s in the tDLNs (Fig. S20).”

Fig. S20. CLSM images of tDLN showing the presence of AC-NPs and the model antigen AF488-OVA within the injected cDC1s that migrated to tDLNs. This study was conducted in the MC38 tumor model. OVA (AF488 labeled), AC-NP (DiI labeled), and cDC1s (IVISense 680 labeled) were sequentially intratumorally injected (15 mins apart). tDLN was collected 20 hours after cDC1 injection. Scale bars: 20 μ m.

6. In multiple tumor model experiments, the authors administered a certain dose of DOX intratumorally prior to the ACT-DC injection to enhance tumor immunogenicity. Could the residual DOX in the tumor tissue potentially kill the injected cDC1 cells? Further evaluation of the retention or metabolism of different formulations (DOX, cDC1 cells, AC-NPs) at the tumor site following sequential injections is warranted.

Response: We thank the reviewer for this comment. Per the reviewer’s suggestion, we conducted a new experiment to assess the retention of DOX, cDC1s, and AC-NPs at the tumor site after sequential injection. As shown in **Fig. S18**, DOX quickly diffused from the injection site and its signal declined to close to the background value 24 hours after DOX injection (before AC-NP and cDC1s administration). As a result, the residual DOX will unlikely kill the injected cDC1 cells. We have more discussions on this aspect in the revised manuscript.

In addition, our data suggests that cDC1 and AC-NPs signals at the tumor injection site continue to decrease after administration and remain detectable for >7 days. Also, on day 9, injected cDC1s could still be detected in the tumor draining lymph node. We have added the new data (**Fig. S18**) and corresponding discussions in the revised manuscript.

“We subsequently assessed the intratumoral retention of doxorubicin, AC-NP, and cDC1 over time following serial intratumor administration (Fig. S18a). Upon injection, doxorubicin rapidly diffused from the injection site, and by 24 hours post-injection (prior to ACT-DC administration), its signal was almost undetectable (Fig. S18b-c). In contrast, cDC1 and AC-NP exhibited prolonged retention at the injection site (Fig. S18d-f), with their

signals gradually declining over >7 days and becoming undetectable by day 9. Additionally, further imaging of tDLNs revealed that the intratumorally injected cDC1s remained detectable in tDLNs even 9 days after ACT-DC administration (Fig. S18g).”

Fig. S18. Retention kinetics of doxorubicin (DOX) and ACT-DC (AC-NPs and cDC1s) following sequential intratumoral injection. **a**, Schematic showing experimental design. **b**, LagoX images of DOX signals in MC38 tumor mice at different time points. **c**, Fluorescence intensity of DOX at the tumor site following intratumoral injection. Dashed line indicates the background value from the control mice at 24 h. **d**, LagoX images of cDC1 and AC-NP signals at different time points. **e**, Fluorescence intensity of cDC1 (labeled by IVISense DiR) at the tumor site. Dashed line indicates the background value from the control mice on day 9. **f**, Fluorescence intensity

of AC-NP (labeled by DiD) at the tumor site. Dashed line indicates the background value from the control mice on day 9. g, LagoX images of tDLNs collected on day 9.

Editorial note: Panel a created in BioRender. Zhao, Z. (2025) <https://BioRender.com/3ow34vf>

7. In the therapeutic experiment depicted in Figure 3d, the tumor suppression effects of ACT-DC and ACT-DC+aPD1 show minimal difference (Figure 3e), yet the final survival rate of mice treated with ACT-DC+aPD1 is significantly higher than that of the ACT-DC group (Figure 3f). Further investigation is needed to elucidate the immunological mechanisms underlying the enhanced survival duration of mice following the combination treatment of ACT-DC and aPD1.

Response: We thank the reviewer for this question. The reason why ACT-DC and ACT-DC+aPD1 led to similar average tumor volume curve but significantly different survival was because the ACT-DC+aPD1 group has two non-responder mice (as shown in the individual tumor growth curve in **Fig. S22**). The large volume of these two non-responding tumors raised the average tumor volume of the ACT-DC+aPD1 group similar to that of the ACT-DC group. We have added further discussions to explain this in the revised manuscript.

We have conducted detailed immune cell profiling in the tDLNs, tumor, and spleen following different treatments including ACT-DC and ACT-DC+aPD1 (Fig. 4 and 5). Our data suggest that compared to ACT-DC alone, ACT-DC+aPD1 induced markedly higher numbers of Adpgk antigen-specific T cells in the tDLNs and effector memory CD8 T cells in the tDLN & spleen, which could contribute to the better efficacy of ACT-DC+aPD1 in extending animal survival and rejecting tumor rechallenges than ACT-DC alone. Additionally, it is important to note that although ACT-DC and ACT-DC+aPD1 show comparable efficacy in enhancing T cell infiltration into tumors (Fig. 5), aPD1 could block the immune checkpoint on intratumoral T cells, enhancing their efficacy in killing tumor cells. These factors collectively lead to the better effectiveness of ACT-DC+aPD1 over ACT-DC.

“Notably, the similar average tumor volume curves observed between the ACT-DC and ACT-DC+aPD1 groups during the first 32 days (**Fig. 3e**) were attributed to two non-responder mice in the ACT-DC+aPD1 group that developed large tumors (**Fig. S22**).”

8. Figures 3e-h and 3i-j refer to the same cohort of mice, with the entire treatment period lasting 182 days. The survival curves shown in Figures 3f and 3h indicated a higher mortality rate in the ACT-DC treatment group (comprising a total of 15 mice). It is necessary to confirm whether the number of mice in each group is sufficient to support the statistical analysis of the final survival curve experiment presented in Figure 3j. Additionally, the caption should specify the exact number of surviving mice in each group.

Response: We thank the reviewer for this comment. We have clearly specified the exact number of surviving mice in each group in the figure captions and removed the statistics of the survival curve of the naïve group vs ACT-DC group in Fig. 3i and 3j because there are 2 surviving mice in the ACT-DC group.

9. The TEM statistical data in Figures 4h and 4i are inconsistent. The statistical methods used should be detailed in the methods section to prevent misunderstanding. Additionally, it should be clearly indicated whether Figures 4i and 4j report the total number of TCM and TEM in lymph nodes and spleens, or the cell count per unit weight of tissue.

Response: We thank the reviewer for this comment. Fig. 4h shows representative percentage of memory CD8 T cells while Fig. 4i shows the total number of central and effector memory CD8 T cells in the tDLN. So, these statistical data in Fig. 4h and 4i are different. Per the reviewer's suggestion, we have added detailed information on how the TEM number in Fig. 4i is calculated in the method section. In Fig. 4i and 4j, the reported TCM and

TEM numbers were total cell numbers in the lymph node and spleen. We have further clarified this in the figure captions.

“On day 17, tumor, spleen, and tDLNs of mice were collected, weighed, and processed to single cells. The total number of cells obtained from each whole tumor/spleen/tDLN were counted and recorded. The cells were washed with PBS, and 1.5 million cells were stained with antibodies with titrated concentrations.....The total number of a specific cell population (e.g., effector memory CD8 T cell) in the whole spleen or tDLN was calculated using the following method. First, the percentage of the specific cell population within live single cells (denoted as A) was determined by analyzing flow cytometry data using FlowJo 10. The total number of the specific cell population per whole organ was then calculated as the product of A and the total number of live cells in that organ. The total number of a specific cell population in 100 mg tumor was calculated using a similar method.”

10. In Figure S8b, the AC-NP and NP^{Neg} exhibited stronger cytotoxicity towards DOX-pretreated MC38 tumor cells compared to the NPPEG group. More discussion should be added to explain this phenomenon.

Response: We thank the reviewer for this suggestion. While the exact mechanism for AC-NP resulting in increased cell death in doxorubicin-treated MC38 cells remain to be further studied in future studies, we speculate this could be because these doxorubicin-treated cells are vulnerable to AC-NP induced cell membrane destabilization or rupture. While healthy cells can be resistant to such effects, doxorubicin-treated cells are sensitive to such effects. NP^{PEG} induced less cell death compared to AC-NP or NP^{Neg}, possibly because PEGylation is known to reduce nanoparticle interaction and binding with cells, leading to less binding to cell membranes and binding-induced membrane destabilization.

We have added more discussions to explain the possible reasons for the stronger cytotoxicity toward MC38 tumor cells of AC-NPs and NP^{Neg}.

“Additionally, AC-NP caused increased cell death in doxorubicin-treated MC38 tumor cells (**Fig. S11b**), likely due to their vulnerability to NP-binding, which could induce cell membrane destabilization or rupture. While healthy MC38 cells appear resistant to these effects (**Fig. S11a**), doxorubicin-treated cells are more sensitive. Notably, NP^{PEG} caused less cell death than AC-NP or NP^{Neg}, likely because PEGylation reduces nanoparticle interaction and binding with cells, thereby minimizing binding-induced membrane destabilization.”

11. There are several errors in the manuscript, such as the y-axis label “100 gm tumor” in Figure 5e, the incorrect labeling of primary and distant tumors in Figures 6i and 6m, and the absence of a fluorescence intensity scale bar in Figure S11a, and etc..

Response: We thank the reviewer for pointing out these errors. We have corrected these errors accordingly and checked the entire manuscript to correct similar errors. We have added fluorescence intensity scale bars to all LagoX images.

Reviewer #3 (Remarks to the Author): with expertise in cancer immunology

In this manuscript, Chao et al. engineered and characterized novel antigen capturing nanoparticles (AC-NPs) designed to capture tumor antigens in situ to efficiently transfer them to the antigen-presenting conventional dendritic cells type 1 (cDC1s), hence inducing T-cell activation and anti-tumor immunity. Given the rarity of cDC1s within the tumor microenvironment (TME), the authors have combined AC-NPs with the adoptive transfer of bone marrow-derived cDC1s and named the approach Antigen Capturing nanoparticles Transformed Dendritic Cell therapy (ACT-DC). ACT-DC synergy with the immune check point blocker (ICB) anti-programmed cell death (PD) 1 induced systemic anti-tumor immunity across multiple tumor models, including the immunogenic MC38 model and the less immunogenic B16 and CT-2A models. ACT-DC induced cDC1 activation and migration toward the lymph node (LN), which reshaped the TME and the LN, leading to an increase in the numbers of IFN-

γ^+ CD8⁺ and IFN- γ^+ CD4⁺ T cells. In general, the therapeutic efficacy of ACT-DC is good, the figures are informative, and the experiments are well-designed. Although this study is relevant, there are some concerns that need to be clarified or experimentally addressed.

Response: We appreciate the reviewer's constructive suggestions to further improve our manuscript. We have carefully considered and addressed all the reviewer's comments and incorporated relevant changes in the revised manuscript.

Major concerns:

1. The title of Fig. 3 is misleading. The authors claim that ACT-DC eradicated small tumors as a standalone therapy, however, the tumors were pre-treated with doxorubicin. Also, in the text, the mentioning "standalone treatment" is misleading since mice are treated with dox. What would the ACT-DC therapeutic efficacy give without dox?

Response: We thank the reviewer for this comment. We have revised the wording to avoid potential misleading statements. In addition, we have conducted a new experiment to assess ACT-DC's efficacy without doxorubicin pre-treatment. Our data suggests that without doxorubicin pre-treatment, ACT-DC also shows robust therapeutic efficacy leading to 42.9% tumor free survival, although this efficacy is not as potent as ACT-DC plus doxorubicin treatment. We have added the new data (**Fig. S40**) and corresponding discussions to the revised manuscript.

“Notably, in both the B16F10 and MC38 models, tumors were pre-treated with intratumoral doxorubicin before ACT-DC therapy to promote tumor antigen release. We conducted a comparative study to assess the impact of doxorubicin pre-treatment and its administration route on ACT-DC's therapeutic efficacy (Fig. S40a-d**). Even without doxorubicin pre-treatment, ACT-DC significantly delayed tumor growth and achieved tumor-free survival in 42.9% of treated mice, although its efficacy was less potent than in doxorubicin pre-treated mice. Moreover, the route of doxorubicin administration did not significantly influence ACT-DC's efficacy. Pre-treatment with either intravenous or intratumoral doxorubicin before ACT-DC therapy resulted in comparable tumor growth inhibition and an 85.7% tumor-free survival rate.”**

Fig. S40. Impact of doxorubicin pre-treatment on the therapeutic efficacy of ACT-DC in the B16F10 melanoma model. a, Schematic showing the treatment schedule. **b**, Overall tumor growth curve. **c**, Individual tumor growth curve. **d**, Survival curve. Data in **(b)** are expressed as mean \pm SEM. Statistical analysis in **b** was performed using two-way ANOVA: *** $P < 0.001$, **** $P < 0.0001$. Statistical analysis in **d** was performed using Mantel-Cox tests: *** $P < 0.001$.

Editorial note: Panel a created in BioRender. Zhao, Z. (2025) <https://BioRender.com/3ow34vf>

2. AC-NP was shown to result in increased cell death in doxorubicin-treated MC38 cancer cells. Why is this the case? An explanation or speculation would be helpful.

Response: We thank the reviewer for this question. While the exact mechanism for AC-NP resulting in increased cell death in doxorubicin-treated MC38 cells will be further studied in future studies, we speculate this could be because these doxorubicin-treated cells are vulnerable to AC-NP induced cell membrane destabilization or rupture. While healthy cells can be resistant to such effects, doxorubicin-treated cells are sensitive to such effects. NP^{PEG} induced less cell death compared to AC-NP or NP^{Neg}, possibly because PEGylation is known to reduce nanoparticle interaction and binding with cells, leading to less binding to cell membranes and binding-induced membrane destabilization.

We have added more discussions to explain the possible reasons in the revised manuscript.

“Additionally, AC-NP resulted in increased cell death in doxorubicin-treated MC38 tumor cells (**Fig. S11b**), likely due to their vulnerability to NP-binding, which could induce cell membrane destabilization or rupture. While healthy MC38 cells appear resistant to these effects (**Fig. S11a**), doxorubicin-treated cells are more sensitive. Notably, NP^{PEG} caused less cell death than AC-NP or NP^{Neg}, likely because PEGylation reduces nanoparticle interaction and binding with cells, thereby minimizing binding-induced membrane destabilization.”

3. The impact of ACT-DC on specific CD8⁺ T-cell populations remains unclear. It is recommended to integrate CD8⁺ T-cell differentiation, exhaustion and cytotoxic markers, including TCF1, PD1, TIM3, Lag3, perforins and granzymes into the flow cytometry analysis.

Response: We thank the reviewer for this comment. Per the reviewer’s suggestion, we have performed a new study to assess CD8 T cell differentiation, exhaustion, and cytotoxic markers in the tumor and tDLN. Our data indicate that in both tDLNs and tumors, ACT-DC (alone or in combination with aPD1) significantly increased the number of Granzyme B⁺ and perforin⁺ CD8 T cells, TCF-1⁺ CD8 T cells, and proliferating Ki67⁺ CD8 T cells (**Fig. S27 and S30**).

We also tested the exhaustion markers on tumor infiltrating T cells (**Fig. S32**). Overall, ACT-DC, either alone or in combination with anti-PD1 antibody, did not markedly increase T cell exhaustion. Specifically, ACT-DC (with or without anti-PD1 antibody) did not significantly increase the expression of PD-1, TIM-3, or LAG-3 on intratumoral CD4 T cells. As for intratumoral CD8 T cells, ACT-DC (with or without anti-PD1 antibody) treatment significantly increased the expression of TIM-3 but not LAG-3 or PD-1. As TIM-3 expression is elevated on CD8 T cells after ACT-DC treatment, combination of ACT-DC with TIM-3 blockade may provide a new strategy to further enhance the therapeutic efficacy.

The new data (**Fig. S27, Fig. S30, and Fig. S32**) and corresponding discussions have been added to the revised manuscript.

“Furthermore, ACT-DC also resulted in an enhanced adaptive immune response, as evidenced by the increased number of T cells (**Fig. 4e**), antigen-specific CD8 T cells (**Fig. 4f**), IFN- γ ⁺ CD8 T cells and Th1 cells (**Fig. 4g**), Granzyme B⁺ and perforin⁺ CD8 T cells (**Fig. S27a-b**), TCF-1⁺ CD8 T cells (**Fig. S27c**), and proliferating Ki67⁺ CD8 T cells (**Fig. S27d**). A 3.4-4.1-fold higher number of antigen-specific CD8 T cells against two different epitopes (Adpgk and Rpl18) were observed in the ACT-DC group compared to the PBS group (**Fig. 4f and Fig. S27e**).”

“ACT-DC significantly increased the infiltration of CD4 and CD8 T cells, including Th1 cells, effector CD8 T cells (IFN- γ ⁺ CD8 T cells, Granzyme B⁺ CD8 T cells, and perforin⁺ CD8 T cells), TCF-1⁺ CD8 T cells, proliferating Ki67⁺ CD8 T cells, and antigen-specific CD8 T cells against two different epitopes (Adpgk and Rpl18), compared to the PBS treatment (**Fig. 5b-e and Fig. S30**).”

“Furthermore, data from a separate study indicates that ACT did not significantly increase the expression of T cell exhaustion markers (LAG-3, TIM-3, and PD-1) on intratumoral CD4 T cells (**Fig. S32a-c**). However, ACT-DC led to a significant increase in TIM-3 expression on intratumoral CD8 T cells, without affecting LAG-3 or PD-1 levels (**Fig. S32d-f**). These findings suggest that combining ACT-DC with TIM-3 blockade could be a promising strategy to further enhance the therapeutic efficacy of ACT-DC.”

Fig. S27. Effector, proliferating, and antigen-specific CD8 T cells in the tDLNs after two doses of ACT-DC therapies in the MC38 tumor model. a, Granzyme B+ CD8 T cells. b, Perforin+ CD8 T cells. c, TCF-1+ CD8 T cells. d, Ki67+ CD8 T cells. e, Rpl18 tetramer specific CD8 T cells. Statistical analysis was performed using one-way ANOVA followed by Dunnett test: * $P < 0.05$, ** $P < 0.01$. Data in this figure were collected from a separate study according to the same experimental schedule shown in Fig. 4a.

Fig. S30. Effector, proliferating, and antigen-specific CD8 T cells in the tumor after two doses of ACT-DC therapies in the MC38 tumor model. a, Granzyme B+ CD8 T cells. b, Perforin+ CD8 T cells. c, TCF-1+ CD8 T cells. d, Ki67+ CD8 T cells. e, RPL18 tetramer positive CD8 T cells. Statistical analysis was performed using one-way ANOVA followed by Dunnett test: * $P < 0.05$, ** $P < 0.01$, * $P < 0.001$. Data in this figure were collected from a separate study according to the same experimental schedule shown in Fig. 5a.**

Fig. S32. Expression of exhaustion markers on intratumoral CD4 and CD8 T cells after two doses of ACT-DC therapy in the MC38 tumor model. a-c, Relative expression levels of LAG-3 (a), TIM-3 (b), and PD-1 (c) on CD4 T cells in tumors. **d-f,** Relative expression levels of LAG-3 (d), TIM-3 (e), and PD-1 (f) on CD8 T cells in tumors. Statistical analysis was performed using one-way ANOVA followed by Dunnett test: * $P < 0.05$, ** $P < 0.01$, n.s. indicates not significantly different. Data in this figure were collected from a separate study according to the same experimental schedule shown in Fig. 5a. Fig. S27, Fig. S30, and Fig. 32 were from the same study.

4. Given that ACT-DC increased the numbers of CD8+ T cells, CD4+ T cells, IFN- γ + CD8+ T cells and IFN- γ + CD4+ T cells within the TME, evaluating the impact of CD8+ and/or CD4+ T cell-depletion on the therapeutic efficacy of the ACT-DC would be necessary to assess their role in the therapeutic efficacy.

Response: We thank the reviewer for this suggestion. We have conducted a new experiment to assess the impact of CD8 or CD4 T cell depletion on the therapeutic efficacy of ACT-DC in the B16F10 model. Our data suggest that either CD8 or CD4 depletion significantly reduced the effectiveness of ACT-DC, indicating both CD8 and CD4 T cells are critical for the efficacy of ACT-DC. Notably, CD8 T cell depletion showed a larger impact than CD4 T cell depletion, indicating CD8 T cells potentially play a more profound role than CD4 T cells. We have added the new data (**Fig. 6r-s**) and corresponding discussion to the revised manuscript.

“To investigate the roles of CD4 and CD8 T cells, as well as their egress from lymph nodes, on ACT-DC’s therapeutic efficacy, we treated mice with ACT-DC followed by the administration of anti-CD4 antibody, anti-CD8 antibody, or the T cell egress inhibitor FTY720 (an S1PR inhibitor) (**Fig. 6r**). Depletion of either CD4 or CD8 T cells significantly reduced the efficacy of ACT-DC (**Fig. 6s**), indicating the essential role of both T cell subsets in the ACT-DC approach. Notably, CD8 T cell depletion resulted in a more dramatic reduction in therapeutic efficacy compared to CD4 T cell depletion, suggesting a potentially more critical function of CD8 T cells in ACT-DC’s effectiveness.”

Fig. 6. r-s, Effect of T cell depletion and T egress inhibition on the therapeutic efficacy of ACT-DC. r, Schedule of the T cell depletion and egress inhibition study. s, Survival curve of mice in the T cell depletion and egress inhibition study.

Editorial note: Panel a created in BioRender. Zhao, Z. (2025) <https://BioRender.com/3ow34vf>

5. Given that ACT-DC increased the numbers of tetramer+ CD8+ T cells, CD8+ T cells, CD4+ T cells, IFN- γ + CD8+ T cells and IFN- γ + CD4+ T cells within the tdLN, it is intriguing to assess if the anti-tumor efficacy is dependent on T-cell trafficking and T-cell egress from the LN. To this the end, the therapeutic efficacy of ACT-DC could be evaluated in the presence or absence of the S1PR inhibitor FTY720.

Response: We thank the reviewer for this suggestion. In response to this comment, we conducted a new experiment to compare the therapeutic efficacy of ACT-DC in the presence or absence of FTY720 in the B16F10 model. Our data suggests that FTY720 treatment also significantly impaired ACT-DC's efficacy, indicating that T cell migration and egress from lymph nodes are essential for ACT-DC's therapeutic effectiveness. We have added the new data (Fig. 6r-s) and corresponding discussion to the revised manuscript.

“FTY720 treatment also significantly impaired ACT-DC's efficacy, indicating that T cell trafficking and egress from lymph nodes are essential for ACT-DC's therapeutic effectiveness.”

6. It would also be interesting to assess the contribution of the tumor-residing cDC1s compared to the transferred cDC1s, for example by doing the therapy experiment in *Batf3*KO or XCR1-DTR mice.

Response: We thank the reviewer for this comment and agree that understanding the role of endogenous cDC1s in the therapeutic efficacy of ACT-DC is important. In response to this suggestion, we have conducted an additional experiment to compare the therapeutic efficacy of ACT-DC in wild-type versus *Batf3*KO mice (lacking endogenous cDC1s). Our data reveal that in *Batf3*KO mice which lack endogenous cDC1s, ACT-DC also showed significant therapeutic efficacy in delaying tumor growth. However, ACT-DC in *Batf3*KO mice did not show as strong therapeutic efficacy as in wild-type mice. Specifically, ACT-DC achieved complete tumor-free survival in 85.7% wild-type mice, while in *Batf3*KO mice, the respective tumor-free survival rate is 33.3%. These data suggest that endogenous cDC1s also contribute to the therapeutic efficacy of ACT-DC. We have added the new data (Fig. 6t and Fig. S39) and corresponding discussions in the revised manuscript.

“To evaluate the contribution of endogenous cDC1s to ACT-DC's efficacy, we conducted therapeutic studies in *Batf3*^{-/-} mice, which lack endogenous cDC1s (Fig. S39a and Fig. 6t). While ACT-DC treatment in *Batf3*^{-/-} mice significantly delayed tumor growth (Fig. S39b) and improved survival rates (Fig. 6t), its therapeutic efficacy was markedly reduced compared to that in the wide-type mice. These data indicate that in addition to the adoptively transferred cDC1s, endogenous cDC1s are also critical to the success of the ACT-DC approach.”

Fig. S39. Effect of endogenous cDC1s on the therapeutic efficacy of ACT-DC in the B16F10 tumor model. **a**, Schedule of the study. **b**, Tumor growth curve. Data are expressed as mean \pm SEM. Statistical analysis was performed using two-way ANOVA: * $P < 0.05$, ** $P < 0.01$, **** $P < 0.0001$.

Editorial note: Panel a created in BioRender. Zhao, Z. (2025) <https://BioRender.com/3ow34vf>

Fig. 6. t, Effect of endogenous cDC1s on the therapeutic efficacy of ACT-DC. Survival curves of wild-type or $Batf3^{-/-}$ mice after ACT-DC treatment are shown.

Minor concerns:

1. For full transparency, please also show the pre-gating strategy in fig S4. Since the expression of Clec9a is relatively low, to which extent does the gated population also contain cDC2s or monocyte-derived cells?

Response: We thank the reviewer for this comment. Per the suggestion, we have added the pre-gating strategy in Fig. S4. Despite the Clec9a expression on the cultured CD103+ cDC1s is relatively low, the expression of CD103 on these cells (a characteristic migratory cDC1 marker) is high. Specifically, 92.6% of the obtained CD103+ cDC1s are CD103 positive, so the other DC populations such as cDC2s or monocyte-derived DCs are only in a small portion. We have added more discussions in the revised manuscript for further clarification.

“Specifically, 92.6% of the obtained cells are CD103-positive, indicating that CD103+ cDC1s constitute the majority, while other DCs, such as cDC2s and monocyte-derived DCs, represent only a small fraction.”

2. In the legend of Fig. 2b-c, it says that the mice were either untreated or injected with ACT-DC, but in the bar graph it says otherwise. Were they untreated or treated with AC-NP?

Response: Thank you for pointing out this inconsistency. They were untreated, and we have corrected this in the revised Fig. 2b-c.

3. Line 167: replace “macrophage” by “myeloid cell” in the text, as these could also be neutrophils or other CD11b+CD11c cells.

Response: We thank the reviewer for this comment. We have replaced “macrophages” with “myeloid cells” per the suggestion.

4. Add statistics in Fig S8.

Response: We thank the reviewer for this suggestion. We have added statistics to Fig. S8 (new **Fig. S11**).

5. In Fig. S14 and S15, the ACT-DC-treated group had a total of 15 mice while all the other treatment groups had 7-8 mice each. This can be misleading and the reasons behind this difference should be pointed out.

Response: We thank the reviewer for this question. The reason for having double the number of mice in the ACT-DC group as compared to other groups was to obtain adequate number of surviving mice for re-challenging studies. We have added further clarifications in the revised manuscript.

“Additionally, we used more mice (15) in the ACT-DC group than the other treatment groups (7-8) in this study to ensure an adequate number of surviving mice for subsequent re-challenge studies.”

6. The arrangement of the panels in Fig. 4 is slightly confusing and in 4h it should be “ACT-DC” instead of “ACD-DC”.

Response: We thank the reviewer for pointing out this error. We have corrected accordingly and re-arranged the panels of Fig. 4 to avoid confusion.

Reviewer #4 (Remarks to the Author): with expertise in cancer immunology

Chao et al have demonstrated a novel therapy called ACT-DC which consists of antigen capturing nanoparticles that enhance migratory cDC1 activation and T cell immune responses. They demonstrated this with various tumor models (MC38, B16F10, B16F10-OVA and CT-2A) which reinforced the strength of such therapy. However, some experiments were not done under physiological conditions, hence limiting the strength of some conclusions. Moreover, some more experiments should be done to fully demonstrate the efficacy of the ACT-DC.

Response: We appreciate the reviewer’s constructive suggestions to improve our manuscript. We have carefully considered and addressed all the reviewer’s comments and incorporated relevant changes in the revised manuscript.

Major comments:

1) In figure 1E, could the authors please specify how they were able to quantify the specific number of proteins retained? Especially for tumor lysate.

Response: We thank the reviewer for this question. Per the reviewer's suggestion, we have specified the method to quantify the amount of proteins bound to nanoparticles in the method section of the revised manuscript.

“Evaluation of the antigen capturing efficiency of NPs. The antigen capturing ability of different NPs was studied using the model antigen ovalbumin (AF647-OVA) and MC38 tumor lysates. The protein content in the tumor lysate was pre-quantified using a bicinchoninic acid (BCA) assay. In brief, 500 µg of NPs were added to a solution containing different concentrations of AF647-OVA or tumor lysates and incubated at 37°C for 30 minutes. Following incubation, the mixture was centrifuged at 12,000 g, and the pellet containing NPs bound with proteins was collected. The unbound protein in the supernatant was quantified using a BCA assay (for tumor lysates) or a fluorescence-based method using plate reader (for AF647-OVA). The amount of protein bound to the NPs was determined by subtracting the unbound protein from the total feed protein. The charge and size of NPs after protein binding were measured using DLS.”

2) In figure 1, the authors examined the uptake of the ACT-DC with cDC1. However, this model was very subjective since the BMDCs were forced to generate cDC1. It would be interesting if the authors could test this in a more physiological representation. In general, cDC2 exceeded the amount of cDC1 in both humans and mice. Hence, could the authors re-test the uptake of the ACT-DC in mixture of cDC1 and cDC2. This would provide an actual advantage of this system.

Response: We thank the reviewer for this question. We agree with the reviewer that there are more cDC2s than cDC1s in human and mouse tumors. However, in the ACT-DC approach, cDC1s are intratumorally administered, which alters the ratio of cDC1s to cDC2s within the tumor. Our data shown in Fig. 5f-i indicate that ACT-DC treatment changed the intratumoral DC profiles to be cDC1 dominant. Therefore, the use of cDC1s alone should be a simplified and physiologically relevant method for evaluating AC-NP enhanced antigen delivery to cDC1s.

We agree with the reviewer that it is important to understand the antigen delivery to cDC1s in a more physiologically relevant setting. Therefore, we conducted a new experiment to test this. In this experiment, instead of mixing cDC1 with cDC2, we used a mixture of cells dissociated from a MC38 tumor, which closely represents the diversity of cells within the actual tumor microenvironment, to co-incubate with AC-NPs and test the relative uptake of AC-NPs to cDC1 versus cDC2. As shown in **Fig. S13**, our data indicates that AC-NPs were more efficiently taken up by cDC1s over cDC2s, indicating the capability of AC-NPs to better target cDC1s than cDC2s.

Moreover, as shown in Fig. 2j-n, our data also demonstrated that ACT-DC could enhance model antigen delivery to cDC1s as well as cDC1 activation in the *in vivo* setting. All the above data collectively showed the advantage of ACT-DC.

We have added the new data (**Fig. S13**) and relevant discussions in the revised manuscript.

“Moreover, when co-incubated with a mixture of cells which were dissociated from a MC38 tumor, AC-NPs were more efficiently taken up by cDC1s than by cDC2s (Fig. S13).”

Fig. S13. AC-NPs showed preferable uptake to cDC1s versus cDC2s when co-incubated with a cell mixture dissociated from a MC38 tumor for 4 hours. AC-NPs were labeled by DiD. After co-incubation, the cell mixture was stained for CD45, CD11c, CD11b, and CD103. a, Representative flow cytometry graphs showing the gating strategy for cDC1/cDC2 and relative uptake of AC-NPs by cDC1/cDC2. b, MFI of DiD-labeled AC-NPs taken up into cDC1 versus cDC2. Statistical analysis in (b) was performed using two-tail student's t test: ** $P < 0.01$.

3) With regard to in vivo trafficking of ACT-DC and cell-level distribution of AC-NPs, why was AF647-OVA (50 μg) administered? B16OVA by itself should be expressing OVA. This system is again forcing certain results and not demonstrating physiological/clinical conditions.

Response: We thank the reviewer for this question. AF647-OVA was administered as a model antigen for fluorescent tracking purposes to assess the delivery of antigens to lymph nodes. Although OVA-transduced tumor cell lines like B16OVA express OVA, these proteins can't be fluorescently tracked for biodistribution studies. Injecting fluorescent model antigens like OVA into tumors for tumor antigen distribution studies is a frequently used method. Despite not truly in a physiological condition, this method closely mimics tumor antigens in tumor physiological environment and provides a feasible approach to study tumor antigen distribution. In response to the reviewer's comment, we have added discussions in the revised manuscript to acknowledge the limitation of this method.

“In this study, we injected AF647-labeled OVA as a model tumor antigen to fluorescently track its delivery to cDC1s and lymph nodes (Fig. 2j). Notably, we acknowledge that AF647-OVA is not an endogenous tumor antigen, although its use as a model antigen could provide a feasible method for monitoring antigen delivery.”

4) Why is doxorubicin injected intratumorally (IT)? The normal dose of such therapy in the clinic IV. Moreover, nanoparticles are given IV also in the clinic. These conditions are not realistic. Could the authors please repeat therapeutic experiments where therapies are administered according to FDA/EMA accepted routes?

Response: We thank the reviewer for this very helpful comment. We agree with the reviewer that many nanoparticles (e.g., Doxil and Abrexane) are often administered intravenously in the clinic. However, local administration including intratumoral injection of nanoparticles are also approved by FDA for cancer treatment. For example, intratumoral administration of NBTXR3, a hafnium oxide nanoparticle, was approved for treating advanced squamous cell carcinoma. The AC-NPs reported in our manuscript are designed to capture native tumor antigens *in situ*. They were intratumorally administered to ensure optimal access to the intratumoral antigens. We have added more discussions to clarify this point in the revised manuscript.

“As AC-NPs are designed to capture native tumor antigens *in situ*, their intratumoral administration can maximize direct access to tumor antigens within the tumor microenvironment. While many clinically approved NPs (e.g., Doxil and Abrexane) are often administered intravenously, some NPs (e.g., NBTXR3, a hafnium oxide nanoparticle) have also been approved for intratumoral injection in cancer therapy.”

In this work, doxorubicin was injected intratumorally to promote tumor antigen release. We agree with the reviewer that doxorubicin is most often injected intravenously in the clinic. In response to the reviewer’s suggestion, we have conducted a new *in vivo* experiment to investigate the therapeutic efficacy of ACT-DC with intravenous doxorubicin. Our data indicates that ACT-DC with intravenous doxorubicin showed similar efficacy to ACT-DC with intratumoral doxorubicin in the B16F10 model. We have added this new data (Fig. S40) and related discussions to the revised manuscript.

“Notably, in both the B16F10 and MC38 models, tumors were pre-treated with intratumoral doxorubicin before ACT-DC therapy to promote tumor antigen release. We conducted a comparative study to assess the impact of doxorubicin pre-treatment and its administration route on ACT-DC’s therapeutic efficacy (Fig. S40a-d). Even without doxorubicin pre-treatment, ACT-DC significantly delayed tumor growth and achieved tumor-free survival in 42.9% of treated mice, although its efficacy was less potent than in doxorubicin pre-treated mice. Moreover, the route of doxorubicin administration did not significantly influence ACT-DC’s efficacy. Pre-treatment with either intravenous or intratumoral doxorubicin before ACT-DC therapy resulted in comparable tumor growth inhibition and an 85.7% tumor-free survival rate.”

Fig. S40. Impact of doxorubicin pre-treatment on the therapeutic efficacy of ACT-DC in the B16F10 melanoma model. a, Schematic showing the treatment schedule. b, Overall tumor growth curve. c, Individual tumor growth curve. d, Survival curve. Data in (b) are expressed as mean \pm SEM. Statistical analysis in (b) was performed using two-way ANOVA: * $P < 0.001$, **** $P < 0.0001$. Statistical analysis in (d) was performed using Mantel-Cox tests: *** $P < 0.001$.**

Editorial note: Panel a created in BioRender. Zhao, Z. (2025) <https://BioRender.com/3ow34vf>

5) In the therapeutic models, could the authors shed some light on the activation status of the DC cells? It was shown in previous experiments and even hypothesized that ACT-DC could increase activation markers for DCs.

Response: We thank the reviewer for this suggestion. ACT-DC indeed relies on AC-NPs to activate cDC1s to trigger their migration to tDLNs for antigen presentation and T cell priming. Per the reviewer's suggestion, we analyzed the activation status of DCs in tDLNs in the MC38 therapeutic model according to the experimental schedule shown in Fig. 4a. As shown in Fig. S26, our data suggests that compared to PBS treatment, ACT-DC, either alone or in combination with aPD1, increased the expression of DC activation marker CD86 on cDC1s in the tDLN by 1.8 to 1.9-fold. Additionally, ACT-DC led to an increase in CD86 expression on cDC2s compared to PBS, although this difference is not significantly different. The reason why ACT-DC showed a more profound effect on cDC1 activation than cDC2 activation might be due to AC-NPs mainly targeting adoptively injected cDC1s in the tumor as shown in Fig. 2g.

We have added the new data (Fig. S26) and related discussions to the revised manuscript.

“Additionally, compared to PBS treatment, ACT-DC, either alone or in combination with aPD1, increased the expression of DC activation marker CD86 on cDC1s in the tDLN by 1.8- to 1.9-fold (Fig. S26a-b). ACT-DC also led to an increase in CD86 expression on cDC2s compared to PBS, although this difference is not significantly different (Fig. S26c).”

Fig. S26. Activation status of cDC1 and cDC2 in the tDLNs after two doses of different therapies. a, Representative flow plot showing the expression of CD86 on CD103+CD11c+ cDC1s in the tDLNs. b, MFI of CD86 on CD103+CD11c+ cDC1s in tDLN. c, MFI of CD86 on CD11b+CD11c+ cDC2s in tDLNs. Statistical analysis was performed using one-way ANOVA followed by Dunnett test: * $P < 0.001$.**

6) In figure 3, the authors have nice data concerning the memory phenotyping of CD8+ T cells and tetramer staining. Could the authors correlate the enhanced TEM population with the tetramer staining. This would really provide solid information that the ACT-DC therapy is inducing a tumor specific TEM population.

Response: We thank the reviewer for this suggestion. This is a very important point. Per the reviewer's comment, we have further analyzed our flow cytometry data to identify antigen-specific (Adpgk tetramer positive) effector CD8 T cells in the blood. As shown in **Fig. S24a-b**, the long-term survivor mice treated by ACT-DC+aPD1 showed a 2.1-fold and 2.0-fold higher number of antigen-specific CD8 T_{EM} and antigen-specific CD8 T_{CM} compared to the naïve tumor mice, respectively. This indicates ACT-DC indeed induced tumor-specific memory CD8 T cell populations.

We have added the new data (**Fig. S24a-b**) and related discussions in the revised manuscript.

“In comparison to age-matched naïve mice, mice treated with ACT-DC plus aPD1 showed elevated levels of antigen-specific CD8 T cells (Fig. 3k-l**), memory CD8 T cells (**Fig. 3m-n**), antigen-specific memory CD8 T cells (**Fig. S24a-b**), memory CD4 T cells (**Fig. S24c-d**), along with reduced numbers of immunosuppressive cells, including regulatory T cells (Tregs) and myeloid-derived suppressor cells (MDSCs) (**Fig. S24e-j**).”**

Fig. S24. Immune cell profiles in the blood of mice 15 days after the 2nd rechallenge in the large-established MC38 tumor model. a-b, Representing flow gating strategy (a) and quantification (b) of Adpgk tetramer positive effector memory and central memory CD8 T cells in the blood.

7) In figure 4 and 5, the authors are representing some graphs with cell number. However, in the methods it is not described how this is done. Was this done with counting beads or what other methods were used to quantify the number of cells?

Response: We thank the reviewer for this question. In response to the reviewer's comment, we have added a detailed description of the method for calculating cell numbers in tumor, lymph nodes, and spleen to the Method section.

“On day 17, tumor, spleen, and tDLNs of mice were collected, weighed, and processed to single cells. The total number of cells obtained from each whole tumor/spleen/tDLN were counted and recorded. The cells were washed with PBS, and 1.5 million cells were stained with antibodies with titrated concentrations.....The total number of a specific cell population (e.g., effector memory CD8 T cell) in the whole spleen or tDLN was calculated using the following method. First, the percentage of the specific cell population within live single cells (denoted as A) was determined by analyzing flow cytometry data using FlowJo 10. The total number of the specific cell population per whole organ was then calculated as the product of A and the total number of live cells in that organ. The total number of a specific cell population in 100 mg tumor was calculated using a similar method.”

8) The authors show a nice increase of TEM CD4+ population with B16F10. However, they do not demonstrate this with MC38. Could this be done?

Response: We thank the reviewer for this question. The CD4 memory T cell data is shown in **Fig. S24c-d**. Our data indicate that after rechallenge, ACT-DC+aPD1 treated mice showed a markedly higher percentage of effector

memory and central memory CD4 T cells compared to naïve tumor mice. Please note that we didn't include the ACT-DC group in this study (Fig. 3k-n and Fig. S24) because only two mice remained survival after the rechallenge which is not sufficient for statistical analysis.

Fig. S24. Immune cell profiles in the blood of mice 15 days after the 2nd rechallenge in the large-established MC38 tumor model. c-d, Representative flow plot and quantification of memory CD4 T cells in the blood.

9) In figure 6i, how sure are the authors that the ACT-DC therapy induced a strong and effective T cell populations against both OVA or TRP2? Could it not be that the TRP2 specific T cells induced tumor killing in the B16F10-OVA since they also express TRP2? The authors should try to isolate the TRP2 and OVA specific T cells to demonstrate that the therapy did indeed induce two powerful T cell populations against two different epitopes. This would show a clear mitigation of immunodominance and advantage to combat tumor heterogeneity.

Response: We thank the reviewer for this question. In the bilateral tumor model, only the B16F10-OVA tumor was treated with ACT-DC while the B16F10 tumor was not treated. We agree with the reviewer that the shrinkage of the B16F10-OVA tumor could not directly indicate ACT-DC induced two antigen-specific CD8 T cells. However, we collected the B16F10-OVA draining lymph node and found that ACT-DC treatment significantly increased the number of both OVA tetramer+ CD8 T cells and TRP-2 tetramer+ CD8 T cells (Fig. 6p-q). This indicates that ACT-DC generated CD8 T cells against both OVA and TRP-2. We have rephrased to avoid potential confusion.

Per the reviewer's suggestion, we conducted a new experiment to test antigen-specific CD8 T cells against different epitopes in the tumor draining lymph node and tumor in the MC38 model. Our data (Fig. S27e and Fig. 30e) revealed that, apart from the adpgk-specific T cells, ACT-DC also induced significantly more CD8 T cells against another epitope, Rpl18, in both the tDLNs and the tumor. This new data further provides another evidence in a different tumor model to support that ACT-DC could induce T cell populations against two different epitopes.

We have included the new data and relevant discussions in the revised manuscript.

“A 3.4-4.1-fold higher number of antigen-specific CD8 T cells against two different epitopes (Adpgk and Rpl18) were observed in the ACT-DC group compared to the PBS group (Fig. 4f and Fig. S27e).”

“ACT-DC significantly increased the infiltration of CD4 and CD8 T cells, including Th1 cells, effector CD8 T cells (IFN- γ + CD8 T cells, Granzyme B+ CD8 T cells, and perforin+ CD8 T cells), TCF-1+ CD8 T cells, proliferating Ki67+ CD8 T cells, and antigen-specific CD8 T cells against two different epitopes (Adpgk and Rpl18), compared to the PBS treatment (Fig. 5b-e and Fig. S30)”

Fig. S27. Effector, proliferating, and antigen-specific CD8 T cells in the tDLNs after two doses of ACT-DC therapies in the MC38 tumor model. a, Granzyme B+ CD8 T cells. b, Perforin+ CD8 T cells. c, TCF-1+ CD8 T cells. d, Ki67+ CD8 T cells. **e, Rpl18 tetramer specific CD8 T cells.** Statistical analysis was performed using one-way ANOVA followed by Dunnett test: * P < 0.05, ** P < 0.01.

Fig. S30. Effector, proliferating, and antigen-specific CD8 T cells in the tumor after two doses of ACT-DC therapies in the MC38 tumor model. a, Granzyme B+ CD8 T cells. b, Perforin+ CD8 T cells. c, TCF-1+ CD8 T cells. d, Ki67+ CD8 T cells. **e, RPL18 tetramer positive CD8 T cells.** Statistical analysis was performed using one-way ANOVA followed by Dunnett test: * P < 0.05, ** P < 0.01, *** P < 0.001.

10) Could the authors clarify how were TAMCs phenotyped?

Response: We thank the reviewer for this question. In the glioma studies (Fig. 7), we used CD45 and CD11b to gate and differentiate TAMCs and TILs. Specifically, TAMCs are known to be CD45+CD11b+. We have included a representative flow cytometry gating strategy (Fig. S45) to help clarify this. The phenotype of TAMCs in the glioma model is beyond the scope of this work and will be further studied in future studies. We have added a sentence in the revised manuscript for clarification.

“Notably, the activation status and phenotype of TAMCs following ACT-DC treatment needs to be further investigated in future studies.”

Minor comments:

1) Line 46-49, DCs used to target multiple antigens is not the most effective strategy. Immunodominance can exist and limit multiple T cell responses against different epitopes. Even in the reference used they state this. Hence, I would not word it in such a form but rather saying that DCs can be used to tackle different antigens which may be personalized with different patients.

Response: We thank the reviewer for this comment. We agree with the reviewer and have revised the wording accordingly.

“In contrast to genetically engineered cell therapies like chimeric antigen receptor (CAR) T cells and CAR natural killer (NK) cells, which are tailored to target a single tumor antigen, DC-based cell therapies can be used to tackle different tumor antigens, which can more effectively counteract the tumor's antigen heterogeneity and may be personalized with different patients.”

2) In figure 2m, why is there count on the y-axis? The count should then give you a histogram plot and not what is represented.

Response: We thank the reviewer for bringing this error to our attention. The y-axis should be SSA. We have corrected this in the revised manuscript.

3) In lines 174-175 “These retained cDC1s, activated by AC-NPs, could reshape the local tumor microenvironment, enhancing immune cell infiltration after induction of an antitumor immune response in the tDLNs.” This is discussion and not results. Please remove this and add it to the discussion section.

Response: We thank the reviewer for this suggestion. We agree with the reviewer and have moved this sentence to the discussion section.

4) The authors would further boost the quality of this paper if they could further look at the activated and exhausted profile of the T cells. By simply using PD-1 and TIGIT/LAG3/TIM3 this could be achieved.

Response: We thank the reviewer for this suggestion. We have conducted an additional experiment to assess the proliferation, exhaustion, and cytotoxic status of intratumoral T cells. We have added the new data (**Fig. S27, Fig. S30, and Fig. S32**) and corresponding discussions in the revised manuscript.

“Furthermore, ACT-DC also resulted in an enhanced adaptive immune response, as evidenced by the increased number of T cells (**Fig. 4e**), antigen-specific CD8 T cells (**Fig. 4f**), IFN- γ + CD8 T cells and Th1 cells (**Fig. 4g**), Granzyme B+ and perforin+ CD8 T cells (**Fig. S27a-b**), TCF-1+ CD8 T cells (**Fig. S27c**), and proliferating Ki67+ CD8 T cells (**Fig. S27d**). A 3.4-4.1-fold higher number of antigen-specific CD8 T cells against two different epitopes (Adpgk and Rpl18) were observed in the ACT-DC group compared to the PBS group (**Fig. 4f and Fig. S27e**).”

“ACT-DC significantly increased the infiltration of CD4 and CD8 T cells, including Th1 cells, effector CD8 T cells (IFN- γ + CD8 T cells, Granzyme B+ CD8 T cells, and perforin+ CD8 T cells), TCF-1+ CD8 T cells, proliferating Ki67+ CD8 T cells, and antigen-specific CD8 T cells against two different epitopes (Adpgk and Rpl18), compared to the PBS treatment (**Fig. 5b-e and Fig. S30**).”

“Furthermore, data from a separate study indicates that ACT did not significantly increase the expression of T cell exhaustion markers (LAG-3, TIM-3, and PD-1) on intratumoral CD4 T cells (**Fig. S32a-c**). However, ACT-DC led to a significant increase in TIM-3 expression on intratumoral CD8 T cells, without affecting LAG-3 or PD-1 levels (**Fig. S32d-f**). These findings suggest that combining ACT-DC with TIM-3 blockade could be a promising strategy to further enhance the therapeutic efficacy of ACT-DC.”

Fig. S27. Effector, proliferating, and antigen-specific CD8 T cells in the tDLNs after two doses of ACT-DC therapies in the MC38 tumor model. a, Granzyme B+ CD8 T cells. b, Perforin+ CD8 T cells. c, TCF-1+ CD8 T cells. d, Ki67+ CD8 T cells. e, Rpl18 tetramer specific CD8 T cells. Statistical analysis was performed using one-way ANOVA followed by Dunnett test: * $P < 0.05$, ** $P < 0.01$. Data in this figure were collected from a separate study according to the same experimental schedule shown in Fig. 4a.

Fig. S30. Effector, proliferating, and antigen-specific CD8 T cells in the tumor after two doses of ACT-DC therapies in the MC38 tumor model. a, Granzyme B+ CD8 T cells. b, Perforin+ CD8 T cells. c, TCF-1+ CD8 T cells. d, Ki67+ CD8 T cells. e, RPL18 tetramer positive CD8 T cells. Statistical analysis was performed using one-way ANOVA followed by Dunnett test: * $P < 0.05$, ** $P < 0.01$, * $P < 0.001$. Data in this figure were collected from a separate study according to the same experimental schedule shown in Fig. 5a.**

Fig. S32. Expression of exhaustion markers on intratumoral CD4 and CD8 T cells after two doses of ACT-DC therapy in the MC38 tumor model. a-c, Relative expression levels of LAG-3 (a), TIM-3 (b), and PD-1 (c) on CD4 T cells in tumors. **d-f,** Relative expression levels of LAG-3 (d), TIM-3 (e), and PD-1 (f) on CD8 T cells in tumors. Statistical analysis was performed using one-way ANOVA followed by Dunnett test: * P < 0.05, ** P < 0.01, n.s. indicates not significantly different. Data in this figure were collected from a separate study according to the same experimental schedule shown in Fig. 5a. Fig. S27, Fig. S30, and Fig. 32 were from the same study.